# Enriching Disentanglement:
# From Logical Definitions to Quantitative Metrics

**Yivan Zhang**
The University of Tokyo, RIKEN AIP
Tokyo, Japan
yivanzhang@ms.k.u-tokyo.ac.jp

**Masashi Sugiyama**
RIKEN AIP, The University of Tokyo
Tokyo, Japan
sugi@k.u-tokyo.ac.jp

## Abstract

Disentangling the explanatory factors in complex data is a promising approach for generalizable and data-efficient representation learning. While a variety of quantitative metrics for learning and evaluating disentangled representations have been proposed, it remains unclear what properties these metrics truly quantify. In this work, we establish algebraic relationships between logical definitions and quantitative metrics to derive theoretically grounded disentanglement metrics. Concretely, we introduce a compositional approach for converting a higher-order predicate into a real-valued quantity by replacing (i) equality with a strict premetric, (ii) the Heyting algebra of binary truth values with a quantale of continuous values, and (iii) quantifiers with aggregators. The metrics induced by logical definitions have strong theoretical guarantees, and some of them are easily differentiable and can be used as learning objectives directly. Finally, we empirically demonstrate the effectiveness of the proposed metrics by isolating different aspects of disentangled representations.

## 1 Introduction

In *supervised learning*, we usually use a real-valued cost function $\ell : Y \times Y \to \mathbb{R}_{\geq 0}$ to measure how close an output $f(x)$ of a function $f : X \to Y$ is to a target label $y$, i.e., $\ell(f(x), y)$, to quantify the cost of inaccurate prediction. Then, we can use the *total cost* over a collection of input-output pairs to measure the performance of this function. From a functional perspective, this construction induces a quantitative metric $L : [X, Y] \times [X, Y] \to \mathbb{R}_{\geq 0}$ between functions:[1]

$$L(f, g) := \sum_{x \in X} \ell(f(x), g(x)), \tag{1}$$

where $g : X \to Y$ is a "*ground-truth function*" that maps each input $x$ to its target label $y$. This metric can be used as both *learning objective* and *evaluation metric* for the learning model $f : X \to Y$. What does $L(f, g)$ quantify? It quantifies the extent to which two functions $f$ and $g$ are *equal*:

$$(f =_{[X,Y]} g) := \forall x \in X. \ f(x) =_Y g(x).[2] \tag{2}$$

Considering the equality as a predicate, we can observe a parallel between

- (binary-valued) equality $=_Y : Y \times Y \to \{\top, \bot\}$ and (real-valued) cost $\ell : Y \times Y \to \mathbb{R}_{\geq 0}$,[3]
- universal quantifier ("*for all*") $\forall x \in X$ and summation $\sum_{x \in X}$, and
- function equality $=_{[X,Y]} : [X, Y] \times [X, Y] \to \{\top, \bot\}$ and total cost $L : [X, Y] \times [X, Y] \to \mathbb{R}_{\geq 0}$.

We would like to ask: *Is it possible to measure and optimize other properties in the same way?*

---

[1] $[X, Y]$ denotes the set of all functions from a set $X$ to a set $Y$.

[2] In this paper, the domains of equality predicates are explicitly subscripted.

[3] $\{\top, \bot\}$ denotes the set of binary truth values: *true* $\top$ and *false* $\bot$.

38th Conference on Neural Information Processing Systems (NeurIPS 2024).

In *representation learning* [Bengio et al., 2013], measuring and optimizing the performance of a learning model becomes a non-trivial task. The quality of a model cannot always be measured by how close it is to a fixed ground truth. Instead, we often need to consider the properties of the model architecture or learned representation itself, such as *convexity* [Amos et al., 2017], *uniformity* [Wang and Isola, 2020], *invariance* [Kvinge et al., 2022], and *equivariance* [Lee et al., 2019, Brehmer et al., 2023]. A proper comprehension of what constitutes good representations and how to assess their quality is important for designing suitable models, learning objectives, and evaluation metrics.

**Disentangled representation learning: definitions, metrics, and methods** *Disentanglement* is an important property in representation learning, which intuitively means that different explanatory factors in data should be encoded separately [Bengio et al., 2013]. However, disentanglement has no universally agreed-upon formal definition [Higgins et al., 2018, Suter et al., 2019, Shu et al., 2020, Fumero et al., 2021], and it is typically viewed not as a single property but rather as a combination of several requirements [Ridgeway and Mozer, 2018, Eastwood and Williams, 2018, Do and Tran, 2020, Tokui and Sato, 2022]. While many metrics for measuring disentanglement have been proposed [Carbonneau et al., 2022], it remains unclear what properties these metrics truly quantify and how they can be optimized directly. Often, a new evaluation metric is introduced along with a new learning method, but it is usually unproven that the method can optimize the new metric [Higgins et al., 2017, Kim and Mnih, 2018, Chen et al., 2018, Li et al., 2020]. This lack of theoretical understanding makes it difficult to design learning models that can effectively learn disentangled representations.

**A logical and algebraic approach to defining and measuring disentangled representations** Recently, Zhang and Sugiyama [2023] proposed a general and abstract definition of disentanglement, shedding light on the common structures underlying the algebraic, statistical, and topological definitions of disentanglement. It was shown that the abstract concept of *product* [Mac Lane, 1978] underlies an essential property of disentanglement called *modularity* [Ridgeway and Mozer, 2018], and other properties of learning models, such as *informativeness* [Eastwood and Williams, 2018], can also be defined abstractly using only the composition and identity of morphisms. Following this algebraic approach, we aim to derive theoretically grounded quantitative metrics of disentanglement from the logical definitions of the desired properties, extending the parallel between Eqs. (1) and (2).

**Contributions** In this paper, we focus on logically defined properties of disentangled representation learning, such as modularity and informativeness (Section 2). We introduce a compositional approach to converting a *higher-order equational predicate* into a *real-valued quantity* (Table 1), which serves as a quantitative metric of the extent to which a function satisfies the predicate (Section 3). Our analysis on the relationship between the logical definitions and the induced quantitative metrics provides theoretical guarantee on the properties of the optimal functions (Theorem 1). Then, we demonstrate the usefulness of this conversion method by deriving quantitative metrics for measuring properties of disentangled representations, and we analyze these metrics in terms of computation, optimization, and differentiability (Section 4). Lastly, we compare the derived metrics with several existing ones in a fully controlled experiment and demonstrate that the proposed metrics are able to isolate different aspects of disentangled representations (Section 5).

## 2 Logical definitions of disentangled representations

In this section, let us first take a closer look at the logical definitions of two properties of disentangled representation learning — informativeness [Eastwood and Williams, 2018] and modularity [Ridgeway and Mozer, 2018], which are arguably more important than other properties [Carbonneau et al., 2022]. We limit our discussion to sets and functions, but the generalization to other morphisms, such as equivariant, stochastic, or continuous functions, is straightforward.

### 2.1 Informativeness: injectivity or retractability of a learning model

Being informative, expressive, faithful, or useful is a basic requirement for learned representations [Bengio et al., 2013]. We want a representation learning model to preserve explanatory factors in data that are informative to the downstream tasks. For functions, this criterion could be formulated as follows: If two factors $y$ and $y'$ are different, then their representations $m(y)$ and $m(y')$ extracted by a function $m : Y \to Z$ should be different too. This means that the function $m$ should be *injective*:

**Definition 1** (Injective function). A function $m : Y \to Z$ is *injective* if
$$p_{\text{injective}}(m : Y \to Z) := \forall y \in Y. \, \forall y' \in Y. \, (m(y) =_Z m(y')) \to (y =_Y y'). \tag{3}$$

Alternatively, because injective functions are precisely functions with *retractions* (left inverses) [Lawvere and Rosebrugh, 2003, Chapter 2], we can measure the *retractability* instead:

**Definition 2** (Retractable function). A function $m : Y \to Z$ has a *retraction* $h : Z \to Y$ if
$$p_{\text{retractable}}(m : Y \to Z) := \exists h : Z \to Y. \, h \circ m =_{[Y,Y]} \mathrm{id}_Y. \tag{4}$$

Note that these properties are *predicates* $p_{\text{injective}}, p_{\text{retractable}} : [Y, Z] \to \{\top, \bot\}$ on the set $[Y, Z]$ of all functions from $Y$ to $Z$. Analogous to using the total cost in Eq. (1) to measure function equality in Eq. (2), if we want to measure the *injectivity* in Eq. (3) or *retractability* in Eq. (4), we need to find quantitative counterparts of the **implication** $\to$, **universal quantifier** $\forall$, and **existential quantifier** $\exists$ used in their logical definitions. Generally, it is desirable to extend the parallel between Eqs. (1) and (2) to other predicates by finding quantitative operations corresponding to logical connectives and quantifiers. This correspondence allows us to construct and analyze quantitative metrics for machine learning models in a *compositional* manner [Boole, 1854].

## 2.2 Modularity: product structure preserved by a learning model

*Modularity* [Ridgeway and Mozer, 2018] is an essential property of disentangled representation learning, which means that the explanatory factors in data, such as the color and shape of an object, are separated into independent components in the learned representation [Bengio et al., 2013].

As shown in Fig. 1, modularity can be defined as follows. We assume that data with multiple explanatory factors (e.g., color and shape) is generated via a function $g : Y \to X$ from a product $Y := Y_1 \times Y_2$ of *factors*. An encoder $f : X \to Z$ is a function to a product $Z := Z_1 \times Z_2$ of *codes*. Then, an encoder is said to be *modular* if it can **reconstruct the product structure**, such that the composition $m := f \circ g : Y \to Z$ of the generator $g$ and the encoder $f$ is a product function.[4]

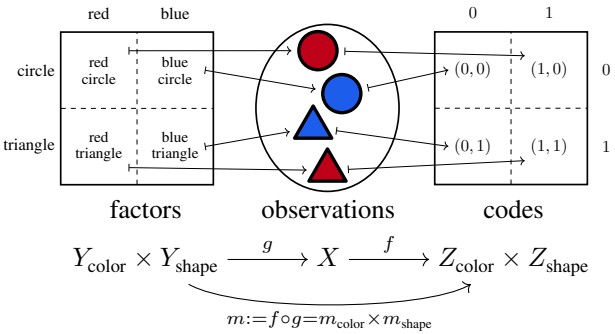

$$Y_{\text{color}} \times Y_{\text{shape}} \xrightarrow{\;g\;} X \xrightarrow{\;f\;} Z_{\text{color}} \times Z_{\text{shape}}$$
$$m := f \circ g = m_{\text{color}} \times m_{\text{shape}}$$

Figure 1: Disentangled representation learning

Formally, being a product function is also a property that can be represented as a predicate:

**Definition 3** (Product function). Let $Y := Y_1 \times Y_2$ and $Z := Z_1 \times Z_2$ be products of sets. A function $m : Y \to Z$ is a *product function* if
$$p_{\text{product}}(m : Y \to Z) := \exists m_{1,1} : Y_1 \to Z_1. \, \exists m_{2,2} : Y_2 \to Z_2. \, m =_{[Y,Z]} m_{1,1} \times m_{2,2}. \tag{5}$$

**Example 1.** Let us compare the following two functions from $Y := \{0, 1\}^2$ to $Z := \mathbb{R}^2$:

$$m := \begin{cases} (0,0) \mapsto (1,2) \\ (0,1) \mapsto (3,4) \\ (1,0) \mapsto (5,6) \\ (1,1) \mapsto (7,8) \end{cases} \quad (6) \qquad m' := \begin{cases} (0,0) \mapsto (a,c) \\ (0,1) \mapsto (a,d) \\ (1,0) \mapsto (b,c) \\ (1,1) \mapsto (b,d) \end{cases} = \underbrace{\begin{cases} 0 \mapsto a \\ 1 \mapsto b \end{cases}}_{m_{1,1}} \times \underbrace{\begin{cases} 0 \mapsto c \\ 1 \mapsto d \end{cases}}_{m_{2,2}} \quad (7)$$

where $a$, $b$, $c$, and $d$ are arbitrary real numbers. According to Definition 3, only $m' = m_{1,1} \times m_{2,2}$ is a product function, whose first/second output depends only on the first/second input.

In Example 1, although $m$ is not a product function, we want to address the following questions:

- (Metric) Can we quantify the extent to which it resembles a product function?
- (Approximation) Can we find a product function that is closest to it?
- (Differentiability) Can we make it slightly closer to a product function?

Answers to these questions will be given in the following sections.

---

[4]For two functions $f : A \to C$ and $g : B \to D$, their *product* $f \times g : A \times B \to C \times D$ applies these two functions "*in parallel*" by mapping a pair $(a, b)$ in $A \times B$ to a pair $(f(a), g(b))$ in $C \times D$.

# 3 Enrichment: from logic to metric

In Appendices A and B, we describe in detail the theory of converting a higher-order predicate into a real-valued quantity. In this section, we only introduce the conversion procedure using concrete examples and present the theoretical results. A summary of the conversion is given in Table 1.

First of all, let us clarify the terms predicate and quantity. In the realm of classical logic, a *predicate* $p : A \to \{\top, \bot\}$ on a set $A$ is a function from the set $A$ to the set $\{\top, \bot\}$ of binary truth values. For example, the predicates $p_{\text{injective}}$, $p_{\text{retractable}}$, and $p_{\text{product}}$ in Definitions 1 to 3 are functions from the set $[Y, Z]$ of functions to the set $\{\top, \bot\}$. They are *logical definitions* of some properties of functions. On the other hand, in this work, a *quantity* $q : A \to [0, \infty]$ on a set $A$ is defined as a function to the set $[0, \infty]$ of extended non-negative real numbers. The quantities associated with a predicate will serve as *quantitative metrics* for the property defined by the predicate.

Table 1: From logic to metric

| Logic | | Metric | |
|---|---|---|---|
| truth values | $\{\top, \bot\}$ | real values | $[0, \infty]$ |
| equality | $=$ | strict premetric | $d$ |
| conjunction | $\wedge$ | addition | $+$ |
| disjunction | $\vee$ | minimum | $\min$ |
| implication | $\to$ | subtraction* | $\dot{-}$ |
| universal quantifier | $\forall$ | aggregator** | $\triangledown$ |
| existential quantifier | $\exists$ | infimum | $\inf$ |

* truncated subtraction: $b \dot{-} a := \max\{b - a, 0\}$
** e.g., maximum, sum, mean, and mean square

## 3.1 From equality predicate to strict premetric

In this work, a predicate of central importance is the *equality predicate* $=_A: A \times A \to \{\top, \bot\}$ [Mazur, 2008]. A quantity associated with the equality predicate should be a strict premetric:

**Definition 4** (Strict premetric). *A strict premetric on a set $A$ is a function $d_A : A \times A \to [0, \infty]$ that*

$$\forall a \in A. \ \forall a' \in A. \ (d_A(a, a') = 0) \leftrightarrow (a =_A a'). \tag{8}$$

## 3.2 From logical operation to quantitative operation

Next, let us have a look at the logical connectives and quantifiers used in the definitions of properties. The product of sets and functions plays a significant role in this work. Two functions $f, g : C \to A \times B$ to a product are equal if and only if all their component functions are equal:

$$(f =_{[C, A \times B]} g) := (f_1 =_{[C, A]} g_1) \wedge (f_2 =_{[C, B]} g_2).^5 \tag{9}$$

Note that the **conjunction** $\wedge : \{\top, \bot\} \times \{\top, \bot\} \to \{\top, \bot\}$, a logical connective, is used in Eq. (9). To obtain a corresponding quantity, we replace it with the *addition* $+ : [0, \infty] \times [0, \infty] \to [0, \infty]$:

$$d_{[C, A \times B]}(f, g) := d_{[C, A]}(f_1, g_1) + d_{[C, B]}(f_2, g_2). \tag{10}$$

The **universal quantifier** on a set $A$ is a specific (second-order) predicate $\forall_A : \{\top, \bot\}^A \to \{\top, \bot\}$ on the set $\{\top, \bot\}^A$ of predicates. We can replace it with the *supremum* $\sup : [0, \infty]^A \to [0, \infty]$. We can also choose a function from the (i) *maximum*, (ii) *sum*, (iii) *mean*, and (iv) *mean square* when the set $A$ is finite. More generally, we can replace it with a quantity $\triangledown_A : [0, \infty]^A \to [0, \infty]$ on the set $[0, \infty]^A$ of quantities that satisfies some conditions, which we refer to as a (universal) *aggregator*. Intuitively, a universal aggregator should output 0 if and only if all inputs are 0. Therefore, the median, mode, and range are non-examples. Different choices of aggregators yield metrics with different characteristics in computation and optimization. For example, the function equality predicate

$$(f =_{[A, B]} g) := \forall a \in A. \ f(a) =_B g(a) \tag{11}$$

converts to a quantity whose aggregator $\triangledown$ is not limited to the sum (cf. Eqs. (1) and (2)):

$$d_{[A, B]}(f, g) := \underset{a \in A}{\triangledown} d_B(f(a), g(a)). \tag{12}$$

Dually, we also need to consider the **disjunction** $\vee$ and the **existential quantifier** $\exists$. We replace them with the *minimum* and the *infimum*, respectively. Lastly, we replace the **implication** $a \to b$ with the (truncated) *subtraction* $b \dot{-} a := \max\{b - a, 0\}$. These operations are illustrated in Fig. 2.

---

[5]For a function $f : C \to A \times B$ to a product $A \times B$, its *component functions* $f_1 : C \to A := p_1 \circ f$ and $f_2 : C \to A := p_2 \circ f$ are denoted by numeric subscripts, where $p_1 : A \times B \to A := (a, b) \mapsto a$ and $p_2 : A \times B \to B := (a, b) \mapsto b$ are *projection functions*.

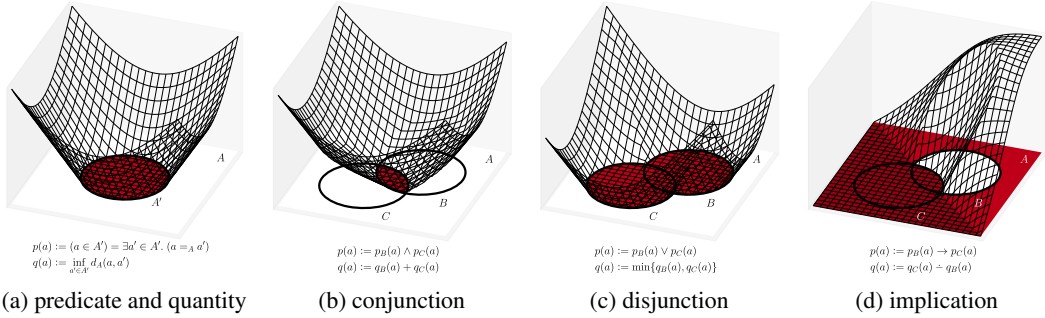

| (a) predicate and quantity | (b) conjunction | (c) disjunction | (d) implication |

Figure 2: From predicates and logical operations to quantities and quantitative operations

## 3.3 From compound predicate to compound quantity

Following Table 1, we can convert any *compound predicate* defined using equational predicates and logical operations into a corresponding *compound quantity* defined using strict premetrics and quantitative operations. Our main result on their relationship is as follows:

**Theorem 1.** *Let $p : A \to \{\top, \bot\}$ be a predicate on a set $A$, and let $q : A \to [0, \infty]$ be a quantity converted from $p$ according to Table 1. Then, for any $a \in A$, $q(a) = 0$ implies $p(a) = \top$. Conversely, for any $a \in A$, $p(a) = \top$ implies $q(a) = 0$ if and only if $p$ does not contain the implication.*

The implication is special because we must sacrifice logical equivalence for the sake of continuity, which is necessary for gradient-based optimization. We will explore this through a concrete example regarding injectivity in Section 4.3 and discuss it in detail in Appendices B and D.

## 3.4 (Sub)homomorphism from metric to logic

Finally, for readers interested in the theoretical background, we briefly introduce the following algebraic concepts and a proof sketch underlying Table 1 and Theorem 1.

**Definition 5** (Zero predicate). The *zero predicate* $\zeta : [0, \infty] \to \{\top, \bot\} := x \mapsto (x = 0)$ is a function that maps 0 to true $\top$ and any positive value to false $\bot$.

**Definition 6** ((Sub)homomorphism from a quantity to a predicate). Let $A$ be a set. A quantity $q : A \to [0, \infty]$ on the set $A$ is *homomorphic* to a predicate $p : A \to \{\top, \bot\}$ via the zero predicate $\zeta : [0, \infty] \to \{\top, \bot\}$ if $\zeta \circ q = p$, and is *subhomomorphic* to $p$ if $\zeta \circ q \to p$.[6]

**Definition 7** ((Sub)homomorphism from a quantitative operation to a logical operation). Let $n \in \mathbb{N}$ be a natural number. An $n$-ary quantitative operation $\alpha : [0, \infty]^n \to [0, \infty]$ is *homomorphic* to a logical operation $\beta : \{\top, \bot\}^n \to \{\top, \bot\}$ via the zero predicate $\zeta : [0, \infty] \to \{\top, \bot\}$ if $\zeta \circ \alpha = \beta \circ \zeta^n$, and is *subhomomorphic* to $\beta$ if $\zeta \circ \alpha \to \beta \circ \zeta^n$.[7]

**Definition 8** ((Sub)homomorphism from an aggregator to a quantifier). Let $A$ be a set. An aggregator $\alpha_A : [0, \infty]^A \to [0, \infty]$ on the set $A$ is *homomorphic* to a quantifier $\beta_A : \{\top, \bot\}^A \to \{\top, \bot\}$ via the zero predicate $\zeta : [0, \infty] \to \{\top, \bot\}$ if $\zeta \circ \alpha_A = \beta_A \circ \zeta^A$, and is *subhomomorphic* to $\beta_A$ if $\zeta \circ \alpha_A \to \beta_A \circ \zeta^A$.[8]

Homomorphic quantities, quantitative operations, and aggregators can be illustrated as follows:

$$
\begin{array}{ccc}
A \xrightarrow{\mathrm{id}_A} A & [0,\infty]^n \xrightarrow{\zeta^n} \{\top,\bot\}^n & [0,\infty]^A \xrightarrow{\zeta^A} \{\top,\bot\}^A \\
\downarrow{q} \quad \downarrow{p} & \downarrow{\alpha} \quad \downarrow{\beta} & \downarrow{\alpha_A} \quad \downarrow{\beta_A} \\
[0,\infty] \xrightarrow{\zeta} \{\top,\bot\} & [0,\infty] \xrightarrow{\zeta} \{\top,\bot\} & [0,\infty] \xrightarrow{\zeta} \{\top,\bot\}
\end{array}
\tag{13}
$$

---

[6]We use the infix notation, so $\zeta \circ q = p$ means that $\forall a \in A. (q(a) = 0) \leftrightarrow p(a)$, and $\zeta \circ q \to p$ means that $\forall a \in A. (q(a) = 0) \to p(a)$.

[7]$\zeta^n : [0, \infty]^n \to \{\top, \bot\}^n := (q_1, \ldots, q_n) \mapsto (q_1 = 0, \ldots, q_n = 0)$ is the $n$-fold *product* of the zero predicate $\zeta : [0, \infty] \to \{\top, \bot\}$.

[8]$\zeta^A : [0, \infty]^A \to \{\top, \bot\}^A := \zeta \circ (-)$ is the *postcomposition* with the zero predicate $\zeta : [0, \infty] \to \{\top, \bot\}$ that maps a quantity $q : A \to [0, \infty]$ to the predicate $\zeta \circ q : A \to \{\top, \bot\}$.

Based on these algebraic concepts, we can say that strict premetrics are homomorphic to equality predicates, addition is homomorphic to conjunction (since the sum is zero if and only if both addends are zero), minimum is homomorphic to disjunction, truncated subtraction is *subhomomorphic* to implication, and universal aggregators are homomorphic to the universal quantifier.

Theorem 1 means that any compound quantity is (sub)homomorphic to the corresponding compound predicate if each component (quantities, quantitative operations, aggregators) is (sub)homomorphic to the corresponding component (predicates, logical operations, quantifiers). For implication, we use the truncated subtraction, which is only subhomomorphic, since there is no *continuous* operation that is homomorphic to implication (see also Appendix D).

More abstractly and concisely, we can say that we replace the **Heyting algebra** of truth values $\{\top, \bot\}$ with a *quantale* of extended non-negative real numbers $[0, \infty]$, and we replace the quantifiers $\forall$ and $\exists$ with aggregators $\triangledown$ and $\inf$ (see also Appendix B). In this way, we can derive quantitative metrics for any logically defined properties of learning models *compositionally*.

## 4 Quantitative metrics of disentangled representations

In this section, we demonstrate how to apply the conversion method introduced above to derive quantitative metrics for measuring the modularity (Definition 3) and informativeness (Definitions 1 and 2) of disentangled representations.

In Sections 4.1 and 4.2, we introduce modularity metrics based on two approaches and discuss their differences in terms of computation and optimization. We point out that the main obstacle lies in the optimization step, resulting from the existential quantifiers in the definition. Then, we show that we can derive easily computable and differentiable metrics from a logically equivalent definition. In Section 4.3, we introduce informativeness metrics and present a result of Theorem 1.

### 4.1 Modularity metrics via product approximation

We begin with *modularity*, which is an essential property of disentangled representation learning. Recall that modularity can be defined using the *product function* (Definition 3). For easier reference, we provide the following diagram, which shows the domains and codomains of the functions involved in the upcoming discussion:

$$\begin{array}{ccccc}
Y_1 & \xleftarrow{\quad p_1 \quad} & \begin{array}{c} Y_1 \times Y_2 \\ (y_1, y_2) \end{array} & \xrightarrow{\quad p_2 \quad} & Y_2 \\
y_1 & & & & y_2 \\
\end{array}$$

(14)

From Definition 3, we can derive the following metric:

**Definition 9** (Product approximation). Let $m : Y \to Z$ be a function from a product $Y := Y_1 \times Y_2$ of sets to another product $Z := Z_1 \times Z_2$ of sets. The extent to which $m$ resembles a *product function* can be measured by a distance between $m$ and its best product function approximation:

$$q_{\text{product}}(m : Y \to Z) := \inf_{m_{1,1} \in [Y_1, Z_1]} \inf_{m_{2,2} \in [Y_2, Z_2]} d_{[Y,Z]}(m, m_{1,1} \times m_{2,2}). \tag{15}$$

The derivation of $q_{\text{product}}$ from $p_{\text{product}}$ follows the conversion described in Table 1: replacing the equality $=_{[Y,Z]} : [Y, Z] \times [Y, Z] \to \{\top, \bot\}$ with a strict premetric $d_{[Y,Z]} : [Y, Z] \times [Y, Z] \to [0, \infty]$ and the existential quantifiers $\exists$ with the infimum operators $\inf$.

This modularity metric can be interpreted as a distance from a point $m \in [Y, Z]$ to a subset $\{m_{1,1} \times m_{2,2} \mid m_{1,1} \in [Y_1, Z_1], m_{2,2} \in [Y_2, Z_2]\} \subset [Y, Z]$ of product functions (cf. the Hausdorff distance [Lawvere, 1986, Tuzhilin, 2016]). Following from Theorem 1, $q_{\text{product}}(m) = 0$ if and only if $p_{\text{product}}(m) = \top$. This means that the minimizers of this metric are precisely product functions.

However, we still face two obstacles: the product operation and the minimization problem. For the product operation, we can employ Eqs. (10) and (12) to rewrite $q_{\text{product}}$ into a more computable form:

**Proposition 2.** *The quantity $q_{\text{product}}(m : Y \to Z)$ equals*

$$\mathop{\nabla}_{y_1 \in Y_1} \mathop{\nabla}_{y_2 \in Y_2} d_{Z_1}(m_1(y_1, y_2), m_{1,1}^*(y_1)) + \mathop{\nabla}_{y_2 \in Y_2} \mathop{\nabla}_{y_1 \in Y_1} d_{Z_2}(m_2(y_1, y_2), m_{2,2}^*(y_2)), \qquad (16)$$

*where the functions $m_{1,1}^* : Y_1 \to Z_1$ and $m_{2,2}^* : Y_2 \to Z_2$ are given by*

$$m_{1,1}^* : Y_1 \to Z_1 := y_1 \mapsto \arg\inf_{z_1 \in Z_1} \mathop{\nabla}_{y_2 \in Y_2} d_{Z_1}(m_1(y_1, y_2), z_1), \qquad (17)$$

$$m_{2,2}^* : Y_2 \to Z_2 := y_2 \mapsto \arg\inf_{z_2 \in Z_2} \mathop{\nabla}_{y_1 \in Y_1} d_{Z_2}(m_2(y_1, y_2), z_2). \qquad (18)$$

A detailed derivation can be found in Appendix C. Note that we can obtain the optimal product function approximation $m_{1,1}^* \times m_{2,2}^*$ explicitly via Eqs. (17) and (18). Intuitively, we need to find an approximation of a (multi)set of codes with one factor fixed and other factors varying, and then we use the aggregation of all the approximation errors as a modularity metric.

The second obstacle — the minimization problem — still needs to be addressed. Since the code spaces $Z_1$ and $Z_2$ can be infinite sets, the minimization problem may not have a closed-form minimizer or even an exact solver. Even if an exact solver exists, the solution may not be differentiable with respect to the inputs. Let us examine some concrete examples of $q_{\text{product}}$ by choosing different aggregators $\nabla$ in Eqs. (17) and (18). In the following three examples, we assume that the code spaces $Z_1$ and $Z_2$ are Euclidean spaces equipped with the usual Euclidean distances.

**Example 2.** If the aggregator $\nabla$ is the *supremum*, the best approximation is the *center* of the smallest bounding sphere [Megiddo, 1983], and the approximation error is the *radius*.

This metric has the advantage of being definable even when the factor spaces $Y_1$ and $Y_2$ are infinite sets, and it can be computed using either randomized [Welzl, 1991] or exact [Fischer et al., 2003] algorithms. However, it is not easy to calculate its gradient. Thus, we cannot use it as a learning objective and directly optimize it using gradient-based optimization.

**Example 3.** If the aggregator $\nabla$ is the *mean*, the best approximation is the *(geometric) median* [Weiszfeld, 1937], and the approximation error is the *mean absolute deviation around the median*.

It is known that there is no exact algorithm for obtaining the geometric median [Cockayne and Melzak, 1969], but it can be effectively approximated using convex optimization [Cohen et al., 2016]. The geometric median has found applications in robust estimation in the fields of statistics and machine learning [Meer et al., 1991, Minsker, 2015, Pillutla et al., 2022, Guerraoui et al., 2023].

**Example 4.** If the aggregator $\nabla$ is the *mean square*, the best approximation is the *mean*, and the approximation error is the *variance*. In this case, $q_{\text{product}}(m)$ can be simplified to

$$\mathop{\text{mean}}_{y_1 \in Y_1} \mathop{\text{var}}_{y_2 \in Y_2} m_1(y_1, y_2) + \mathop{\text{mean}}_{y_2 \in Y_2} \mathop{\text{var}}_{y_1 \in Y_1} m_2(y_1, y_2). \qquad (19)$$

The variance is easier to compute and differentiate than the radius of the smallest bounding sphere and the mean absolute deviation around the median, but it is also more susceptible to outliers and noise. Further work could explore the theoretical implications of these metrics, especially in cases where only partial combinations of factors or noisy annotations are available.

Then, let us revisit our motivating example in Example 1:

**Example 5.** Let us consider the function $m : \{0, 1\}^2 \to \mathbb{R}^2$ in Eq. (6). Its best product function approximation is

$$m^* : \{0, 1\}^2 \to \mathbb{R}^2 := \begin{cases} (0,0) \mapsto (2,4) \\ (0,1) \mapsto (2,6) \\ (1,0) \mapsto (6,4) \\ (1,1) \mapsto (6,6) \end{cases} = \underbrace{\begin{cases} 0 \mapsto 2 \\ 1 \mapsto 6 \end{cases}}_{m_{1,1}^*} \times \underbrace{\begin{cases} 0 \mapsto 4 \\ 1 \mapsto 6 \end{cases}}_{m_{2,2}^*} \qquad (20)$$

because $m_{1,1}^*(0) = 2$ is the center/median/mean of the set $\{m_1(0,0) = 1, m_1(0,1) = 3\}$, and so on. The modularity metric is a distance between $m$ and $m^*$.

## 4.2 Modularity metrics via constancy

Upon analyzing the metrics above, it becomes evident that what we need is not the best approximation itself (e.g., the mean) but rather the approximation error (e.g., the variance) — a measure of the *constancy* of a set of codes. Following this insight, our next objective is to formulate a modularity metric that eliminates the need for an optimization step. Zhang and Sugiyama [2023] have proved that a function is a product function if and only if the *curried functions*[9] of its component functions are constant, as shown in the following example:

**Example 6.** Consider the functions $m, m' : \{0,1\}^2 \to \mathbb{R}^2$ in Eqs. (6) and (7) and the curried functions of their second component functions $m_2, m_2' : \{0,1\}^2 \to \mathbb{R}$:

$$
m_2 = \begin{cases} (0,0) \mapsto 2 \\ (0,1) \mapsto 4 \\ (1,0) \mapsto 6 \\ (1,1) \mapsto 8 \end{cases} \cong \begin{cases} 0 \mapsto \begin{cases} 0 \mapsto 2 \\ 1 \mapsto 4 \end{cases} \\ 1 \mapsto \begin{cases} 0 \mapsto 6 \\ 1 \mapsto 8 \end{cases} \end{cases} \quad (21) \qquad m_2' = \begin{cases} (0,0) \mapsto c \\ (0,1) \mapsto d \\ (1,0) \mapsto c \\ (1,1) \mapsto d \end{cases} \cong \begin{cases} 0 \mapsto \begin{cases} 0 \mapsto c \\ 1 \mapsto d \end{cases} \\ 1 \mapsto \begin{cases} 0 \mapsto c \\ 1 \mapsto d \end{cases} \end{cases} \quad (22)
$$

The curried function $\widehat{m_2} : \{0,1\} \to [\{0,1\}, \mathbb{R}]$ is not constant, while $\widehat{m_2'}$ is constant with value $\{0 \mapsto c, 1 \mapsto d\} \in [\{0,1\}, \mathbb{R}]$ (and so is $\widehat{m_1'}$), indicating that $m'$ is a product function.

Based on this fact, we propose an alternative approach for measuring modularity:

**Definition 10** (Constancy of curried function). Let $m_1$ and $m_2$ be the component functions of a function $m : Y \to Z$ from a product $Y := Y_1 \times Y_2$ of sets to another product $Z := Z_1 \times Z_2$ of sets. The extent to which $m$ resembles a product function can be measured by the *constancy* of the curried functions of $m_1$ and $m_2$:

$$
q_{\text{const-curry}}(m : Y \to Z) := q_{\text{const}}(\widehat{m_1}) + q_{\text{const}}(\widehat{m_2}), \tag{23}
$$

where $q_{\text{const}}$ is a quantity for constant functions.

To complete this construction, we adopt the following definition and metric of the constant function:

**Definition 11** (Constant function). A function $f : A \to B$ is *constant* if

$$
p_{\text{const}}(f : A \to B) := \forall a \in A. \ \forall a' \in A. \ f(a) =_B f(a'), \tag{24}
$$

which can be measured by

$$
q_{\text{const}}(f : A \to B) := \bigtriangledown_{a \in A} \bigtriangledown_{a' \in A} d_B(f(a), f(a')). \tag{25}
$$

This constancy metric $q_{\text{const}}$ only needs to compute pairwise distances between the outputs, requiring $|A|^2$ times distance computation but no optimization. Incorporating $q_{\text{const}}$ into $q_{\text{const-curry}}$, we can get the following metric:

**Proposition 3.** *The quantity* $q_{\text{const-curry}}(m : Y \to Z)$ *equals*

$$
\begin{aligned}
& \bigtriangledown_{y_1 \in Y_1} \bigtriangledown_{y_2 \in Y_2} \bigtriangledown_{y_2' \in Y_2} d_{Z_1}(m_1(y_1, y_2), m_1(y_1, y_2')) \\
& + \bigtriangledown_{y_2 \in Y_2} \bigtriangledown_{y_1 \in Y_1} \bigtriangledown_{y_1' \in Y_1} d_{Z_2}(m_2(y_1, y_2), m_2(y_1', y_2)).
\end{aligned} \tag{26}
$$

Here are two examples of $q_{\text{const}}$ and $q_{\text{const-curry}}$ using different aggregators $\bigtriangledown$ in Eqs. (25) and (26).

**Example 7.** If the aggregator $\bigtriangledown$ is the *maximum*, $q_{\text{const}}$ is the *diameter* (the maximum pairwise distance) of the outputs. In this case, $q_{\text{const-curry}}(m)$ can be simplified to

$$
\max_{y_1 \in Y_1} \operatorname{diam}_{y_2 \in Y_2} m_1(y_1, y_2) + \max_{y_2 \in Y_2} \operatorname{diam}_{y_1 \in Y_1} m_2(y_1, y_2). \tag{27}
$$

**Example 8.** If the aggregator $\bigtriangledown$ is the *mean square*, $q_{\text{const}}$ is the mean pairwise squared distance, which equals the *variance*. In this case, $q_{\text{const-curry}}(m)$ coincides with Eq. (19).

In summary, Eqs. (19) and (27) are easily computable and differentiable metrics, and their minimizers are precisely product functions. They do not contain any hyperparameters or stochastic components and thus can serve as both learning objectives and evaluation metrics.

---

[9]For a binary function $f : A \times B \to C$, its *curried function* $\widehat{f} : A \to [B, C]$ is a unary function such that for all $a \in A$ and $b \in B$, $f(a, b) = \widehat{f}(a)(b)$ [Curry, 1980].

## 4.3 Informativeness metrics

If an encoder $f : X \to Z$ is constant, mapping everything to the same value, according to Definition 3, it is perfectly modular. However, a constant encoder is also completely useless. In this subsection, we shift our focus to the property of *informativeness* — a measurement of usefulness.

Informativeness is not a unique requirement for disentangled representations. Other representation learning paradigms, such as contrastive learning [Jaiswal et al., 2020, Wang and Isola, 2020] and metric learning [Musgrave et al., 2020], also emphasize the importance of mapping dissimilar data to far-apart locations in the representation space. While one could integrate this requirement into a single disentanglement score (e.g., [Higgins et al., 2017, Kim and Mnih, 2018]), we argue that it is better to evaluate the usefulness of representations separately for a more fine-grained assessment [Carbonneau et al., 2022].

One straightforward way to measure informativeness is to measure how much we can invert the encoding process:

**Definition 12** (Retraction approximation). Let $m : Y \to Z$ be a function. The extent to which $m$ is *retractable* can be measured by a distance between the composition of $m$ and its best retraction approximation and the identity function:

$$q_{\text{retractable}}(m : Y \to Z) := \inf_{h \in [Z,Y]} d_{[Y,Y]}(h \circ m, \text{id}_Y) = \inf_{h \in [Z,Y]} \bigvee_{y \in Y} d_Y(h(m(y)), y). \quad (28)$$

This metric $q_{\text{retractable}}$ is derived from Definition 2 following the conversion procedure in Table 1. This informativeness metric also involves an optimization step similar to the modularity metric $q_{\text{product}}$, potentially introducing randomness or higher computation costs. Note that we may use a parameterized subset of the set $[Z, Y]$ of all functions from codes $Z$ to factors $Y$, such as the set of linear functions. Then, the problem becomes a regression/classification problem, and the metric is the performance of the predictor. A number of existing works adopted this approach and used the accuracy, the area under the ROC curve (AUC-ROC), or the mean squared error (MSE) to measure the informativeness [Ridgeway and Mozer, 2018, Eastwood and Williams, 2018, Eastwood et al., 2023]. However, such metrics necessitate additional hyperparameter tuning and are more likely to exhibit varying behavior across different implementations [Carbonneau et al., 2022].

It raises the question of whether we can measure the informativeness of an encoder without approximating its retraction. We propose to measure informativeness by directly measuring the injectivity of the encoding process:

**Definition 13** (Contraction). Let $m : Y \to Z$ be a function. The extent to which $m$ is *injective* can be measured by how much $m$ contracts pairs of inputs:

$$q_{\text{injective}}(m : Y \to Z) := \bigvee_{y \in Y} \bigvee_{y' \in Y} d_Y(y, y') \dotminus d_Z(m(y), m(y')). \quad (29)$$

This metric $q_{\text{injective}}$ is derived from Definition 1 following the conversion procedure in Table 1. According to Theorem 1, we know that $q_{\text{retractable}}(m) = 0$ if and only if $m$ is retractable. However, $q_{\text{injective}}(m) = 0$ implies the injectivity of $m$ but not the other way around:

$$(p_{\text{retractable}}(m) = \top) \xleftrightarrow[\text{equivalent}]{\text{logically}} (p_{\text{injective}}(m) = \top)$$

$$\text{Theorem 1} \Updownarrow \qquad\qquad\qquad \Updownarrow \text{Theorem 1} \qquad (30)$$

$$(q_{\text{retractable}}(m) = 0) \qquad\qquad (q_{\text{injective}}(m) = 0)$$

In other words, a minimizer of $q_{\text{injective}}$ is required to be *non-contractive*, which is a stronger condition than being injective. For example, let us consider the function $m : [0, 1] \to \mathbb{R} := y \mapsto 0.01 \times y$. Although it is injective, its outputs are less distinguishable from each other in terms of the Euclidean distance. Therefore, $q_{\text{injective}}$ still assign a non-zero value to this function.

Although not all injective functions necessarily minimize $q_{\text{injective}}$, according to Theorem 1, we can guarantee that minimizing $q_{\text{injective}}$ will not lead to non-injective functions. Moreover, $q_{\text{injective}}$ does not require training regressors or classifiers to approximate the retraction. Consequently, it does not need any time-consuming hyperparameter tuning or cross-validation like existing informativeness metrics [Eastwood and Williams, 2018, Ridgeway and Mozer, 2018].

Table 2: Supervised disentanglement metrics

| | | Modularity | | | | | | Informativeness | | | | | | Existing metrics | | | | |
| | | Product approx. | | | Constancy | | | Retraction approx. | | | Contraction | | Pair | | Info. | Regressor | | |
| | | Rad. | MAD | Var. | Diam. | MPD | | ME | MAE | MSE | Max | Mean | Beta[a] | Factor[b] | MIG[c] | Dis.[d] | Com.[d] | Info.[d] |
|---|---|---|---|---|---|---|---|---|---|---|---|---|---|---|---|---|---|---|
| entanglement | ✗ | 0.44 | 0.75 | 0.96 | 0.19 | 0.82 | ✓ | 0.76 | 0.96 | 0.99 | 0.44 | 0.78 | 0.89 | 0.83 | 0.18 | 0.28 | 0.28 | 1.00 |
| rotation | ✗ | 0.22 | 0.51 | 0.80 | 0.05 | 0.64 | ✓ | 1.00 | 1.00 | 1.00 | 1.00 | 1.00 | 0.96 | 0.34 | 0.17 | 0.40 | 0.40 | 1.00 |
| duplicate | ✗ | 0.24 | 0.43 | 0.67 | 0.06 | 0.56 | ✓ | 1.00 | 1.00 | 1.00 | 1.00 | 1.00 | 1.00 | 1.00 | 1.00 | 1.00 | 0.59 | 1.00 |
| complement | ✗ | 0.12 | 0.28 | 0.55 | 0.01 | 0.42 | ✓ | 1.00 | 1.00 | 1.00 | 1.00 | 1.00 | 1.00 | 0.00 | 1.00 | 1.00 | 0.63 | 1.00 |
| misalignment | ✗ | 0.22 | 0.44 | 0.74 | 0.05 | 0.58 | ✓ | 1.00 | 1.00 | 1.00 | 1.00 | 1.00 | 1.00 | 0.00 | 1.00 | 1.00 | 1.00 | 1.00 |
| redundancy | ✓ | 1.00 | 1.00 | 1.00 | 1.00 | 1.00 | ✓ | 1.00 | 1.00 | 1.00 | 1.00 | 1.00 | 1.00 | 0.33 | 1.00 | 1.00 | 0.93 | 1.00 |
| contraction | ✓ | 1.00 | 1.00 | 1.00 | 1.00 | 1.00 | ✓ | 1.00 | 1.00 | 1.00 | 0.18 | 0.49 | 1.00 | 1.00 | 1.00 | 1.00 | 1.00 | 1.00 |
| nonlinear | ✓ | 1.00 | 1.00 | 1.00 | 1.00 | 1.00 | ✓ | 0.79 | 0.93 | 0.99 | 0.65 | 0.95 | 1.00 | 1.00 | 0.88 | 1.00 | 1.00 | 1.00 |
| constant | ✓ | 1.00 | 1.00 | 1.00 | 1.00 | 1.00 | ✗ | 0.42 | 0.76 | 0.90 | 0.18 | 0.48 | 0.33 | 0.33 | 0.00 | 0.00 | 0.00 | 0.00 |
| random | ✗ | 0.22 | 0.48 | 0.78 | 0.05 | 0.61 | ✗ | 0.42 | 0.76 | 0.90 | 0.22 | 0.83 | 0.34 | 0.33 | 0.00 | 0.00 | 0.00 | 0.04 |

[a] [Higgins et al., 2017] [b] [Kim and Mnih, 2018] [c] [Chen et al., 2018] [d] [Eastwood and Williams, 2018]

# 5 Experiments

In this section, we empirically demonstrate the effectiveness of the proposed metrics. Following Carbonneau et al. [2022], we did not learn representations on datasets but directly defined functions $m : Y \to Z$ from factors to codes, which allows us to capture typical failure patterns.

We evaluated modularity metrics based on (i) the radius of the smallest bounding sphere, (ii) mean absolute deviation (MAD) around the median, (iii) variance, (iv) diameter, and (v) mean pairwise distance (MPD) introduced in Sections 4.1 and 4.2. We evaluated informativeness metrics based on retraction approximation using the maximum error (ME), mean absolute error (MAE), and mean squared error (MSE), and we calculated the contraction discussed in Section 4.3. To compare with existing metrics [Higgins et al., 2017, Kim and Mnih, 2018, Chen et al., 2018, Eastwood and Williams, 2018], we transformed the results isomorphically using $e^{-x} : [0, \infty] \to [0, 1]$, meaning that 1 is the perfect score. The results are shown in Table 2, and our observations are as follows.

**If a representation is given a perfect score by a proposed metric, it must satisfy the property that the metric quantifies**, which confirms our theoretical result. In Table 2, the light cells show that the proposed metrics can assign a perfect score when the function truly satisfies the properties, which is indicated by ✓ and ✗. The dark cells are supposed to be perfect scores, but they fall short due to the limited expressiveness of the linear models used for the approximation. Meanwhile, some existing metrics that only provide a single score may entangle modularity and informativeness.

**The metrics derived from equivalent definitions may differ in terms of computation cost and differentiability.** Concretely, the radius, MAD, ME, MAE, and MSE are not differentiable due to the inner optimization problem, while the variance, diameter, MPD, and max/mean contraction are differentiable. In terms of computation, they are much faster than metrics requiring hyperparameter tuning, such as DCI [Eastwood and Williams, 2018]. Further comparisons are provided in Appendix E.

**Different metrics may rank imperfect representations differently, even though they have exactly the same minimizers.** This difference can lead to differences in risk preferences, sensitivity to outliers, and learning dynamics when these metrics are used as learning objectives. Illustrations and further discussion can be found in Appendix D.

Further, the proposed metrics can be used in *weakly supervised* or *fine-grained* evaluation. See Appendix E for detailed data configuration and further experimental results.

# 6 Conclusion

In this work, we developed a systematic and rigorous method for converting logical definitions of properties of representation learning models into quantitative metrics (Table 1). We applied this method to assess two important and distinct properties of disentangled representations: modularity and informativeness. We derived two families of metrics for each property based on their logically equivalent definitions. We theoretically analyzed the minimizers of these metrics (Theorem 1) and compared their differences in terms of computation cost and differentiability. Future research could compare metrics derived from different aggregators and design appropriate models to optimize these metrics with minimal supervision.

# Acknowledgments

We thank Paolo Perrone for generously sharing insights on Markov categories enriched in divergence spaces. We thank Ken Sakayori for reviewing an earlier version of Appendices A and B and providing constructive suggestions. We thank Zhiyuan Zhan for insightful discussions on metrics, topology, measures, and optimization of function properties, and for reviewing and proofreading some parts of Appendix D. We thank Masahiro Negishi for valuable discussions on disentanglement metrics and weakly supervised disentanglement. We thank Jingwen Fu for checking the algebraic concepts in Section 3. We also thank Johannes Ackermann, Xin-Qiang Cai, and Tongtong Fang for their valuable feedback on the manuscript.

YZ was supported by JSPS KAKENHI Grant Number 22KJ0880. MS was supported by JST CREST Grant Number JPMJCR18A2 and a grant from Apple, Inc. Any views, opinions, findings, and conclusions or recommendations expressed in this material are those of the authors and should not be interpreted as reflecting the views, policies or position, either expressed or implied, of Apple Inc.

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

# Contents

## List of Figures

## List of Tables

# A  Preliminaries

In this paper, we used abstract mathematical tools such as category theory and topos theory to develop a theory of the relationship between logical definitions and quantitative metrics. However, this level of abstraction may be unfamiliar or even intimidating to some readers, and sometimes unnecessary for machine learning practitioners. Therefore, we have used only the most basic algebraic concepts, such as homomorphism, in the main text. For readers interested in the mathematical background, we provide a brief introduction to the basic categorical concepts in this section.

## A.1  Basic category theory

*Category theory* is a branch of mathematics that studies mathematical structures in an abstract way, which is suitable for identifying and organizing common patterns across various fields of mathematics [Mac Lane, 1978, Adámek et al., 1990, Awodey, 2010]. It has found applications in many fields, including computer science [Barr and Wells, 1990], probability theory [Cho and Jacobs, 2019, Fritz, 2020, Perrone, 2023], and machine learning [de Haan et al., 2020, Shiebler et al., 2021, Cruttwell et al., 2022, Dudzik and Veličković, 2022, Shiebler, 2023, Yuan, 2023, Pearce-Crump, 2023, Chen et al., 2024, Gavranović et al., 2024].

The most fundamental concept is that of a *category*:

**Definition 14.** A *category* $\mathbf{C} = (\mathrm{Obj}, \mathrm{Hom}, \circ, \mathrm{id})$ consists of

- a collection $\mathrm{Obj}$ of objects,
- a set $\mathrm{Hom}(A, B)$ of morphisms between objects,[10]
- a composition function $\circ : \mathrm{Hom}(B, C) \times \mathrm{Hom}(A, B) \to \mathrm{Hom}(A, C)$ for each triple of objects, and
- an identity morphism $\mathrm{id}_A \in \mathrm{Hom}(A, A)$ for each object,

subject to

- associativity: $(h \circ g) \circ f = h \circ (g \circ f)$ and
- identity: $\mathrm{id}_B \circ f = f = f \circ \mathrm{id}_A$.

A crucial example is the category $\mathbf{Set}$ of sets and functions. However, what is of particular interest is not the category itself but its relationships with other categories. Building on the concepts of the *functor* and *natural transformation*, whose definitions are omitted here, we can develop tools to better understand the properties of a category.

Moreover, we can *define* objects in terms of their relations with other objects, employing what is known as *universal construction*. For example, a *terminal object* 1 in a category is an object such that for any object $A$, there exists a unique morphism $e_A : A \to 1$ to it, which we call a *terminal morphism*. In $\mathbf{Set}$, any set $\{*\}$ with only one element is a terminal object. Based on the concept of the terminal object, a *global element* of an object $B$ is defined to be a morphism $b : 1 \to B$ from a terminal object. We write $b_A : A \to B$ as an abbreviation for the *constant morphism* $b \circ e_A : A \xrightarrow{e_A} 1 \xrightarrow{b} B$ with value $b : 1 \to B$. These concepts will be used to develop our theory in Appendix B. Other important universal constructions include the *product*, *pullback*, and *exponential*. The concept of the product is of great importance in disentangled representation learning [Zhang and Sugiyama, 2023].

Regarding the pullback, we need to mention the following useful lemma:

**Lemma 4** (Pullback lemma). *Suppose that in the following commutative diagram, the right square is a pullback.*

$$
\begin{array}{ccccc}
\cdot & \longrightarrow & \cdot & \longrightarrow & \cdot \\
\downarrow & & \downarrow & & \downarrow \\
\cdot & \longrightarrow & \cdot & \longrightarrow & \cdot
\end{array}
\tag{31}
$$

*Then, the left square is a pullback if and only if the outer rectangle is a pullback.*

This lemma is usually left as an exercise in textbooks [e.g., Mac Lane, 1978, p. 72, Exercise 8, Leinster, 2014, Exercise 5.1.35]. A proof can be found in Fong and Spivak [2019, Proposition 7.3]. We need to use this lemma to (de)compose pullbacks. As a side note, we use the asterisk $f^*g$ to denote the pullback of $g$ along $f$.

---

[10]To be more precise, a category whose morphisms are sets is called a *locally small* category.

## A.2 Elementary topos theory

*Topos theory* studies categories that, in some sense, exhibit behavior akin to the category of sets and functions [Lawvere and Rosebrugh, 2003]. Topos theory has found applications in geometry, topology, and logic [Mac Lane and Moerdijk, 1994, Johnstone, 2002, Leinster, 2010, Trimble, 2019]. In this work, we only explore its relation to logic.

To formally define a *topos*, two essential concepts are those of the subobject and subobject classifier. A *subobject* of an object $C$ is simply a monomorphism $b : B \rightarrowtail C$ to the object $C$. The subobject classifier is defined as follows:

**Definition 15.** In a finitely complete category, a *subobject classifier* is a universal subobject $\top : 1 \rightarrowtail \Omega$ such that for every subobject $b : B \rightarrowtail C$, there exists a unique morphism $\chi_b : C \to \Omega$ such that $b$ is a pullback of $\top$ along $\chi_b$. The morphism $\chi_b$ is called the *classifying morphism* of $b$.

$$
\begin{array}{ccc}
B & \dashrightarrow & 1 \\
b \downarrow & \lrcorner & \downarrow \top \\
C & \dashrightarrow[\chi_b] & \Omega
\end{array}
\tag{32}
$$

Alternatively, we can state that

**Proposition 5.** *A subobject classifier is precisely a terminal object in the category of monomorphisms and pullbacks.*

Then, we can study the morphisms to the object $\Omega$:

**Definition 16.** In a category with a subobject classifier $\top : 1 \rightarrowtail \Omega$, a *predicate* on an object $C$ is a morphism $p : C \to \Omega$.

Based on Definition 15, we can state that subobjects of an object $C$ are classified by predicates on $C$.

For example, in **Set**, subobjects are subsets, a function from a singleton $\{*\}$ to a two-element set is a subobject classifier, which is usually denoted by $\top : \{*\} \to \{\top, \bot\}$, a predicate on a set $C$ is a function $p : C \to \{\top, \bot\}$, and a subset precisely corresponds to its indicator function.

Among various equivalent definitions of a topos, a concise one is as follows:

**Definition 17.** An *elementary topos* is a finitely complete and cartesian closed category with a subobject classifier.

Despite its concise definition, a great number of logical structures can be derived from it, which will be explored in Appendix B.

## A.3 Enriched category theory

*Enriched category theory* generalizes the concept of the category by replacing the sets of morphisms with objects in a suitable category [Kelly, 1982]. It has been used to better understand a wide range of domains, from metric spaces [Lawvere, 1973] to language [Bradley et al., 2022].

Let us dive into the definition of an enriched category:

**Definition 18.** A category $\mathbf{C} = (\mathrm{Obj}, \mathrm{Hom}, \circ, \mathrm{id})$ enriched in a monoidal category $(\mathbf{V}, \otimes, I)$ consists of

- a collection $\mathrm{Obj}$ of objects,
- a hom-object $\mathrm{Hom}(A, B) \in \mathrm{Obj}_{\mathbf{V}}$ between objects,
- a composition morphism $\circ : \mathrm{Hom}(B, C) \otimes \mathrm{Hom}(A, B) \to \mathrm{Hom}(A, C)$ for each triple of objects, and
- an identity element $\mathrm{id}_A : I \to \mathrm{Hom}(A, A)$ for each object,

subject to associativity and identity.

Comparing Definitions 14 and 18, we can say that a (locally small) category is a category enriched in the category **Set** of sets and functions. Enrichment is a way to describe the additional structures of morphisms and the properties that need to be respected by composition.

An example is a preorder, which can be seen as a category enriched in the category of boolean values. Another example is a *Lawvere metric space* [Lawvere, 1973], which is a set with a premetric that satisfies the triangle inequality. Note that the transitivity of a preorder and the triangle inequality of a Lawvere metric are described by their composition morphisms, respectively.

In this work, we use enrichment to describe the association of a set of morphisms with additional operations, such as a strict premetric and aggregators.

## B Theory

In this section, we detail the theory of converting logical definitions into their corresponding quantitative metrics based on elementary topos theory and enriched category theory.

### B.1 Subobject quantifier and quantizer

Since our main goal is to develop a multi-valued (possibly continuous and differentiable) quantification of properties defined by a certain type of logic, we begin with a category $\mathbf{E}$ that has sufficient structures to allow the desired logical operations and build the quantification upon these structures.

Firstly, recall that a subobject classifier $\Omega$, if exists, is the *representing object* of the subobject functor $\mathrm{Sub}_{\mathbf{E}}$, such that a subobject $b : B \rightarrowtail C$ corresponds to a unique classifying morphism $\chi_b : C \to \Omega$, and an *external operation* on the set $\mathrm{Sub}_{\mathbf{E}}(C)$ of subobjects that is natural in the object $C$ corresponds to an *internal operation*.

For example, the intersection

$$\cap_C : \mathrm{Sub}_{\mathbf{E}}(C) \times \mathrm{Sub}_{\mathbf{E}}(C) \to \mathrm{Sub}_{\mathbf{E}}(C) \tag{33}$$

corresponds a natural transformation

$$\mathrm{Hom}_{\mathbf{E}}(-, \Omega) \times \mathrm{Hom}_{\mathbf{E}}(-, \Omega) \Rightarrow \mathrm{Hom}_{\mathbf{E}}(-, \Omega), \tag{34}$$

which, because the hom-functor $\mathrm{Hom}_{\mathbf{E}}$ preserves limits, is isomorphic to a natural transformation between hom-functors

$$\mathrm{Hom}_{\mathbf{E}}(-, \Omega \times \Omega) \Rightarrow \mathrm{Hom}_{\mathbf{E}}(-, \Omega), \tag{35}$$

which, by the Yoneda lemma, is isomorphic to an internal operation on the subobject classifier $\Omega$:

$$\wedge : \Omega \times \Omega \to \Omega. \tag{36}$$

Note that the subobject classifier, the classifying morphisms, and those internal operations are determined uniquely up to isomorphism. However, in order to obtain a multi-valued quantification, the requirement for uniqueness might be too restrictive. Thus, we propose to study a weakened concept instead:

**Definition 19.** In a finitely complete category, a *subobject quantifier* is a subobject $o : 1 \rightarrowtail \Psi$ such that for every subobject $b : B \rightarrowtail C$, there exists at least one morphism $\phi_b : C \to \Psi$ such that $b$ is a pullback of $o$ along $\phi_b$. The morphism $\phi_b$ is called a *quantifying morphism* of $b$. If the category has a subobject classifier $\top : 1 \rightarrowtail \Omega$, the *quantizer* $\kappa : \Psi \to \Omega$ of the subobject quantifier $o$ is the classifying morphism of $o$.

$$\tag{37}$$

More succinctly, we can state that (cf. Proposition 5)

**Proposition 6.** *A subobject quantifier is a weakly terminal object in the category of monomorphisms and pullbacks.*

Thus, there is a unique morphism $\chi_b : C \to \Omega$ classifying a subobject $b : B \rightarrowtail C$ of an object $C$, but there could be multiple morphisms $\phi_b : C \to \Psi$ quantifying this subobject.

It is provable that the domain of a terminal object $\top$ in the category of monomorphisms and pullbacks (Proposition 5) must be a terminal object $1$ in the category $\mathbf{E}$, but not all weakly terminal objects in the category of monomorphisms and pullbacks (Proposition 6) are monomorphisms out of a terminal object. We choose Definition 19 because we want only one global element $o : 1 \rightarrowtail \Psi$ to be designated to the truth value $\top : 1 \rightarrowtail \Omega$.

Here, we give two examples to motivate this definition of subobject quantifier. One is related to *three-valued logic* [Bergmann, 2008, Fong and Spivak, 2019, Exercise 2.34]:

**Example 9.** In $\mathbf{Set}$, the function

$$\mathtt{yes} : \{*\} \to \{\mathtt{no}, \mathtt{maybe}, \mathtt{yes}\}, \tag{38}$$

which maps the element $*$ in a singleton set $\{*\}$ (a terminal object in $\mathbf{Set}$) to an element $\mathtt{yes}$ in a three-element set $\{\mathtt{no}, \mathtt{maybe}, \mathtt{yes}\}$ is a subobject quantifier. For any subset $B$ of a set $C$, a quantifying morphism $\phi_b : C \to \Psi$ is a function that maps all elements in the subset $B$ to $\mathtt{yes}$ and all other elements to either $\mathtt{maybe}$ or $\mathtt{no}$.

The other is related to *metric spaces* [Lawvere, 1973] and will be our running example in the following subsections:

**Example 10.** In $\mathbf{Set}$, the function $0 : \{*\} \to [0, \infty]$ selecting the number $0$ out of the set $[0, \infty]$ of extended non-negative real numbers is a subobject quantifier. The quantizer is a function

$$\kappa : [0, \infty] \to \{\top, \bot\} := n \mapsto \begin{cases} \top & n = 0, \\ \bot & n > 0, \end{cases} \tag{39}$$

which maps $0$ to $\top$ and any non-zero number to $\bot$.

Intuitively, with a subobject quantifier, there is only one way to be true, but there may be many ways to be false. In $\mathbf{Set}$, a quantizer $\kappa$ maps multiple "*degrees of truth*" from a potentially large, even infinite set $\Psi$ to a smaller set $\Omega$ of truth values, hence the name.

Next, we define the counterpart of the concept of predicate (Definition 16):

**Definition 20.** In a category with a subobject quantifier $o : 1 \rightarrowtail \Psi$, a *quantity* on an object $C$ is a morphism $q : C \to \Psi$.

Since we weakened the requirement for uniqueness, there is no one-to-one correspondence between subobjects and quantities. However, they are still related as follows:

**Lemma 7.** *In a category with a subobject classifier $\top : 1 \rightarrowtail \Omega$, a subobject quantifier $o : 1 \rightarrowtail \Psi$, and a quantizer $\kappa : \Psi \to \Omega$, a quantity $q : C \to \Psi$ on an object $C$ is a quantifying morphism of a subobject $q^* o$ of the object $C$, which is isomorphic to a subobject $(\kappa \circ q)^* \top$.*

*Proof.* $q^* o$ and $(\kappa \circ q)^* \top$ are both subobjects of $C$ because pullbacks preserve subobjects. Their isomorphism follows from the pullback lemma. $\square$

**Lemma 8.** *In a category with a subobject classifier $\top : 1 \rightarrowtail \Omega$, a subobject quantifier $o : 1 \rightarrowtail \Psi$, and a quantizer $\kappa : \Psi \to \Omega$, a quantity $q : C \to \Psi$ on an object $C$ is a quantifying morphism of a subobject $b : B \rightarrowtail C$ if and only if $\kappa \circ q = \chi_b$, where $\chi_b$ is the classifying morphism of the subobject $b$.*

*Proof.* Necessity follows from the pullback lemma and the uniqueness of the classifying morphism; sufficiency follows from Lemma 7. $\square$

The following relationship between a quantity and the quantizer is also useful:

**Lemma 9.** *In a category with a subobject classifier $\top : 1 \rightarrowtail \Omega$, a subobject quantifier $o : 1 \rightarrowtail \Psi$, and a quantizer $\kappa : \Psi \to \Omega$, for any quantity $q : C \to \Psi$ on an object $C$, $q = o_C$ if and only if $\kappa \circ q = \kappa \circ o_C = \top_C$, where $o_C$ is the constant morphism $o \circ e_C : C \xrightarrow{e_C} 1 \xrightarrow{o} \Psi$ with value $o : 1 \to \Psi$.*

*Proof.* This is due to the universal property of pullback $o$ of $\top$ along $\kappa$. $\square$

We can see that the hom-functor on the quantizer $\mathrm{Hom}_{\mathbf{E}}(-, \kappa) : \mathrm{Hom}_{\mathbf{E}}(-, \Psi) \Rightarrow \mathrm{Hom}_{\mathbf{E}}(-, \Omega)$ is a natural transformation which maps the quantities $\mathrm{Hom}_{\mathbf{E}}(C, \Psi)$ on an object $C$ to the predicates $\mathrm{Hom}_{\mathbf{E}}(C, \Omega)$ on $C$, which are precisely subobjects of $C$.

## B.2 Equality and premetric

Next, we take a closer look at a concrete and important predicate — equality — and its corresponding quantities.

**Definition 21.** In a category with a subobject classifier $\top : 1 \to \Omega$, the *equality predicate* $=_C : C \times C \to \Omega$ on an object $C$ is the classifying morphism of the diagonal morphism $\Delta_C : C \rightarrowtail C \times C := \langle \mathrm{id}_C, \mathrm{id}_C \rangle$.

By Lemma 8, a quantity $d_C : C \times C \to \Psi$ is a quantifying morphism of $\Delta_C$ if and only if $\kappa \circ d_C = =_C$, depicted in the following diagram:

$$
\begin{array}{ccccc}
C & \dashrightarrow & 1 & \dashrightarrow & 1 \\
\Delta_C \downarrow & \lrcorner & o \downarrow & \lrcorner & \downarrow \top \\
C \times C & \xrightarrow{d_C} & \Psi & \dashrightarrow{\kappa} & \Omega
\end{array}
\tag{40}
$$
$$=_C$$

In **Set**, we have the following definitions:

**Definition 22.** A *premetric* on a set $C$ is a binary function $d_C : C \times C \to [0, \infty]$ such that

$$\forall c \in C.\ d_C(c, c) = 0. \tag{41}$$

Or equivalently, $d_C \circ \Delta_C = 0_C$, depicted in the following diagram:

$$
\begin{array}{ccc}
C & \longrightarrow & \{*\} \\
\Delta_C \downarrow & & \downarrow 0 \\
C \times C & \xrightarrow{d_C} & [0, \infty]
\end{array}
\tag{42}
$$

**Definition 23.** A *strict premetric* on a set $C$ is a premetric $d_C : C \times C \to [0, \infty]$ such that

$$\forall c_1 \in C.\ \forall c_2 \in C.\ (d_C(c_1, c_2) = 0) \to (c_1 =_C c_2). \tag{43}$$

Or equivalently, $\Delta_C$ is a pullback of $0$ along $d_C$:

$$
\begin{array}{ccc}
C & \longrightarrow & \{*\} \\
\Delta_C \downarrow & \lrcorner & \downarrow 0 \\
C \times C & \xrightarrow{d_C} & [0, \infty]
\end{array}
\tag{44}
$$

In other words, strict premetrics are precisely quantifying morphisms of the diagonal morphism $\Delta_C$ in the category **Set** with $0 : \{*\} \to [0, \infty]$ as a subobject quantifier.

Note that the symmetry

$$\forall c_1 \in C.\ \forall c_2 \in C.\ d_C(c_1, c_2) = d_C(c_2, c_1) \tag{45}$$

and the triangle inequality

$$\forall c_1 \in C.\ \forall c_2 \in C.\ \forall c_3 \in C.\ d_C(c_1, c_2) + d_C(c_2, c_3) \geq d_C(c_1, c_3), \tag{46}$$

which make $d_C$ a *metric*, are not required. The addition $+$ and the order $\geq$ on the set $[0, \infty]$ are not needed to define a strict premetric. However, they are necessary for defining other operations and properties, which will be discussed in the next subsection.

## B.3 Preorder

There is a preorder $\subseteq_C$ of inclusion defined on the set $\mathrm{Sub}_\mathbf{E}(C)$ of subobjects of an object $C$: for two subobjects $a : A \rightarrowtail C$ and $b : B \rightarrowtail C$, $a \subseteq_C b$ if and only if there exists a morphism $f : A \to B$ such that $a = b \circ f$:

$$
\begin{array}{ccc}
A & \xrightarrow{f} & B \\
& {}_a \searrow \quad \swarrow {}_b & \\
& C &
\end{array}
\tag{47}
$$

This preorder on the subobjects $\mathrm{Sub}_\mathbf{E}(C)$ induces a preorder on the predicates $\mathrm{Hom}_\mathbf{E}(C, \Omega)$ via the isomorphism. We can generalize this construction and define a preorder on other hom-sets:

**Definition 24.** In a category $\mathbf{E}$ with pullbacks, the inclusion preorder $\subseteq_C$ on the set $\mathrm{Sub}_{\mathbf{E}}(C)$ of subobjects of an object $C$ and a subobject $m : S \rightarrowtail T$ induce a preorder $\preceq_C^m$ on the hom-set $\mathrm{Hom}_{\mathbf{E}}(C,T)$ via pullback of $m$: for any two morphisms $f_1, f_2 : C \to T$, $f_1 \preceq_C^m f_2$ if and only if $f_1^* m \subseteq_C f_2^* m$.

$$
\begin{array}{ccc}
f^* S & \longrightarrow & S \\
{\scriptstyle f^* m}\big\downarrow & \lrcorner & \big\downarrow {\scriptstyle m} \\
C & \xrightarrow{\ f\ } & T
\end{array}
\tag{48}
$$

Based on this definition, we can explore the preorders on any hom-sets. From now, we assume that $\mathbf{E}$ is a category with necessary structures that we need.

Then, the preorder of predicates is $\preceq_C^\top$ on $\mathrm{Hom}_{\mathbf{E}}(C,\Omega)$, and the preorder of quantities is $\preceq_C^o$ on $\mathrm{Hom}_{\mathbf{E}}(C,\Psi)$. By Lemma 7, we know that for two quantities $q_1, q_2 : C \to \Psi$, $q_1 \preceq_C^o q_2$ if and only if $(\kappa \circ q_1) \preceq_C^\top (\kappa \circ q_2)$, which means that $\mathrm{Hom}_{\mathbf{E}}(C,\kappa)$ is an order-preserving function from $(\mathrm{Hom}_{\mathbf{E}}(C,\Psi), \preceq_C^o)$ to $(\mathrm{Hom}_{\mathbf{E}}(C,\Omega), \preceq_C^\top)$.

Next, we will explore the structures of $\mathrm{Hom}_{\mathbf{E}}(C,\Omega)$ and $\mathrm{Hom}_{\mathbf{E}}(C,\Psi)$. To begin with, $\top_C$ is a top in $\mathrm{Hom}_{\mathbf{E}}(C,\Omega)$, and $o_C$ is a top in $\mathrm{Hom}_{\mathbf{E}}(C,\Psi)$, because $\mathrm{id}_C$ is a top in $\mathrm{Sub}_{\mathbf{E}}(C)$.

$$
\begin{array}{ccccc}
C & \dashrightarrow & 1 & \dashrightarrow & 1 \\
{\scriptstyle \mathrm{id}_C}\big\downarrow & \lrcorner & {\scriptstyle o}\big\downarrow & \lrcorner & \big\downarrow {\scriptstyle \top} \\
C & \xrightarrow{\ o_C\ } & \Psi & \dashrightarrow{\ \kappa\ } & \Omega
\end{array}
\tag{49}
$$
$$\top_C$$

The inclusion preorder on $\mathrm{Sub}_{\mathbf{E}}(C)$ also has a bottom — the initial morphism $i_C : 0 \rightarrowtail C$ from an initial object $0$ in the category $\mathbf{E}$ to the object $C$. Then, its classifying morphism $\bot_C : C \to \Omega$ is a bottom in $\mathrm{Hom}_{\mathbf{E}}(C,\Omega)$. It can be proven that $\bot_C$ is a constant morphism with value $\bot : 1 \to \Omega$, which is the classifying morphism of the initial/terminal morphism $0 \rightarrowtail 1$. Any quantifying morphism $\psi_C : C \to \Psi$ of $i_C$ is a bottom in $\mathrm{Hom}_{\mathbf{E}}(C,\Psi)$, but it is not necessarily a constant morphism.

$$
\begin{array}{ccccc}
0 & \dashrightarrow & 1 & \dashrightarrow & 1 \\
{\scriptstyle i_C}\big\downarrow & \lrcorner & {\scriptstyle o}\big\downarrow & \lrcorner & \big\downarrow {\scriptstyle \top} \\
C & \xrightarrow{\ \psi_C\ } & \Psi & \dashrightarrow{\ \kappa\ } & \Omega
\end{array}
\tag{50}
$$
$$\bot_C$$

The preorder on the global elements plays a special role:

**Example 11.** In $\mathbf{Set}$, $\preceq_1^\top$, also denoted by $\vdash$, is a preorder on the set $\{\top, \bot\}$ with only one non-identity relation $\bot \vdash \top$.

**Example 12.** In $\mathbf{Set}$, $\preceq_1^0$ is a preorder on the set $[0, \infty]$ where $n \preceq_1^0 0$ for any number $n$, and $m \preceq_1^0 n$ and $n \preceq_1^0 m$ for any positive numbers $m$ and $n$.

By definition, $(\{\top, \bot\}, \vdash)$ and $([0, \infty], \preceq_1^0)$ are equivalent. However, we can consider a *suborder* of $([0, \infty], \preceq_1^0)$, e.g., the usual "*greater than or equal to*" $\geq$ total order, to further differentiate positive numbers. Note that $0$ remains the top in this suborder $([0, \infty], \geq)$.

### B.4 Operation: algebra over the product endofunctor

In an elementary topos $\mathbf{E}$, the subobjects $\mathrm{Sub}_{\mathbf{E}}(C)$ of an object $C$ not only forms a preorder by inclusion but also are equipped with certain *set operations* (e.g., intersection and disjoint union), which are defined in terms of the universal properties of their corresponding *order operations* (e.g., meet and join). Further, these operations are reflected in the structures of the subobject classifier (e.g., conjunction and disjunction).

Here, we establish a link between the structures of the subobject classifier and those of a subobject quantifier. Our primary result is as follows:

**Theorem 10.** *Consider a category with a subobject classifier $\top : 1 \rightarrowtail \Omega$, a subobject quantifier $o : 1 \rightarrowtail \Psi$, and a quantizer $\kappa : \Psi \to \Omega$.*

*Let $n$ be a natural number. Let $\beta : \Omega^n \to \Omega$ be an $n$-ary logical operation on $\Omega$, and let $\alpha : \Psi^n \to \Psi$ be an $n$-ary quantitative operation on $\Psi$.*

*For $i \in \{1, \ldots, n\}$, let $p_i : C \to \Omega$ be a predicate on an object $C$, and let $q_i : C \to \Psi$ be a quantity on the object $C$ such that $p_i = \kappa \circ q_i$. Let $p = \langle p_1, \ldots, p_n \rangle$ be the tupling of the predicates, and let $q = \langle q_1, \ldots, q_n \rangle$ be the tupling of the quantities. Let $b : B \rightarrowtail C := (\beta \circ p)^* \top$ be the subobject classified by $\beta \circ p$, and let $a : A \rightarrowtail C := (\alpha \circ q)^* o$ be the subobject quantified by $\alpha \circ q$.*

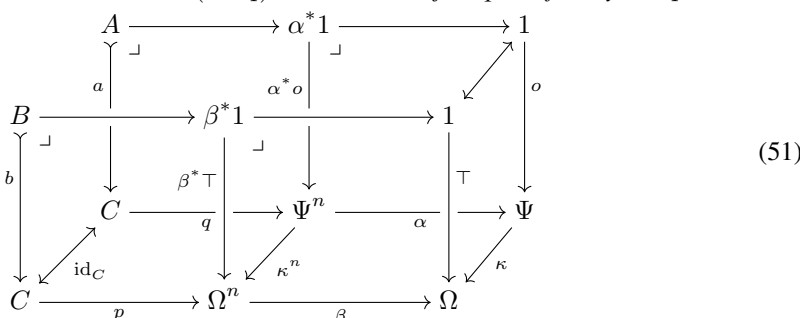

$$(51)$$

*Then, we have*

    *(i)* $\kappa^n \circ q = p$
    *(ii)* $(\beta \circ \kappa^n \circ \alpha^* o = \top_{\alpha^* 1}) \to ((\beta \circ p) \circ a = \top_A)$
    *(iii)* $(\beta \circ \kappa^n = \kappa \circ \alpha) \to (\beta \circ \kappa^n \circ \alpha^* o = \top_{\alpha^* 1})$
    *(iv)* $(\beta \circ \kappa^n = \kappa \circ \alpha) \to (\kappa \circ (\alpha \circ q) = \beta \circ p)$
    *(v)* $(\beta \circ \kappa^n = \kappa \circ \alpha) \to ((\alpha \circ q) \circ b = o_B)$

For convenience, we call an $n$-ary operation $\alpha : \Psi^n \to \Psi$ (as an algebra over the product endofunctor $(-)^n$) *homomorphic* to $\beta : \Omega^n \to \Omega$ via a morphism $\kappa : \Psi \to \Omega$ if

$$\beta \circ \kappa^n = \kappa \circ \alpha \tag{52}$$

and *subhomomorphic* to $\beta : \Omega^n \to \Omega$ if it satisfies the condition

$$\beta \circ \kappa^n \circ \alpha^* o = \top_{\alpha^* 1}. \tag{53}$$

*Proof.* (i) follows from the property of tupling and product: $\kappa^n \circ q = \langle \kappa \circ q_1, \ldots, \kappa \circ q_n \rangle = \langle p_1, \ldots, p_n \rangle = p$.

(ii) states that if $\alpha$ is subhomomorphic to $\beta$, then $\beta \circ p$ is the classifying morphism of the subobject quantified by $\alpha \circ q$.

$$\beta \circ p \circ a \tag{54}$$
$$= \beta \circ \kappa^n \circ q \circ a \hspace{4cm} \text{(i)} \tag{55}$$
$$= \beta \circ \kappa^n \circ \alpha^* o \circ (\alpha^* o)^* q \hspace{2.5cm} \text{(pullback)} \tag{56}$$
$$= \top_{\alpha^* 1} \circ (\alpha^* o)^* q \hspace{2.5cm} \text{(subhomomorphism)} \tag{57}$$
$$= \top_A \hspace{5cm} \text{(composition)} \tag{58}$$

(iii) means that $\alpha$ being homomorphic to $\beta$ is a stronger condition than being merely subhomomorphic to $\beta$.

$$\beta \circ \kappa^n \circ \alpha^* o \tag{59}$$
$$= \kappa \circ \alpha \circ \alpha^* o \hspace{3.5cm} \text{(homomorphism)} \tag{60}$$
$$= \kappa \circ o \circ o^* \alpha \hspace{4cm} \text{(pullback)} \tag{61}$$
$$= \top_{\alpha^* 1} \hspace{4.5cm} \text{(composition)} \tag{62}$$

(iv) shows the relationship between the predicate $\beta \circ p$ and the quantity $\alpha \circ q$ when $\alpha$ is homomorphic to $\beta$.

$$\kappa \circ \alpha \circ q \tag{63}$$
$$= \beta \circ \kappa^n \circ q \hspace{3.5cm} \text{(homomorphism)} \tag{64}$$
$$= \beta \circ p \hspace{5cm} \text{(i)} \tag{65}$$

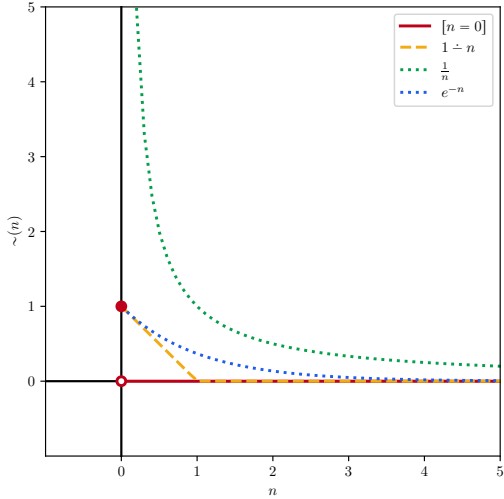

Figure 3: Negation

(v) means that if $\alpha$ is homomorphic to $\beta$, then $\alpha \circ q$ is a quantifying morphism of $b$.

$$\kappa \circ \alpha \circ q \circ b \tag{66}$$
$$= \beta \circ p \circ b \qquad \qquad \text{(iv)} \tag{67}$$
$$= \top_B \qquad \qquad \text{(pullback)} \tag{68}$$

$\alpha \circ q \circ b = o_B$ follows from Lemma 9. $\qquad\square$

In summary, if $\alpha$ is subhomomorphic to $\beta$, then $a$ is included in $b$ ((ii)); if $\alpha$ is homomorphic to $\beta$, then $a$ and $b$ are isomorphic ((iii) and (v)). We consider this weaker condition because subhomomorphic but non-homomorphic operations may exhibit favorable properties in other aspects, such as continuity. We will discuss several concrete examples in the following subsections.

### B.5 Negation

First, let us take a closer look at a unary logical operation — *negation* $\neg : \Omega \to \Omega$, which is defined as the classifying morphism of $\bot : 1 \to \Omega$. Recall that $\bot$ is the classifying morphism of $0 \rightarrowtail 1$.

Let us consider a unary quantitative operation $\sim : \Psi \to \Psi$. If $\sim$ is homomorphic to $\neg$ via the quantizer $\kappa$, it means that $\neg \circ \kappa = \kappa \circ \sim$, or the following diagram commutes:

$$
\begin{array}{ccc}
\Psi & \xrightarrow{\ \sim\ } & \Psi \\
{\scriptstyle \kappa}\downarrow & & \downarrow{\scriptstyle \kappa} \\
\Omega & \xrightarrow{\ \neg\ } & \Omega
\end{array}
\tag{69}
$$

Let us consider the set $[0, \infty]$ in **Set**. A quantitative operation $\sim : [0, \infty] \to [0, \infty]$ *homomorphic to* the negation $\neg : \{\top, \bot\} \to \{\top, \bot\}$ is a function

$$\sim(n) := [n = 0] \times n_0 = \begin{cases} n_0 & n = 0, \\ 0 & n > 0, \end{cases} \tag{70}$$

which maps $0$ to a non-zero number $n_0$ and any non-zero number to $0$.[11] However, this function is discontinuous at $0$. On the other hand, if we consider a quantitative operation $\sim$ *subhomomorphic* to the negation $\neg$, then the only requirement is that for all $n$, $\sim(n) = 0$ implies $n > 0$, or, by contraposition, $\sim(0) > 0$. Continuous choices include the *hinge function* $n \mapsto 1 \dotdiv n = \max\{1 - n, 0\}$, *reciprocal function* $n \mapsto \frac{1}{n}$, and *exponential decay function* $n \mapsto e^{-n}$ (see Fig. 3). Note that the latter

---

[11] $[-] : \{\top, \bot\} \to [0, \infty] := \begin{cases} \bot \mapsto 0, \\ \top \mapsto 1. \end{cases}$

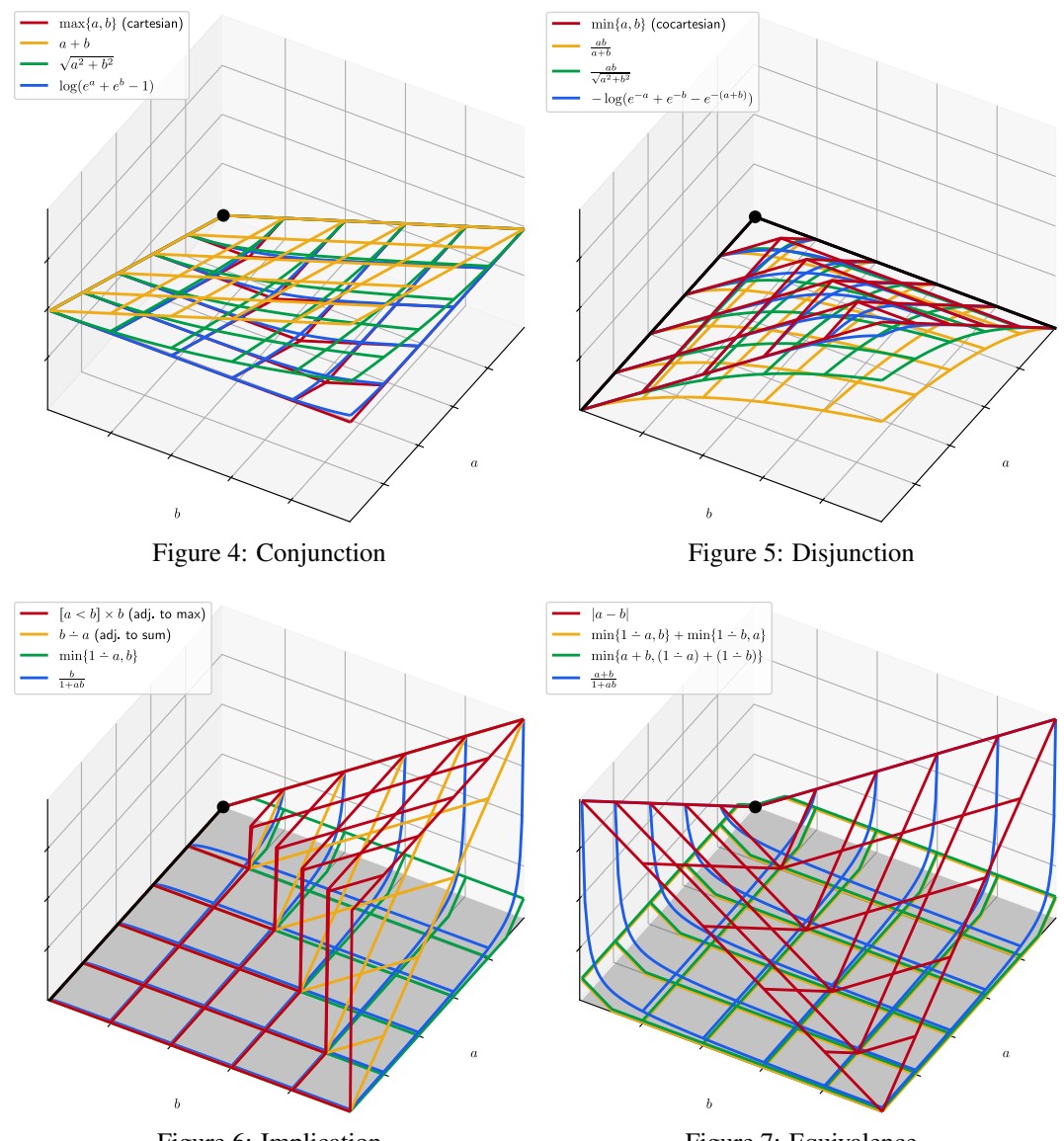

Figure 4: Conjunction

Figure 5: Disjunction

Figure 6: Implication

Figure 7: Equivalence

two are actually homomorphic to the constant false $\perp$ because their outputs are always non-zero. The hinge function $1 \dotminus q$, as discussed later, can be seen as derived from the implication $p \rightarrow \perp$.

In this way, if we have a quantity $q$ for a predicate $p$, we can obtain a quantity $\sim q$ for the negation $\neg p$ of the predicate as well. If the quantitative operation $\sim$ is subhomomorphic but not homomorphic to the logical operation $\neg$, we can guarantee that for any $x$, $\sim q(x) = 0$ implies $\neg p(x) = \top$, but not vice versa.

## B.6 Conjunction

Next, let us move on to an important binary logical operation — *conjunction* $\wedge : \Omega \times \Omega \rightarrow \Omega$, which is defined as the classifying morphism of $\langle \top, \top \rangle : 1 \rightarrowtail \Omega \times \Omega$. Similarly, we consider a binary quantitative operation $\otimes : \Psi \times \Psi \rightarrow \Psi$ homomorphic to the conjunction $\wedge$ via the quantizer $\kappa$:

$$
\begin{array}{ccc}
\Psi \times \Psi & \xrightarrow{\otimes} & \Psi \\
{\scriptstyle \kappa \times \kappa} \downarrow & & \downarrow {\scriptstyle \kappa} \\
\Omega \times \Omega & \xrightarrow[\wedge]{} & \Omega
\end{array}
\tag{71}
$$

By abuse of notation, the conjunction $\wedge$ also denotes a binary operation on the set $\mathrm{Hom}_{\mathbf{E}}(C, \Omega)$ of predicates, such that for any two predicates $p_1, p_2 \in \mathrm{Hom}_{\mathbf{E}}(C, \Omega)$,

$$p_1 \wedge p_2 := \wedge \circ \langle p_1, p_2 \rangle. \tag{72}$$

The same goes for the quantitative operation $\otimes$.

For the conjunction, we can prove a stronger result:

**Theorem 11.** *Consider a category with a subobject classifier $\top : 1 \rightarrowtail \Omega$, a subobject quantifier $o : 1 \rightarrowtail \Psi$, and a quantizer $\kappa : \Psi \to \Omega$.*

*Let the conjunction $\wedge : \Omega \times \Omega \to \Omega$ be the classifying morphism of $\langle \top, \top \rangle : 1 \rightarrowtail \Omega \times \Omega$, and let $\otimes : \Psi \times \Psi \to \Psi$ be a binary operation on $\Psi$ homomorphic to the conjunction $\wedge$ via the quantizer $\kappa$.*

*Let $p_1, p_2 : C \to \Omega$ be two predicates on an object $C$, and let $q_1, q_2 : C \to \Psi$ be two quantities on the object $C$ such that $p_1 = \kappa \circ q_1$ and $p_2 = \kappa \circ q_2$. Let $p = \langle p_1, p_2 \rangle$ be the pairing of the predicates, and let $q = \langle q_1, q_2 \rangle$ be the pairing of the quantities. Let $b : B \rightarrowtail C := (p_1 \wedge p_2)^* \top$ be the subobject classified by $p_1 \wedge p_2$.*

$$\begin{array}{ccccc}
B & \longrightarrow & 1 & \longrightarrow & 1 \\
\downarrow{\scriptstyle b} & \lrcorner & \downarrow{\scriptstyle \langle o,o \rangle} & \lrcorner & \downarrow{\scriptstyle o} \\
C & \xrightarrow{\langle q_1,q_2 \rangle} & \Psi \times \Psi & \xrightarrow{\otimes} & \Psi \\
\downarrow{\scriptstyle \mathrm{id}_C} & & \downarrow{\scriptstyle \kappa \times \kappa} & & \downarrow{\scriptstyle \kappa} \\
C & \xrightarrow{\langle p_1,p_2 \rangle} & \Omega \times \Omega & \xrightarrow{\wedge} & \Omega
\end{array} \tag{73}$$

*Then, $\otimes$ is a quantifying morphism of $\langle o, o \rangle$, and $b$ is a pullback of $\langle o, o \rangle$ along $\langle q_1, q_2 \rangle$.*

*Proof.* If $q_1 \otimes q_2 = o_C$, then $(\kappa \circ q_1) \wedge (\kappa \circ q_2) = \top_C$, which leads to $\langle \kappa \circ q_1, \kappa \circ q_2 \rangle = \langle \top, \top \rangle \circ e_C = \langle \top_C, \top_C \rangle$ due to the universal property of pullback. Then, we have $\kappa \circ q_1 = \kappa \circ q_2 = \top_C$ due to the universal property of pairing, and consequently $q_1 = q_2 = o_C$ according to Lemma 9, i.e., $\langle q_1, q_2 \rangle = \langle o, o \rangle \circ e_C$. Therefore, $\langle o, o \rangle$ is a pullback of $o$ along $\otimes$. According to Theorem 10, $b$ is a pullback of $o$ along $q_1 \otimes q_2$. Then, following the pullback lemma, $b$ is a pullback of $\langle o, o \rangle$ along $\langle q_1, q_2 \rangle$. $\qquad\square$

In other words, the pullback square symbols in Eq. (73) are unambiguous — the top row, the left column, the right column, the top-left square, and the top-right square are all pullbacks.

Note that for any quantities $a, b, c : C \to \Psi$, we have

$$\kappa \circ ((a \otimes b) \otimes c) = \kappa \circ (a \otimes (b \otimes c)), \tag{74}$$
$$\kappa \circ (a \otimes b) = \kappa \circ (b \otimes a), \tag{75}$$
$$\kappa \circ (o_C \otimes a) = \kappa \circ a, \tag{76}$$

due to the associativity, commutativity, and unitality of the conjunction, but the quantitative operation $\otimes$ is not required to satisfy these properties, i.e., it is possible that

$$(a \otimes b) \otimes c \neq a \otimes (b \otimes c), \tag{77}$$
$$a \otimes b \neq b \otimes a, \tag{78}$$
$$o_C \otimes a \neq a. \tag{79}$$

However, in the following examples, we mainly consider quantitative operations $\otimes$ such that these properties are satisfied. In such cases, $(\mathrm{Hom}_{\mathbf{E}}(C, \Psi), \otimes, o_C)$ forms a commutative monoid, and consequently $\mathrm{Hom}_{\mathbf{E}}(C, \kappa)$ is a monoid homomorphism from it to $(\mathrm{Hom}_{\mathbf{E}}(C, \Omega), \wedge, \top_C)$.

In Appendix B.3, we introduced preorder structures on the predicates $\mathrm{Hom}_{\mathbf{E}}(C, \Omega)$ and the quantities $\mathrm{Hom}_{\mathbf{E}}(C, \Psi)$. The conjunction $\wedge$ is a commutative monoidal structure compatible with the preorder $(\mathrm{Hom}_{\mathbf{E}}(C, \Omega), \preceq_C^\top)$ because it is the meet operation. For the set $\mathrm{Hom}_{\mathbf{E}}(C, \Psi)$ of quantities, we can choose the quantitative operation $\otimes$ to be the meet operation as well. Alternatively, we can only require it to be compatible with the preorder, in the sense that the monoid product is order-preserving: for any quantities $q_1, q_1', q_2, q_2' \in \mathrm{Hom}_{\mathbf{E}}(C, \Psi)$, if $q_1 \preceq_C^o q_1'$ and $q_2 \preceq_C^o q_2'$, then $q_1 \otimes q_2 \preceq_C^o q_1' \otimes q_2'$. In other words, we require $(\mathrm{Hom}_{\mathbf{E}}(C, \Psi), \preceq_C^o, \otimes, o_C)$ to be a *symmetric monoidal preorder* [Fong and Spivak, 2019, Definition 2.2].

**Example 13.** Let us consider two commutative monoidal structures on the preorder $([0,\infty], \geq)$. The max operation $\max : [0,\infty] \times [0,\infty] \to [0,\infty]$ is the meet, i.e., cartesian product, while the addition $+ : [0,\infty] \times [0,\infty] \to [0,\infty]$ is a monoidal product. They are both semicartesian because the top $0$ is the unit.

Note that $([0,\infty], +, 0)$, $([0,1], \times, 1)$, and $([1,\infty], \times, 1)$ are isomorphic to each other with the following isomorphisms:

$$([0,\infty], +, 0)$$

$$\begin{array}{c}
\nearrow\ \ \ \nwarrow \\
{}_{-\log(x)}\ \ {}_{\exp(-x)}\ \ {}_{\log(x)}\ \ {}^{\exp(x)} \\
\swarrow\ \ \ \searrow \\
([0,1], \times, 1) \xrightleftharpoons[\frac{1}{x}]{\frac{1}{x}} ([1,\infty], \times, 1)
\end{array} \tag{80}$$

We can also induce a monoidal structure on a set if it is isomorphic to a monoid:

**Lemma 12.** *Let $f : A \rightleftarrows B : g$ be a pair of bijections between sets $A$ and $B$. If $(B, \otimes_B, I_B)$ is a monoid, then $(A, \otimes_A := g \circ \otimes_B \circ (f \times f), I_A := g \circ I_B)$ is also a monoid.*

*Proof.* Associativity:

$$(a \otimes_A b) \otimes_A c \tag{81}$$
$$= g((f(a) \otimes_B f(b)) \otimes_B f(c)) \tag{82}$$
$$= g(f(a) \otimes_B (f(b) \otimes_B f(c))) \tag{83}$$
$$= a \otimes_A (b \otimes_A c) \tag{84}$$

Left unitality:

$$I_A \otimes_A a \tag{85}$$
$$= g(I_B) \otimes_A a \tag{86}$$
$$= g(f(g(I_B)) \otimes_B f(a)) \tag{87}$$
$$= g(I_B \otimes_B f(a)) \tag{88}$$
$$= g(f(a)) \tag{89}$$
$$= a \tag{90}$$

Right unitality can be proven similarly. $\qquad\square$

In this way, we can obtain a richer choice of monoidal structures on the set $[0,\infty]$ beyond the addition (see Fig. 4):

**Example 14** (Semicartesian monoidal product)**.**

$$e^x - 1 : ([0,\infty], \otimes, 0) \rightleftarrows ([0,\infty], +, 0) : \log(1+x) \qquad a \otimes b := \log(e^a + e^b - 1) \tag{91}$$

$$x^2 : ([0,\infty], \otimes, 0) \rightleftarrows ([0,\infty], +, 0) : \sqrt{x} \qquad a \otimes b := \sqrt{a^2 + b^2} \tag{92}$$

$$\sqrt{x} : ([0,\infty], \otimes, 0) \rightleftarrows ([0,\infty], +, 0) : x^2 \qquad a \otimes b := a + b + 2\sqrt{ab} \tag{93}$$

$$x + 1 : ([0,\infty], \otimes, 0) \rightleftarrows ([1,\infty], \times, 1) : x - 1 \qquad a \otimes b := a + b + ab \tag{94}$$

In summary, the max operation on $[0,\infty]$ can be regarded as a *continuous conjunction*, whereas a semicartesian monoidal product, such as the addition, can be viewed as a *soft max*.

## B.7 Disjunction

Dually, the *disjunction* $\vee : \Omega \times \Omega \to \Omega$ reflects the join of the inclusion preorder of subobjects. Similarly, we want to find a quantitative operation $\oplus : \Psi \times \Psi \to \Psi$ homomorphic to the disjunction $\vee$ via the quantizer $\kappa$:

$$\begin{array}{ccc}
\Psi \times \Psi & \xrightarrow{\oplus} & \Psi \\
{\scriptstyle \kappa \times \kappa} \downarrow & & \downarrow {\scriptstyle \kappa} \\
\Omega \times \Omega & \xrightarrow{\vee} & \Omega
\end{array} \tag{95}$$

Using the same technique as in Lemma 12, we can obtain several monoidal structures on the set $[0,\infty]$ homomorphic to the disjunction (see Fig. 5):

**Example 15** (Semicocartesian monoidal product).

$$\frac{1}{x} : ([0,\infty], \oplus, \infty) \rightleftarrows ([0,\infty], +, 0) : \frac{1}{x} \qquad a \oplus b := \frac{ab}{a+b} \tag{96}$$

$$\frac{1}{x^2} : ([0,\infty], \oplus, \infty) \rightleftarrows ([0,\infty], +, 0) : \frac{1}{\sqrt{x}} \qquad a \oplus b := \frac{ab}{\sqrt{a^2 + b^2}} \tag{97}$$

$$1 - e^{-x} : ([0,\infty], \oplus, \infty) \rightleftarrows ([0,1], \times, 1) : -\log(1-x) \quad a \oplus b := -\log(e^{-a} + e^{-b} - e^{-(a+b)}) \tag{98}$$

$$\tanh : ([0,\infty], \oplus, \infty) \rightleftarrows ([0,1], \times, 1) : \mathrm{arctanh} \qquad a \oplus b := \mathrm{arctanh}(\tanh(a)\tanh(b)) \tag{99}$$

However, we usually choose the quantitative operation $\oplus$ to be the join operation $\min$ of the preorder $([0,\infty], \geq)$. In this way, $\otimes$ distributes over $\oplus$, i.e., for any quantities $a, b, c : C \to \Psi$, we have

$$a \otimes (b \oplus c) = (a \otimes b) \oplus (a \otimes c). \tag{100}$$

A typical example is the min-plus semiring $([0,\infty], \min, +)$ [Pin, 1998].

## B.8 Implication

Finally, let us construct a quantitative counterpart of the *implication* $\to : \Omega \times \Omega \to \Omega$, which is right adjoint to the conjunction $\wedge$. For global elements $a, b, c : 1 \to \Omega$, this means that

$$c \wedge a \vdash b \text{ if and only if } c \vdash a \to b. \tag{101}$$

There are two ways to construct a quantitative operation $\multimap : \Psi \times \Psi \to \Psi$ corresponding to the implication $\to$. One way is to find a quantitative operation $\multimap$ homomorphic to the implication $\to$ via the quantizer $\kappa$:

$$\begin{array}{ccc} \Psi \times \Psi & \xrightarrow{\ \multimap\ } & \Psi \\ {\scriptstyle \kappa \times \kappa} \downarrow & & \downarrow {\scriptstyle \kappa} \\ \Omega \times \Omega & \xrightarrow[\ \to\ ]{} & \Omega \end{array} \tag{102}$$

**Example 16.** For the set $[0,\infty]$, a quantitative operation $\multimap : [0,\infty] \times [0,\infty] \to [0,\infty]$ homomorphic to the implication $\to : \{\top, \bot\} \times \{\top, \bot\} \to \{\top, \bot\}$ is a function

$$(a, b) \mapsto [a = 0] \times [b > 0] \times f(b) = \begin{cases} f(b) & a = 0 \text{ and } b > 0, \\ 0 & \text{otherwise,} \end{cases} \tag{103}$$

where $f : (0,\infty] \to (0,\infty]$ is an arbitrary function to non-zero numbers. Note that this function is discontinuous at the line $a = 0$ and $b > 0$ (cf. Appendix B.5).

The other way is to find a quantitative operation $\multimap$ right adjoint to an operation $\otimes$ homomorphic to the conjunction $\wedge$, i.e., the *internal hom* of the *monoidal closed preorder* [Fong and Spivak, 2019, Definition 2.79].

**Example 17.** For the meet-semilattice $([0,\infty], \geq, \max, 0)$, the function

$$(a, b) \mapsto [a < b] \times b = \begin{cases} 0 & a \geq b, \\ b & a < b, \end{cases} \tag{104}$$

is right adjoint to the max because

$$\max\{c, a\} \geq b \text{ if and only if } c \geq \begin{cases} 0 & a \geq b, \\ b & a < b. \end{cases} \tag{105}$$

While this function is subhomomorphic to the implication, it is still discontinuous at the line $a = b$.

**Example 18.** For the monoidal preorder $([0,\infty], \geq, +, 0)$, the truncated subtraction $\dot{-}$ (a.k.a. *monus* [Amer, 1984])

$$b \dot{-} a := \max\{b - a, 0\} = \begin{cases} 0 & a \geq b, \\ b - a & a < b, \end{cases} \tag{106}$$

is right adjoint to the addition because

$$c + a \geq b \text{ if and only if } c \geq b \mathbin{\dot{-}} a. \tag{107}$$

The truncated subtraction is continuous and subhomomorphic to the implication. Note that the hinge function $n \mapsto \max\{1 - n, 0\} = 1 \mathbin{\dot{-}} n$ for the negation can be interpreted as the quantitative operation derived from $\neg n = n \to \bot$, where $1$ is homomorphic to the constant false $\bot$.

Similarly to Lemma 12, if two symmetric monoidal preorders are isomorphic and one of them is closed, we can induce that the other is also closed:

**Lemma 13.** *Let* $f : A \rightleftarrows B : g$ *be a pair of isomorphisms between symmetric monoidal preorders* $(A, \preceq_A, \otimes_A, I_A)$ *and* $(B, \preceq_B, \otimes_B, I_B)$. *If* $(B, \preceq_B, \otimes_B, I_B, \multimap_B)$ *is closed, then* $(A, \preceq_A, \otimes_A, I_A, \multimap_A := g \circ \multimap_B \circ (f \times f))$ *is also closed.*

*Proof.* For all $a, b, c \in A$, we have

$$c \preceq_A (a \multimap_A b) \tag{108}$$
$$\equiv c \preceq_A g(f(a) \multimap_B f(b)) \tag{109}$$
$$\equiv f(c) \preceq_B f(a) \multimap_B f(b) \tag{110}$$
$$\equiv f(c) \otimes_B f(a) \preceq_B f(b) \tag{111}$$
$$\equiv c \otimes_A a \preceq_A b \tag{112}$$

This means that $\multimap_A$ is right adjoint to $\otimes_A$. $\qquad\square$

In this way, we can find the internal homs corresponding to the monoidal products discussed in Appendix B.6 (see Fig. 6).

As a side note, we can use the quantitative operations discussed above to define quantitative operations for other logical connectives. For example, the logical equivalence $a \leftrightarrow b$ can be represented as $(a \to b) \land (b \to a)$ (bi-implication), $(\neg a \lor b) \land (\neg b \lor a)$ (conjunctive normal form (CNF)), or $(a \land b) \lor (\neg a \land \neg b)$ (disjunctive normal form (DNF)), and its quantitative operations can be defined accordingly. Some examples are shown in Fig. 7.

### B.9 Heyting algebra, quantale, and ordered semiring

Now, having constructed the logical operations and their corresponding quantitative operations, we can compare the structures of the subobject classifier with those of a subobject quantifier.

It is known that the global elements of the subobject classifier with the logical operations form a Heyting algebra $(\Omega, \vdash, \top, \bot, \land, \lor, \to)$:

**Definition 25.** A *Heyting algebra* is a cartesian closed bounded lattice.

In categorical terms, $\top$ is the terminal object, $\bot$ is the initial object, $\land$ is the product, $\lor$ is the coproduct, and $\to$ is the exponential in the preorder $(\Omega, \vdash)$ (as a thin category).

We weakened the requirements to construct the algebraic structures of the subobject quantifier, which usually forms what is called a quantale $(\Psi, \preceq, 1, 0, \otimes, \oplus, \multimap)$ [Mulvey, 1986, Dudzik, 2017]:

**Definition 26.** A (unital) *quantale* is a monoidal closed suplattice.

This means that we can consider a preorder $(\Psi, \preceq)$ on the subobject quantifier as a thin category, where $1$ is the terminal object, $0$ is the initial object, $\otimes$ is a monoidal product and not necessarily the product, $\oplus$ is still the coproduct, and $\multimap$ is the internal hom right adjoint to the monoidal product $\otimes$. An example is the Lawvere quantale $([0, \infty], \geq, 0, \infty, +, \min, \mathbin{\dot{-}})$ [Lawvere, 1973, Bacci et al., 2023]. Then, the quantizer $\kappa : \Psi \to \Omega$ is a homomorphism preserving some or all the structures.

Note that due to the adjoint functor theorem and the fact that left adjoints preserve colimits, the product distributes over the coproduct in a Heyting algebra, and the monoidal product distributes over the coproduct in a quantale. We can further relax the requirement for $\oplus$ to be the coproduct and instead consider a monoidal product (Appendix B.7). If we still require the distributivity for the two monoidal structures $\otimes$ and $\oplus$, the algebraic structure is an *ordered semiring* [Fujii, 2023]. Further investigation is left for future work.

### B.10 Quantifier: algebra over the exponentiation endofunctor

Up to this point, our focus has been on $n$-ary operations in propositional logic (Appendix B.4). Next, we introduce the universal quantification $\forall$ and existential quantification $\exists$ used in predicate logic and their quantitative counterparts.

Externally, the universal quantification and the existential quantification are right and left adjoint to the pullback of projection, respectively [Lawvere, 1969]. Internally, the universal quantifier $\forall_D : \Omega^D \to \Omega$ and existential quantifier $\exists_D : \Omega^D \to \Omega$ are given by morphisms from the power object $\Omega^D$ of an object $D$ to the subobject classifier $\Omega$.

Recall that the power object $\Omega^D$ is also an exponential object into the subobject classifier $\Omega$, so the universal quantifier $\forall_D$ and existential quantifier $\exists_D$ can also be viewed as algebras over the exponentiation endofunctor $(-)^D$ of exponentiation on the subobject classifier $\Omega$. Then, we can consider algebras on the subobject quantifier $\Psi$ homomorphic to them, which serve as quantitative counterparts of these quantifiers.

Our main result is as follows (cf. Theorem 10):

**Theorem 14.** *Consider an elementary topos with a subobject classifier $\top : 1 \rightarrowtail \Omega$, a subobject quantifier $o : 1 \rightarrowtail \Psi$, and a quantizer $\kappa : \Psi \to \Omega$.*

*Let $p : C \times D \to \Omega$ be a predicate on a product, and let $q : C \times D \to \Psi$ be a quantity such that $p = \kappa \circ q$. Let $\widehat{p} : C \to \Omega^D$ and $\widehat{q} : C \to \Psi^D$ be their exponential transposes.*

*Let $\beta_D : \Omega^D \to \Omega$ be a predicate, and let $\alpha_D : \Psi^D \to \Psi$ be a quantity homomorphic to $\beta_D$ via the quantizer $\kappa$, i.e., $\beta_D \circ \kappa^D = \kappa \circ \alpha_D$.*

*Let $b : B \rightarrowtail C := (\beta_D \circ \widehat{p})^* \top$ be the subobject classified by $\beta_D \circ \widehat{p}$, and let $a : A \rightarrowtail C := (\alpha_D \circ \widehat{q})^* o$ be the subobject quantified by $\alpha_D \circ \widehat{q}$.*

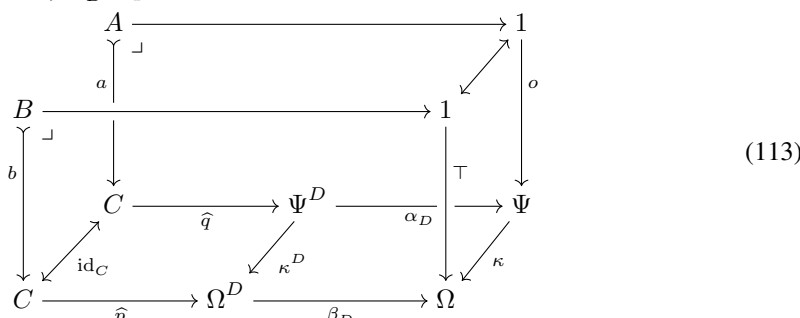

$$(113)$$

*Then, we have*

    *(i)* $\kappa^D \circ \widehat{q} = \widehat{p}$
    *(ii)* $\kappa \circ (\alpha_D \circ \widehat{q}) = \beta_D \circ \widehat{p}$
    *(iii)* $(\beta_D \circ \widehat{p}) \circ a = \top_A$
    *(iv)* $(\alpha_D \circ \widehat{q}) \circ b = o_B$

*Proof.* (i) follows from the property of exponential.

(ii) shows the relationship between the predicate $\beta_D \circ \widehat{p}$ and the quantity $\alpha_D \circ \widehat{q}$ when $\alpha_D$ is homomorphic to $\beta_D$.

$$\kappa \circ \alpha_D \circ \widehat{q} \tag{114}$$
$$= \beta_D \circ \kappa^D \circ \widehat{q} \qquad \text{(homomorphism)} \tag{115}$$
$$= \beta_D \circ \widehat{p} \qquad \text{(i)} \tag{116}$$

(iii) means that $\beta_D \circ \widehat{p}$ is a classifying morphism of $a$.

$$\beta_D \circ \widehat{p} \circ a \tag{117}$$
$$= \kappa \circ \alpha_D \circ \widehat{q} \circ a \qquad \text{(ii)} \tag{118}$$
$$= \kappa \circ o_A \qquad \text{(pullback)} \tag{119}$$
$$= \top_A \qquad \text{(composition)} \tag{120}$$

(iv) means that $\alpha_D \circ \widehat{q}$ is a quantifying morphism of $b$.

$$\kappa \circ \alpha_D \circ \widehat{q} \circ b \tag{121}$$
$$= \beta_D \circ \widehat{p} \circ b \qquad \text{(ii)} \quad (122)$$
$$= \top_B \qquad \text{(pullback)} \quad (123)$$

$\alpha_D \circ \widehat{q} \circ b = o_B$ follows from Lemma 9. $\qquad\qquad \square$

**Definition 27** (Universal aggregator). A *universal aggregator* $\triangledown_D : \Psi^D \to \Psi$ is a quantity that is homomorphic to the universal quantifier $\forall_D : \Omega^D \to \Omega$:

$$
\begin{array}{ccc}
\Psi^D & \xrightarrow{\triangledown_D} & \Psi \\
{\scriptstyle \kappa^D} \downarrow & & \downarrow {\scriptstyle \kappa} \\
\Omega^D & \xrightarrow[\forall_D]{} & \Omega
\end{array}
\tag{124}
$$

**Definition 28** (Existential aggregator). An *existential aggregator* $\triangle_D : \Psi^D \to \Psi$ is a quantity that is homomorphic to the existential quantifier $\exists_D : \Omega^D \to \Omega$:

$$
\begin{array}{ccc}
\Psi^D & \xrightarrow{\triangle_D} & \Psi \\
{\scriptstyle \kappa^D} \downarrow & & \downarrow {\scriptstyle \kappa} \\
\Omega^D & \xrightarrow[\exists_D]{} & \Omega
\end{array}
\tag{125}
$$

**Example 19.** For the set $[0, \infty]$, the canonical choices of universal aggregator $\triangledown_D$ and existential aggregator $\triangle_D$ are $\sup$ and $\inf$. If the set $D$ is finite, we can also use $\operatorname{sum}$ and $\operatorname{mean}$ as the universal aggregator. Non-examples of universal aggregator include $\operatorname{median}$ and $\operatorname{mode}$, which are not homomorphic to the universal quantifier.

Note that the universal quantifier and existential quantifier are commutative up to isomorphism:

$$\forall_{A \times B} \cong \forall_B \circ \forall_A^B \cong \forall_A \circ \forall_B^A \cong \forall_{B \times A}, \tag{126}$$
$$\exists_{A \times B} \cong \exists_B \circ \exists_A^B \cong \exists_A \circ \exists_B^A \cong \exists_{B \times A}. \tag{127}$$

However, we are free to choose different aggregators for different objects that are not necessarily commutative. For example, the sum of max is usually not equal to the max of sum.

### B.11 Enrichment

Lastly, we describe the conversion based on enrichment.

First, let us define the enriching category:

**Definition 29.** Let $(\Psi, \preceq, \otimes, \oplus, \multimap)$ be an internal quantale object in **Set**. We define $\Psi\text{-}\mathbf{Set}$ to be a category whose objects are tuples consisting of a set $C$, a $\Psi$-valued strict premetric $d_C$ on $C$, and universal and existential aggregators on $C$:

$$(C, d_C : C \times C \to \Psi, \triangledown_C, \triangle_C : \Psi^C \to \Psi), \tag{128}$$

and morphisms from $(A, d_A, \triangledown_A, \triangle_A)$ to $(B, d_B, \triangledown_B, \triangle_B)$ are functions $f : A \to B$.

**Definition 30.** A bifunctor $\boxtimes$ on $\Psi\text{-}\mathbf{Set}$ is given by the Cartesian product of sets and functions, together with the following products of strict premetrics and aggregators:

$$d_A \boxtimes d_B : (A \times B) \times (A \times B) \cong (A \times A) \times (B \times B) \xrightarrow{d_A \times d_B} \Psi \times \Psi \xrightarrow{\otimes} \Psi. \tag{129}$$

$$\triangledown_A \boxtimes \triangledown_B : \Psi^{A \times B} \cong \left(\Psi^A\right)^B \xrightarrow{\triangledown_A^B} \Psi^B \xrightarrow{\triangledown_B} \Psi, \tag{130}$$

$$\triangle_A \boxtimes \triangle_B : \Psi^{A \times B} \cong \left(\Psi^A\right)^B \xrightarrow{\triangle_A^B} \Psi^B \xrightarrow{\triangle_B} \Psi. \tag{131}$$

**Proposition 15.** *The product $d_A \boxtimes d_B$ of strict premetrics given in Definition 30 is again a strict premetric.*

*Proof.* This is a result of Theorem 11. Consider the following diagram:

$$
\begin{array}{ccccccc}
A \times B & \xrightarrow{\mathrm{id}_{A \times B}} & A \times B & \longrightarrow & 1 & \longrightarrow & 1 \\
{\scriptstyle \Delta_{A \times B}} \downarrow & \lrcorner & {\scriptstyle \Delta_A \times \Delta_B} \downarrow & \lrcorner & {\scriptstyle \langle o, o \rangle} \downarrow & \lrcorner & \downarrow {\scriptstyle o} \\
(A \times B)^2 & \xrightarrow[\cong]{} & A^2 \times B^2 & \xrightarrow[d_A \times d_B]{} & \Psi \times \Psi & \xrightarrow[\otimes]{} & \Psi
\end{array}
\tag{132}
$$

$\Delta_A \times \Delta_B$ is a pullback of $o$ along $\otimes \circ (d_A \times d_B)$ because $\langle o, o \rangle$ is a pullback of $o$ along $\otimes$ according to Theorem 11, $\Delta_A \times \Delta_B$ is a pullback of $\langle o, o \rangle$ along $d_A \times d_B$, and we can apply the pullback lemma. $\Delta_A \times \Delta_B$ is isomorphic to $\Delta_{A \times B}$, which means that $\otimes \circ (d_A \times d_B)$ is a strict premetric. $\square$

Based on this definition, we can show that

$$
(d_A \boxtimes d_B) \boxtimes d_C \cong d_A \boxtimes (d_B \boxtimes d_C),
\tag{133}
$$

because $\times$ and $\otimes$ are associative up to isomorphism.

Further, we have

$$
(\triangledown_A \boxtimes \triangledown_B) \boxtimes \triangledown_C \cong \triangledown_A \boxtimes (\triangledown_B \boxtimes \triangledown_C),
\tag{134}
$$

$$
(\triangle_A \boxtimes \triangle_B) \boxtimes \triangle_C \cong \triangle_A \boxtimes (\triangle_B \boxtimes \triangle_C),
\tag{135}
$$

because composition is associative.

The singleton $(\{*\}, o_{\{*\} \times \{*\}}, \mathrm{id}_{\{*\}}, \mathrm{id}_{\{*\}})$ is the unit of the bifunctor $\boxtimes$. In this way, $(\Psi\text{-}\mathbf{Set}, \boxtimes, \{*\})$ forms a monoidal category.

However, note that the bifunctor $\boxtimes$ is not symmetric because $\triangledown_A \boxtimes \triangledown_B$ and $\triangle_A \boxtimes \triangle_B$ are not necessarily symmetric, and $d_A \boxtimes d_B \cong d_B \boxtimes d_A$ if and only if the monoidal product $\otimes$ of the quantale $\Psi$ is commutative.

Since $\Psi\text{-}\mathbf{Set}$ is a monoidal category, we can consider a strict monoidal functor $F_\Psi : \mathbf{Set} \to \Psi\text{-}\mathbf{Set}$, which equips products of sets with product strict premetrics and product aggregators:

$$
d_{A \times B} := d_A \boxtimes d_B,
\tag{136}
$$

$$
\triangledown_{A \times B} := \triangledown_A \boxtimes \triangledown_B,
\tag{137}
$$

$$
\triangle_{A \times B} := \triangle_A \boxtimes \triangle_B.
\tag{138}
$$

Such a monoidal functor induces a functor from a ($\mathbf{Set}$-enriched) category to a $\Psi\text{-}\mathbf{Set}$-enriched category, called the base change of enriching category. This means that we have a systematic way to equip a set $[A, B]$ of morphisms with a strict premetric

$$
d_{[A,B]} : [A, B] \times [A, B] \to \Psi,
\tag{139}
$$

a universal aggregator

$$
\triangledown_{[A,B]} : \Psi^{[A,B]} \to \Psi,
\tag{140}
$$

and an existential aggregator

$$
\triangle_{[A,B]} : \Psi^{[A,B]} \to \Psi,
\tag{141}
$$

which is compatible with the product.

Then, Theorem 1 is a special case of this enrichment. The relationship between predicates and quantities follows from Theorems 10 and 14.

## B.12 Summary

Finally, to accommodate readers without a background in category theory, we present the instantiated definitions and theoretical results free of categorical terminology. The non-categorical proofs are omitted. We will be using the following functions:

- the *zero predicate* $\zeta : [0, \infty] \to \{\top, \bot\} := x \mapsto (x = 0)$,
- the *product* $\zeta^n : [0, \infty]^n \to \{\top, \bot\}^n : (q_1, \ldots, q_n) \mapsto (q_1 = 0, \ldots, q_n = 0)$, and
- the *postcomposition* $\zeta^A : [0, \infty]^A \to \{\top, \bot\}^A := q \mapsto \zeta \circ q$ of the zero predicate.

**Definition 31** (Quantity). A quantity $q : A \to [0, \infty]$ is *homomorphic* to a predicate $p : A \to \{\top, \bot\}$ if $p = \zeta \circ q$:
$$\forall a \in A. \, (q(a) = 0) \leftrightarrow p(a). \tag{142}$$
A quantity $q$ is *subhomomorphic* to a predicate $p$ if $\zeta \circ q \to p$:
$$\forall a \in A. \, (q(a) = 0) \to p(a). \tag{143}$$

**Example 20.** A *strict premetric* $d_A : A \times A \to [0, \infty]$ is a quantity on the product set $A \times A$ homomorphic to the equality predicate $=_A : A \times A \to \{\top, \bot\}$.

**Definition 32** (Quantitative operation). Let $n \in \mathbb{N}$ be a natural number. A quantitative operation $\alpha : [0, \infty]^n \to [0, \infty]$ is *homomorphic* to a logical operation $\beta : \{\top, \bot\}^n \to \{\top, \bot\}$ via the zero predicate $\zeta$ if $\zeta \circ \alpha = \beta \circ \zeta^n$:
$$\forall (q_1, \ldots, q_n) \in [0, \infty]^n. \, (\alpha(q_1, \ldots, q_n) = 0) \leftrightarrow \beta(q_1 = 0, \ldots, q_n = 0). \tag{144}$$
A quantitative operation $\alpha$ is *subhomomorphic* to a logical operation $\beta$ via the zero predicate if $\zeta \circ \alpha \to \beta \circ \zeta^n$:
$$\forall (q_1, \ldots, q_n) \in [0, \infty]^n. \, (\alpha(q_1, \ldots, q_n) = 0) \to \beta(q_1 = 0, \ldots, q_n = 0). \tag{145}$$

The relationship between quantitative operations and logical operations is as follows (Theorem 10):

**Proposition 16.** *Let $n \in \mathbb{N}$ be a natural number. For $i \in \{1, \ldots, n\}$, let $p_i : A \to \{\top, \bot\}$ be a predicate, and let $q_i : A \to [0, \infty]$ be a quantity. Let $p : A \to \{\top, \bot\}^n := \langle p_1, \ldots, p_n \rangle$ be the tupling of the predicates, and let $q : A \to [0, \infty]^n := \langle q_1, \ldots, q_n \rangle$ be the tupling of the quantities. Let $\alpha : [0, \infty]^n \to [0, \infty]$ be a quantitative operation, and let $\beta : \{\top, \bot\}^n \to \{\top, \bot\}$ be a logical operation. Assume that for all $i \in \{1, \ldots, n\}$, $q_i$ is homomorphic to $p_i$. Then,*

- *if $\alpha$ is homomorphic to $\beta$, $\alpha \circ q$ homomorphic to $\beta \circ p$; and*
- *if $\alpha$ is subhomomorphic to $\beta$, $\alpha \circ q$ subhomomorphic to $\beta \circ p$.*

In fact, we can also show that if for all $i \in \{1, \ldots, n\}$, $q_i$ is subhomomorphic to $p_i$, and $\alpha$ is subhomomorphic to $\beta$, then $\alpha \circ q$ is subhomomorphic to $\beta \circ p$.

**Definition 33** (Aggregator). An aggregator $\alpha_A : [0, \infty]^A \to [0, \infty]$ is *homomorphic* to a quantifier $\beta_A : \{\top, \bot\}^A \to \{\top, \bot\}$ via the zero predicate $\zeta$ if $\zeta \circ \alpha_A = \beta_A \circ \zeta^A$:
$$\forall q \in [0, \infty]^A. \, \left( \left( \underset{a \in A}{\alpha} q(a) \right) = 0 \right) \leftrightarrow \left( \underset{a \in A}{\beta} (q(a) = 0) \right). \tag{146}$$

**Example 21.** A *universal aggregator* $\nabla_A : [0, \infty]^A \to [0, \infty]$ is a function such that
$$\forall q \in [0, \infty]^A. \, \left( \left( \underset{a \in A}{\nabla} q(a) \right) = 0 \right) \leftrightarrow (\forall a \in A. \, q(a) = 0). \tag{147}$$

**Example 22.** An *existential aggregator* $\triangle_A : [0, \infty]^A \to [0, \infty]$ is a function such that
$$\forall q \in [0, \infty]^A. \, \left( \left( \underset{a \in A}{\triangle} q(a) \right) = 0 \right) \leftrightarrow (\exists a \in A. \, q(a) = 0). \tag{148}$$

The relationship between aggregators and quantifiers is as follows (Theorem 14):

**Proposition 17.** *Let $p : A \times B \to \{\top, \bot\}$ be a predicate, and let $q : A \times B \to [0, \infty]$ be a quantity. Let $\widehat{p} : B \to \{\top, \bot\}^A$ and $\widehat{q} : B \to [0, \infty]^A$ be the exponential transposes of $p$ and $q$. Let $\alpha_A : [0, \infty]^A \to [0, \infty]$ be an aggregator, and let $\beta_A : \{\top, \bot\}^A \to \{\top, \bot\}$ be a quantifier. Then, if $q$ is homomorphic to $p$, and $\alpha$ is homomorphic to $\beta$, then $\alpha_A \circ \widehat{q}$ is homomorphic to $\beta_A \circ \widehat{p}$.*

We have a compositional way to assign a strict premetric and an aggregator to a product of sets:

**Proposition 18.** *Let $d_A : A \times A \to [0, \infty]$ and $d_B : B \times B \to [0, \infty]$ be strict premetrics. Then,*
$$d_{A \times B} : (A \times B) \times (A \times B) \to [0, \infty] := ((a, b), (a', b')) \mapsto d(a, a') + d(b, b') \tag{149}$$
*is a strict premetric on the product set $A \times B$.*

**Proposition 19.** *Let $\nabla_A : [0, \infty]^A \to [0, \infty]$ and $\nabla_B : [0, \infty]^B \to [0, \infty]$ be universal aggregators. Then,*
$$\underset{A \times B}{\nabla} : [0, \infty]^{A \times B} \to [0, \infty] := q \mapsto \underset{b \in B}{\nabla} \, \underset{a \in A}{\nabla} \, q(a, b) \tag{150}$$
*is a universal aggregator on the product set $A \times B$.*

# C Proofs

## C.1 Proposition 2

*Proof.*

$$q_{\text{product}}(m : Y \to Z) \tag{151}$$

$$= \inf_{m_{1,1} \in [Y_1, Z_1]} \inf_{m_{2,2} \in [Y_2, Z_2]} d_{[Y,Z]}(m, m_{1,1} \times m_{2,2}) \tag{152}$$

$$= \inf_{m_{1,1} \in [Y_1, Z_1]} \inf_{m_{2,2} \in [Y_2, Z_2]} (d_{[Y,Z_1]}(m_1, m_{1,1} \circ p_1) + d_{[Y,Z_2]}(m_2, m_{2,2} \circ p_2)) \tag{153}$$

$$= \inf_{m_{1,1} \in [Y_1, Z_1]} d_{[Y,Z_1]}(m_1, m_{1,1} \circ p_1) + \inf_{m_{2,2} \in [Y_2, Z_2]} d_{[Y,Z_2]}(m_2, m_{2,2} \circ p_2) \tag{154}$$

$$= \inf_{m_{1,1} \in [Y_1, Z_1]} \mathop{\triangledown}_{y \in Y} d_{Z_1}(m_1(y), m_{1,1}(y_1)) + \inf_{m_{2,2} \in [Y_2, Z_2]} \mathop{\triangledown}_{y \in Y} d_{Z_2}(m_2(y), m_{2,2}(y_2)) \tag{155}$$

$$= \inf_{m_{1,1} \in [Y_1, Z_1]} \mathop{\triangledown}_{y_1 \in Y_1} \mathop{\triangledown}_{y_2 \in Y_2} d_{Z_1}(m_1(y_1, y_2), m_{1,1}(y_1))$$

$$+ \inf_{m_{2,2} \in [Y_2, Z_2]} \mathop{\triangledown}_{y_2 \in Y_2} \mathop{\triangledown}_{y_1 \in Y_1} d_{Z_2}(m_2(y_1, y_2), m_{2,2}(y_2)) \tag{156}$$

$$= \mathop{\triangledown}_{y_1 \in Y_1} \mathop{\triangledown}_{y_2 \in Y_2} d_{Z_1}(m_1(y_1, y_2), m_{1,1}^*(y_1))$$

$$+ \mathop{\triangledown}_{y_2 \in Y_2} \mathop{\triangledown}_{y_1 \in Y_1} d_{Z_2}(m_2(y_1, y_2), m_{2,2}^*(y_2)), \tag{157}$$

where

$$m_{1,1}^* := \mathop{\arg\inf}_{m_{1,1} \in [Y_1, Z_1]} \mathop{\triangledown}_{y_1 \in Y_1} \mathop{\triangledown}_{y_2 \in Y_2} d_{Z_1}(m_1(y_1, y_2), m_{1,1}(y_1)) \tag{158}$$

$$= y_1 \mapsto \mathop{\arg\inf}_{z_1 \in Z_1} \mathop{\triangledown}_{y_2 \in Y_2} d_{Z_1}(m_1(y_1, y_2), z_1), \tag{159}$$

$$m_{2,2}^* := \mathop{\arg\inf}_{m_{2,2} \in [Y_2, Z_2]} \mathop{\triangledown}_{y_2 \in Y_2} \mathop{\triangledown}_{y_1 \in Y_1} d_{Z_2}(m_2(y_1, y_2), m_{2,2}(y_2)) \tag{160}$$

$$= y_2 \mapsto \mathop{\arg\inf}_{z_2 \in Z_2} \mathop{\triangledown}_{y_1 \in Y_1} d_{Z_2}(m_2(y_1, y_2), z_2). \tag{161}$$

$$\square$$

## C.2 Proposition 3

*Proof.*

$$q_{\text{const-curry}}(m : Y \to Z) \tag{162}$$

$$= q_{\text{const}}(\widehat{m_1}) + q_{\text{const}}(\widehat{m_2}) \tag{163}$$

$$= \mathop{\triangledown}_{y_2 \in Y_2} \mathop{\triangledown}_{y_2' \in Y_2} d_{[Y_1, Z_1]}(\widehat{m_1}(y_2), \widehat{m_1}(y_2')) + \mathop{\triangledown}_{y_1 \in Y_1} \mathop{\triangledown}_{y_1' \in Y_1} d_{[Y_2, Z_2]}(\widehat{m_2}(y_1), \widehat{m_2}(y_1')) \tag{164}$$

$$= \mathop{\triangledown}_{y_2 \in Y_2} \mathop{\triangledown}_{y_2' \in Y_2} \mathop{\triangledown}_{y_1 \in Y_1} d_{Z_1}(\widehat{m_1}(y_2)(y_1), \widehat{m_1}(y_2')(y_1))$$

$$+ \mathop{\triangledown}_{y_1 \in Y_1} \mathop{\triangledown}_{y_1' \in Y_1} \mathop{\triangledown}_{y_2 \in Y_2} d_{Z_2}(\widehat{m_2}(y_1)(y_2), \widehat{m_2}(y_1')(y_2)) \tag{165}$$

$$= \mathop{\triangledown}_{y_2 \in Y_2} \mathop{\triangledown}_{y_2' \in Y_2} \mathop{\triangledown}_{y_1 \in Y_1} d_{Z_1}(m_1(y_1, y_2), m_1(y_1, y_2'))$$

$$+ \mathop{\triangledown}_{y_1 \in Y_1} \mathop{\triangledown}_{y_1' \in Y_1} \mathop{\triangledown}_{y_2 \in Y_2} d_{Z_2}(m_2(y_1, y_2), m_2(y_1', y_2)) \tag{166}$$

$$= \mathop{\triangledown}_{y_1 \in Y_1} \mathop{\triangledown}_{y_2 \in Y_2} \mathop{\triangledown}_{y_2' \in Y_2} d_{Z_1}(m_1(y_1, y_2), m_1(y_1, y_2'))$$

$$+ \mathop{\triangledown}_{y_2 \in Y_2} \mathop{\triangledown}_{y_1 \in Y_1} \mathop{\triangledown}_{y_1' \in Y_1} d_{Z_2}(m_2(y_1, y_2), m_2(y_1', y_2)). \tag{167}$$

$$\square$$

# D  Discussions

## D.1  Background

Defining and measuring the properties of learning models is a core topic in machine learning, especially representation learning [Bengio et al., 2013]. A proper comprehension of what constitutes good representations and how to assess their quality is important for developing suitable learning objectives and evaluation metrics. To define these properties, many important concepts are given by *equational predicates*, such as *independence* of random variables, extensively used in statistical learning and causal learning [Hyvärinen and Oja, 2000, Koller and Friedman, 2009, Schölkopf and von Kügelgen, 2022], and *equivariance* of learning models, reflecting the symmetries and structures of the data [Cohen and Welling, 2016, Zaheer et al., 2017, Higgins et al., 2018, Maron et al., 2019, de Haan et al., 2020, Cohen, 2021, van der Pol et al., 2022, Navon et al., 2023].

Considerable efforts have been put into designing model architectures that perfectly satisfy specific properties, such as *monotonicity* [Sill, 1997, Daniels and Velikova, 2010], *invertibility* [Rezende and Mohamed, 2015, Behrmann et al., 2019, Ishikawa et al., 2023], *convexity* [Amos et al., 2017], and *equivariance* [Lee et al., 2019, Brehmer et al., 2023]. However, hard-coding multiple properties into a model by design could be challenging [Köhler et al., 2020]. Hence, it is desirable to devise quantitative metrics to directly measure these properties, even if the models do not have the properties built-in [Goodfellow et al., 2009, Chen et al., 2020, Kvinge et al., 2022]. Ideally, these metrics should be easily computable or even differentiable, allowing us to directly optimize the properties.

*Disentangled representation learning* [Bengio et al., 2013], our main focus of this paper, is such a field where defining and measuring the desired properties are not straightforward tasks [Carbonneau et al., 2022, Zhang and Sugiyama, 2023]. It has been suggested that disentangling the underlying explanatory factors in complex data is a promising approach for reliable, interpretable, generalizable, and data-efficient representation learning [Locatello et al., 2019a,b, Montero et al., 2021, Dittadi et al., 2021, Xu et al., 2022]. However, in contrast to the wealth of results regarding invariant and equivariant layers, the exploration of designing a "*disentangled layer*" has been relatively limited. One reason is that disentanglement was not considered a singular property but rather a combination of several requirements. The absence of a clear definition and appropriate metrics for disentanglement has created a gap between the learning objectives and evaluation metrics. A new evaluation metric is often introduced along with a new representation learning method [Carbonneau et al., 2022], but it is usually unproven that the method can optimize the new metric, and the metric truly quantifies the alleged property [Higgins et al., 2017, Kim and Mnih, 2018, Chen et al., 2018, Li et al., 2020].

To formally define disentanglement, a line of research utilized group theory and representation theory [Cohen and Welling, 2014, 2015, Higgins et al., 2018], with a focus on the *direct product of groups*. Thanks to the rich algebraic structure, it becomes possible to derive various model architectures and learning objectives from the equational requirements of the product and equivariance [Caselles-Dupré et al., 2019, Pfau et al., 2020, Quessard et al., 2020, Painter et al., 2020, Miyato et al., 2022, Yang et al., 2022, Tonnaer et al., 2022, Keurti et al., 2023]. Another approach adopted a topological perspective, using concepts such as the *product manifold* to define disentanglement [Zhou et al., 2020, Fumero et al., 2021, Zhang et al., 2021, Balabin et al., 2024]. However, theoretically comparing different approaches has been a challenging task.

To quantitatively measure disentanglement, Ridgeway and Mozer [2018] proposed three concepts called *modularity, compactness, and explicitness*, which were defined verbally but not mathematically. Eastwood and Williams [2018] proposed similar three criteria called *disentanglement, completeness, and informativeness* and corresponding evaluation metrics. However, it was unclear what properties these metrics truly quantify. Additionally, due to the necessity for additional training of classifiers along with hyperparameter tuning and the involvement of non-differentiable regressors such as the random forest [Breiman et al., 1984], it is impossible to directly optimize these metrics using gradient-based optimization. Recently, Eastwood et al. [2023] extended this framework with two new metrics called *explicitness/ease-of-use and size* based on the functional capacity. Do and Tran [2020] introduced metrics for *informativeness, separability, independence, and interpretability* from an information-theoretic perspective, while Tokui and Sato [2022] introduced a new metric in terms of *uniqueness, redundancy, and synergy* based on partial information decomposition. These metrics have been mainly used during the evaluation stage, after a model is trained with other learning objectives.

## D.2 Related work

**Equivariance** The work by Kvinge et al. [2022] might be the closest to our approach in spirit. They directly converted *equivariance*, an equational predicate, to a quantitative metric and analyzed their relationship (Proposition 3.2). In contrast, based on our proposed conversion method, we can use the following definition and metric:

**Definition 34** (Equivariant function). Let $A$, $B$, and $C$ be sets. A function $f : A \to B$ is *equivariant* to actions (any binary functions) $\cdot_A : C \times A \to A$ and $\cdot_B : C \times B \to B$ if

$$p_{\text{equivariant}}(f : A \to B) := \forall c \in C.\ f \circ (c \cdot_A -) =_{[A,B]} (c \cdot_B -) \circ f \tag{168}$$

$$= \forall c \in C.\ \forall a \in A.\ f(c \cdot_A a) =_B c \cdot_B f(a), \tag{169}$$

which can be measured by

$$q_{\text{equivariant}}(f : A \to B) := \bigtriangledown_{c \in C} \bigtriangledown_{a \in A} d_B(f(c \cdot_A a), c \cdot_B f(a)). \tag{170}$$

**Calibration** More broadly, the study of the relationship between different metrics in statistical learning is called *calibration analysis* [Steinwart, 2007, Reid and Williamson, 2010, Ni et al., 2019, Bao and Sugiyama, 2020, Bao et al., 2020]. Our work can be seen as an extension of the concept of the calibration to a wider range of properties defined by equational predicates.

**Disentanglement metric** In disentangled representation learning, metrics similar to Eq. (26) have been proposed by Higgins et al. [2017], Kim and Mnih [2018]. Their metrics also fix one factor and vary all others and calculate some constancy metrics (the mean pairwise distance in Higgins et al. [2017] and the variance in Kim and Mnih [2018]). However, both studies took an indirect approach, involving the training of a classifier to predict the fixed factor. Consequently, the resulting metrics are not differentiable anymore and entangle modularity and informativeness. In this work, we argue that it is better to measure these two properties separately.

**Weakly supervised disentanglement** Ridgeway and Mozer [2018] proposed and investigated the similarity supervision and argued that such supervision is easy to obtain via crowdsourcing. Shu et al. [2020] further studied this type of supervision based on distribution matching and referred it as match pairing. Other weaker forms of supervision were also investigated, such as the number of changed factors [Locatello et al., 2020] or paired data with unknown intervention [Brehmer et al., 2022]. Given that our theory can establish connections between logical definitions and quantitative metrics, it holds promise for deriving disentanglement metrics for various types of weak supervision based on logical inference.

**Multi-valued logic** Aristotelian logic assumes that every proposition is either true or false, adhering to the *principle of bivalence*. The law of Aristotelian logic can be algebraically represented on the set $\{0, 1\}$ of binary truth values [Boole, 1854], known as the two-element *Boolean algebra*. The exploration of non-Aristotelian logic involves investigating logical systems that relax or modify this strict binary valuation, allowing for a broader range of truth values and accommodating various forms of uncertainty, vagueness, or context-dependence in reasoning [Hájek, 1998, Malinowski, 2007, Bergmann, 2008].

The mathematical study of multi-valued logic can date back to the seminal work by Łukasiewicz in 1920, who introduced a third truth value interpreted as "*possibility*" and symbolized by $\frac{1}{2}$. Łukasiewicz [1920] examined several principles in this *three-valued logic* such as the principles of identity, implication, syllogism, and contradiction, and discussed its theoretical and practical importance in indeterministic philosophy and deductive sciences. Later, Łukasiewicz and Tarski [1930] proposed *propositional calculus*, a theory of propositions with values from the real interval $[0, 1]$, which is now also commonly known as *Łukasiewicz logic*. Łukasiewicz logic involves new continuous logical connectives such as strong/weak conjunction and disjunction.

Furthermore, Chang [1958] studied the algebraic systems for *many-valued logic*, called *MV-algebras*. Chang and Keisler [1966] then proposed *continuous model theory*, also referred to as *compact-valued logic* [Ben Yaacov, 2022], where the truth values can be in arbitrary compact Hausdorff spaces and a wide variety of quantifiers was studied. Later, Ben Yaacov et al. [2008] studied model theory for metric structures and proposed *(real-valued) continuous first-order logic* [Ben Yaacov and Usvyatsov,

2010], where the space of truth values is a closed, bounded interval of real numbers with the order topology (e.g., $[0, 1]$), and suggested that we only need two canonical quantifiers $\sup$ and $\inf$. From a categorical perspective, Cho [2020] developed categorical semantics of metric spaces and continuous logic by introducing the notion of *continuous subobject classifier*, and Figueroa and van den Berg [2022] studied a topos of continuous logic using the notion of *hyperdoctrine*.

On the other hand, from a categorical perspective, Lawvere [1973] showed that a generalized metric space, also known as a *Lawvere metric space*, is a category enriched over what is now commonly called the *Lawvere quantale* $([0, \infty], \geq, +, 0)$, i.e., the set $[0, \infty]$ of extended non-negative real numbers equipped with addition $+$ as a (semicartesian) monoidal product and truncated subtraction $\dot{-}$ as the internal hom. In other words, a Lawvere metric space is a set $A$ equipped with a function $d : A \times A \to [0, \infty]$ such that for all $a \in A$, we have $0 \geq d(a, a)$ or $d(a, a) = 0$ (identity), which makes $d$ a *premetric*, and for all $a, b, c \in A$, we have $d(b, c) + d(a, b) \geq d(a, c)$ (composition), which means that $d$ satisfies the *triangle inequality*.

Recently, Mardare et al. [2016] took an equational approach to *quantitative algebraic reasoning*, which was later also referred to as *quantitative equational logic* [Mardare et al., 2021], by introducing approximate equality predicates $=_\varepsilon$ indexed by rational numbers $\varepsilon$ (i.e., $a =_\varepsilon b$ if $a$ and $b$ are at most $\varepsilon$ apart), and suggested that this approach essentially involves working with *enriched Lawvere theory*. Dagnino and Pasquali [2022] provided a logical ground to quantitative reasoning in the categorical language of *Lawvere's doctrines* by viewing distances as equality predicates in *linear logic*. Bacci et al. [2023] further studied the natural deduction systems of propositional logics for the Lawvere quantale and introduced what was later called *affine Lawvere logic*, including Łukasiewicz logic and Ben Yaacov's continuous propositional logic. Bacci et al. [2024] extended affine Lawvere logic to *polynomial Lawvere logic* by allowing multiplication as an extra logical connective. These studies are on propositional logic and do not involve predicates and quantifiers. Recently, Capucci [2024] studied a spectrum of quantifiers in $[0, \infty]$-valued quantitative predicate logic.

In the context of machine learning, these relatively recently developed logics have yet to prove their practical importance. While these innovative approaches often hold theoretical promise, they need to demonstrate tangible benefits in real-world applications. Key areas where these new logics might eventually make an impact include neuro-symbolic reasoning and logic/probabilistic programming [d'Avila Garcez et al., 2002, Manhaeve et al., 2018, Sen et al., 2022, Badreddine et al., 2022, Fagin et al., 2024], which hold promise for integrating low-level perception with high-level reasoning, improving model interpretability, enhancing training efficiency, and enabling more robust decision-making processes. However, widespread adoption and validation through practical use cases are necessary to establish their true value and effectiveness in the machine learning landscape.

Under this background, let us contextualize our proposed methodology for deriving $[0, \infty]$-valued quantitative metrics from logical definitions. We highlight three characteristics of our framework:

- We allowed not only metrics, but *strict premetrics*, such as the relative entropy (Kullback–Leibler divergence) [Kullback and Leibler, 1951, Perrone, 2023] widely used in machine learning, as the real-valued counterparts for the equality predicates;
- We focused on whether the metrics are *zero or not* and the (differentiable) optimization of the derived metrics, because our main goal is to guarantee that the minimizers of the derived metrics satisfy the predicate;
- We included a wide range of real-valued quantifiers (or *aggregators* in our terms) beyond $\sup$ and $\inf$, such as mean and mean square, as long as they are homomorphic to the two-valued quantifiers, because the derived metrics may have nicer properties or even analytical solutions.

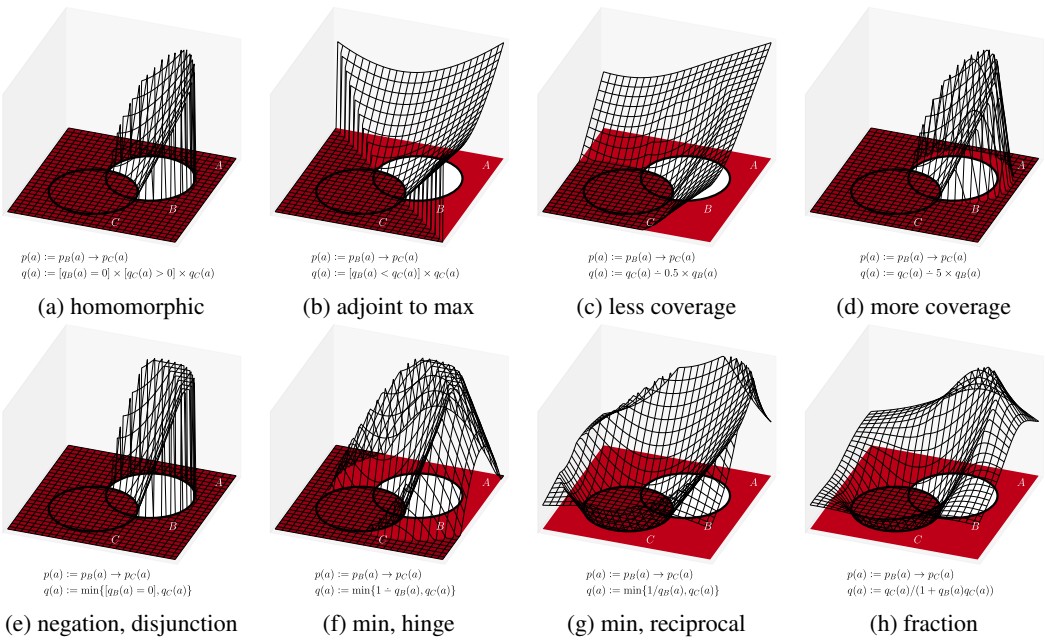

| (a) homomorphic | (b) adjoint to max | (c) less coverage | (d) more coverage |
| --- | --- | --- | --- |

| (e) negation, disjunction | (f) min, hinge | (g) min, reciprocal | (h) fraction |
| --- | --- | --- | --- |

Figure 8: Quantitative operations for implication

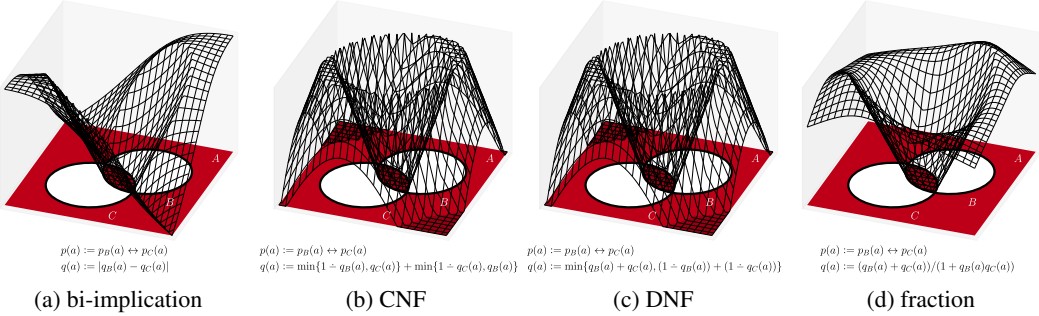

| (a) bi-implication | (b) CNF | (c) DNF | (d) fraction |
| --- | --- | --- | --- |

Figure 9: Quantitative operations for equivalence

### D.3 Implication and equivalence

In Section 3, we only used the truncated subtraction $\dot{-}$ as a quantitative operation for the implication $\rightarrow$ (Table 1). In Theorem 1, we noted that the implication is special because if a predicate involves the implication, then not all elements satisfying the predicate minimize the corresponding quantities (Fig. 2). In Appendix B.8, we discussed other possible quantitative operations $\multimap$ corresponding to the implication $\rightarrow$.

In Fig. 8, we showed eight alternative quantitative operations for the implication. In the first row, the first one is homomorphic to the implication, which means that its minimizers are exactly those that satisfy the predicate (Example 16); the second one is right adjoint to the max (Example 17); and the other two are variants of the truncated subtraction (Example 18), which have more or less coverage. In the second row, we used the logical equivalence between $a \rightarrow b$ and $\neg a \vee b$ to define quantitative operations for the implication using quantitative operations for the negation and disjunction. For example, we can use the hinge function $1 \dot{-} n$ and $\min$, which lead to $\min\{1 \dot{-} a, b\}$, or the reciprocal function $\frac{1}{n}$ and $\frac{ab}{a+b}$, which lead to $\frac{\frac{1}{a}b}{\frac{1}{a}+b} = \frac{b}{1+ab}$.

Similarly, we can use logically equivalent expressions of the logical equivalence $a \leftrightarrow b$, such as $(a \rightarrow b) \wedge (b \rightarrow a)$ (bi-implication), $(\neg a \vee b) \wedge (\neg b \vee a)$ (conjunctive normal form (CNF)), and $(a \wedge b) \vee (\neg a \wedge \neg b)$ (disjunctive normal form (DNF)), to derive quantitative operations for the equivalence, shown in Fig. 9.

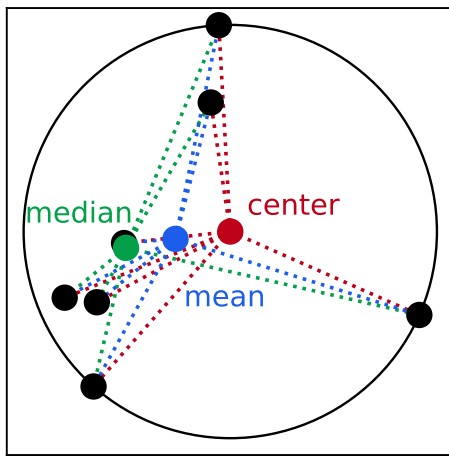
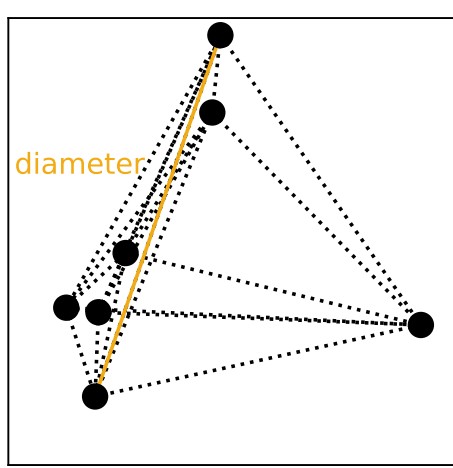

(a) central point

(b) pairwise distance

Figure 10: Two approaches for measuring the *constancy* of a set in $\mathbb{R}^2$: (a) finding a central point, such as the center of the smallest bounding sphere, the geometric median, or the mean, and then measuring the dispersion around this point; and (b) aggregating pairwise distances between points.

Note that a quantitative operation homomorphic to the implication cannot be continuous everywhere, which is undesirable for gradient-based optimization. For example, the following quantity also measures the injectivity of a function $m : Y \to Z$:

$$\underset{y \in Y}{\bigtriangledown} \underset{y' \in Y}{\bigtriangledown} [d_Z(m(y), m(y')) = 0] \times [d_Y(y, y') > 0] \times d_Y(y, y') \tag{171}$$

$$= \underset{y \in Y}{\bigtriangledown} \underset{y' \in Y}{\bigtriangledown} [m(y) =_Z m(y')] \times [y \neq_Y y'] \times d_Y(y, y'). \tag{172}$$

This quantity aggregates distances between pairs of different inputs mapped to the same outputs. However, unlike $q_{\text{injective}}$ introduced in Section 4.3, it is not differentiable with respect to the function $m : Y \to Z$. Thus, we cannot use it to improve the injectivity of a function by gradient descent.

### D.4 Constant function

Note that there are two logically equivalent definitions of a constant function (to a non-empty set). One is based on the equality between all pairs, as in Definition 11. The other is based on the constant output value (see Fig. 10):

**Definition 35** (Constant function with value)**.** A function $f : A \to B$ is a *constant function* with value $b \in B$ if

$$p_{\text{const-v}}(f : A \to B) := \exists b \in B. \, \forall a \in A. \, (f(a) =_B b), \tag{173}$$

which can be measured by

$$q_{\text{const-v}}(f : A \to B) := \inf_{b \in B} \underset{a \in A}{\bigtriangledown} d_B(f(a), b). \tag{174}$$

This quantity $q_{\text{const-v}}$ finds a central point in the codomain that best approximates all the outputs of a function, which is similar to the approach we discussed in Section 4.1. In fact, we can prove that if we use $q_{\text{const-v}}$ in $q_{\text{const-curry}}$, we will end up with the same quantity $q_{\text{product}}$.

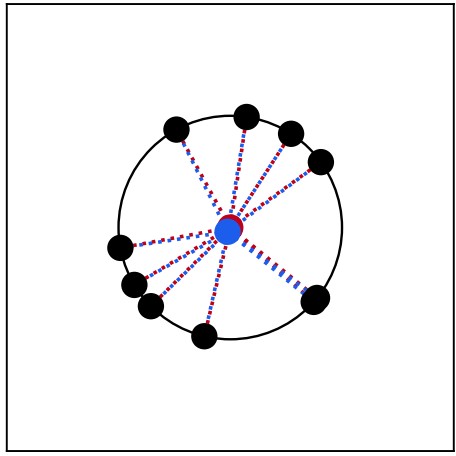 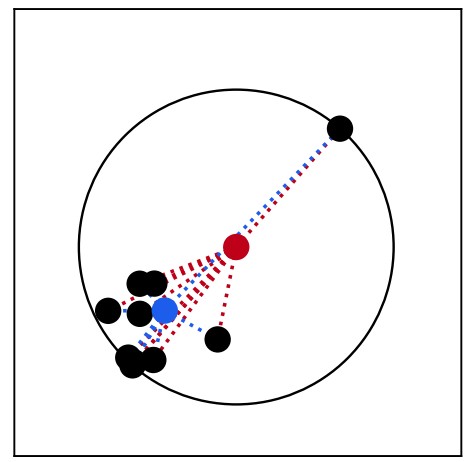

(a) small radius, large variance                    (b) large radius, small variance

Figure 11: Metrics may rank imperfect representations differently.

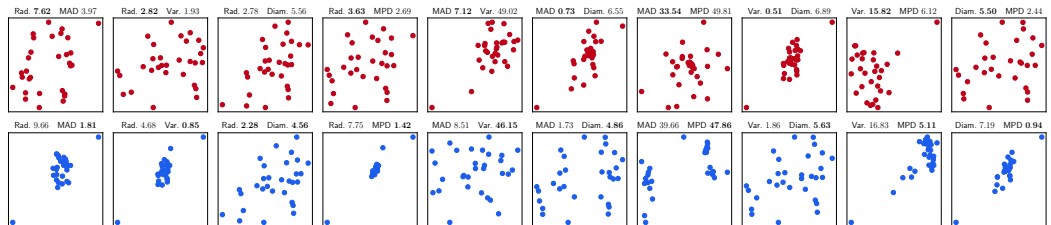

Figure 12: For a pair of constancy metrics (each column), we can find two sets of points in $\mathbb{R}^2$ ranked differently by these metrics, except for the radius and diameter, because for a subset $A_0$ in a set $A$, we have $\inf_{a_0 \in A} \sup_{a \in A_0} d_A(a_0, a) \leq \inf_{a_0 \in A_0} \sup_{a \in A_0} d_A(a_0, a) \leq \sup_{a_0 \in A_0} \sup_{a \in A_0} d_A(a_0, a)$.

## D.5    Rank of imperfect representations

It is worth noting that Theorem 1 only guarantees that the minimizers of different quantitative metrics derived from the same logical definition are the same, but imperfect representations, whose evaluation results are non-zero, may be ranked differently by different metrics.

For example, Fig. 11 illustrates two constancy metrics, the radius of the smallest bounding sphere and the variance, on two sets of points in $\mathbb{R}^2$, where one set has a small radius but a large variance, while the other has a large radius but a small variance. More examples are presented in Fig. 12, and such results can also be observed in Table 2. This difference can lead to differences in risk preferences, sensitivity to outliers, and learning dynamics when these metrics are used as learning objectives. Further investigation of the characteristics of these metrics for imperfect representations is left for future work.

### D.6 Implementation

Thanks to advanced indexing (e.g., NumPy [Harris et al., 2020] and PyTorch [Paszke et al., 2019]) and analytical solutions to some optimization problems (e.g., Eq. (19)), some of the proposed metrics can be easily implemented, even as Python one-liners.

For example, the following function implements a family of modularity metrics:

```python
def q_product(y: np.ndarray, z: np.ndarray, aggregate, deviation):
    return np.sum([aggregate([deviation(zi[yi == yv]) for yv in np.unique(yi)]) for yi, zi in
    zip(y, z)])
```

Here, `y` and `z` are NumPy arrays of shape `(factor, index)`; `aggregate` can be `max`, `mean`, or `sum`; `deviation` can be a function calculating the radius of the smallest bounding sphere,[12] mean absolute deviation around the geometric median,[13] variance, diameter, or mean pairwise distance. Please note, however, that the deviation function can be computationally expensive, depending on the dimension of the codes.

### D.7 Limitations

Lastly, we discuss several aspects that are not covered in this work and potential directions for future research.

**Function equality**  A collection of input-out pairs $\{(x_i, y_i)\}_{i=1}^n \in (X \times Y)^n$ may not define a *function* $g : X \to Y$ for two reasons: First, the set of all inputs $X_0 := \{x_i\}_{i=0}^n$ is unlikely to enumerate all possible inputs (i.e., $X_0 \subsetneq X$), especially when the cardinality of the domain $X$ is infinite (e.g., $\mathbb{R}$), so the data may only define a *partial function* $g : X \rightharpoonup Y$ or a function from a smaller domain $g_0 : X_0 \to Y$. Second, the inputs may not be distinct, e.g., when an input is given multiple labels by different annotators, so the data may define a *multi-valued function*. The extension from functions to relations or stochastic maps is an important future direction of our work.

**Partial combinations**  A more general issue is learning and evaluating disentangled representations given only a subset of all combinations of factors, which is common when dealing with a large number of factors [Träuble et al., 2021, Montero et al., 2021, 2022, Roth et al., 2023]. It is crucial to evaluate and justify whether a metric computed on partial combinations of factors is a reliable proxy for the performance of the model on unseen combinations.

**Unknown projections**  Another common scenario is when the extracted representation is not properly aligned with the underlying factors. For example, a model may extract a three-dimensional representation $z \in \mathbb{R}^3$ for two factors $y \in [0, 1]^2$, and it can project to $((z_1, z_2), z_3)$ or $(z_1, (z_2, z_3))$. How can we determine which is better, without enumerating all possible projections? Finding the optimal assignment [Mahon et al., 2023] and correcting a pre-trained model post hoc [Träuble et al., 2021] based on the proposed metrics are interesting future directions.

### D.8 Broader impact

This paper focuses on the theoretical aspects of disentangled representation learning, and we do not foresee any immediate negative societal consequences. However, we would acknowledge that disentanglement is closely related to data-efficiency and fairness, potentially sparking discussions on ethical considerations. Besides, the application of category theory may facilitate the transfer and integration of knowledge across disciplines, fostering closer connections between various fields of study, even beyond the machine learning community.

---

[12]https://github.com/marmakoide/miniball (MIT License) [Welzl, 1991]

[13]https://github.com/krishnap25/geom_median (GNU General Public License, Version 3 (GPLv3)) [Pillutla et al., 2022]

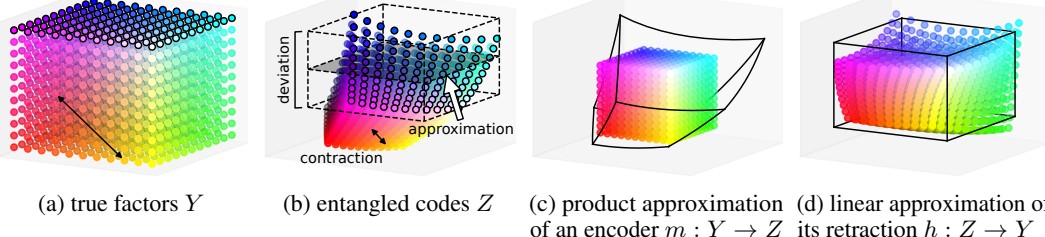

(a) true factors $Y$     (b) entangled codes $Z$     (c) product approximation     (d) linear approximation of
                                              of an encoder $m : Y \to Z$     its retraction $h : Z \to Y$

Figure 13: (a) a set of *factors* $Y$ represented by the RGB color model; (b) a set of entangled *codes* $Z$ extracted by an encoder $m : Y \to Z$; (c) a *product* function approximation; and (d) a linear approximation of the *retraction* $h : Z \to Y$ of the encoder.

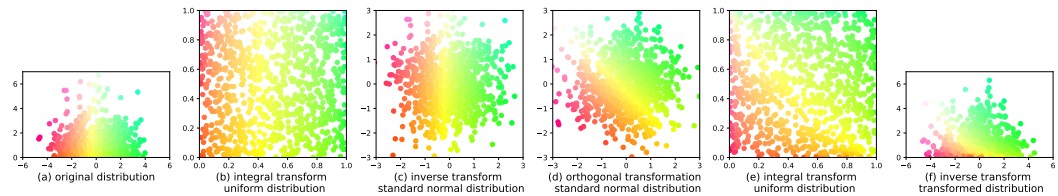

Figure 14: Entangling a multivariate distribution via probability integral transform, inverse transform, and orthogonal transformation of standard normal distribution [Locatello et al., 2019b, Theorem 1].

# E    Experiments

In this section, we provide the detailed data configuration used in Section 5 and further experimental results.

## E.1    Synthetic data

We used a simple synthetic setup to simulate entanglement of factors and common failures patterns. Concretely, we used a Cartesian product $Y := \{0, 0.1, \ldots, 1\}^3$ of three sets as the underlying factors (Fig. 13a). We used a random rotation matrix $R$ to entangle factors and componentwise exponential as a non-linear transformation. We composited this procedure twice and used an affine transformation to normalize the outputs (Fig. 13b). That is, we used the following function as the data generating process:

$$g : Y \to X : y \mapsto a \cdot \exp(R \cdot \exp(R \cdot y)) + b. \tag{175}$$

Note that this function is injective but not a product or linear.

We used the following functions as the function $m : Y \to Z$:

$$
\begin{array}{lll}
\text{entanglement} & (y_1, y_2, y_3) & \mapsto & g(y_1, y_2, y_3) \\
\text{rotation} & (y_1, y_2, y_3) & \mapsto & R \cdot (y_1, y_2, y_3) \\
\text{duplicate} & (y_1, y_2, y_3) & \mapsto & ((y_1, y_2, y_3), (y_1, y_2, y_3), y_3) \\
\text{complement} & (y_1, y_2, y_3) & \mapsto & ((y_2, y_3), (y_1, y_3), (y_1, y_2)) \\
\text{misalignment} & (y_1, y_2, y_3) & \mapsto & (y_2, y_3, y_1) \\
\text{redundancy} & (y_1, y_2, y_3) & \mapsto & ((y_1, -y_1), y_2, y_3) \\
\text{contraction} & (y_1, y_2, y_3) & \mapsto & 0.01 \times (y_1, y_2, y_3) \\
\text{nonlinear} & (y_1, y_2, y_3) & \mapsto & (y_1^2, y_2^2, y_3^2) \\
\text{constant} & (y_1, y_2, y_3) & \mapsto & (0, 0, 0)
\end{array}
$$

The rotation operation entangles factors but can be (linearly) inverted. The duplicate encoder has a modular decoder (projections), but itself is not modular. The redundancy encoder is both modular and informative, but not all codes can be decoded. The constant encoder is perfectly modular but not informative.

Table 3: Supervised modularity metrics

|  | Product approx. | | | Constancy | |
|  | Rad. | MAD | Var. | Diam. | MPD |
| --- | --- | --- | --- | --- | --- |
| entanglement | 0.44 | 0.75 | 0.96 | 0.19 | 0.82 |
| rotation | 0.22 | 0.51 | 0.80 | 0.05 | 0.64 |
| duplicate | 0.24 | 0.43 | 0.67 | 0.06 | 0.56 |
| complement | 0.12 | 0.28 | 0.55 | 0.01 | 0.42 |
| misalignment | 0.22 | 0.44 | 0.74 | 0.05 | 0.58 |
| random | 0.22 | 0.48 | 0.78 | 0.05 | 0.61 |

Table 4: Weakly supervised modularity metrics

|  | Product approx. | | | Constancy | |
|  | Rad. | MAD | Var. | Diam. | MPD |
| --- | --- | --- | --- | --- | --- |
| entanglement | 0.50 | 0.77 | 0.96 | 0.26 | 0.84 |
| rotation | 0.24 | 0.54 | 0.83 | 0.06 | 0.68 |
| duplicate | 0.28 | 0.46 | 0.71 | 0.07 | 0.60 |
| complement | 0.14 | 0.32 | 0.59 | 0.02 | 0.47 |
| misalignment | 0.22 | 0.48 | 0.77 | 0.05 | 0.62 |
| random | 0.24 | 0.52 | 0.81 | 0.06 | 0.64 |

## E.2 Weakly supervised modularity metrics

We briefly comment on the possibility of employing weak supervision for measuring disentangled representations.

For supervised disentanglement metrics we discussed in Section 4, the necessary data consists of observation-factor pairs $(x, y)$, representing a generator $g : Y \to X$. In order to evaluate an encoder $f : X \to Z$, we compose it with a generator and study the properties of the composition $m : Y \to Z := f \circ g$.

It is worth noting that $p_{\text{product}}$ and $p_{\text{injective}}$ are equational predicates, which means that they are *invariant to bijections*. Similarly, $q_{\text{product}}$ and $q_{\text{injective}}$ are *invariant to isometries*. This implies that the exact values of the factors are not important; we only need to know if two factors are equal or not. Hence, we only need weak supervision of the form $(x, x', y_i =_{Y_i} y_i')$ so that we can construct some equivalence classes of factors, and we can still calculate or approximate Eqs. (19) and (27). Note that this type of supervision has been partially investigated by Ridgeway and Mozer [2018], Shu et al. [2020].

For example, suppose we have an object A (e.g., red circle) and an object B (e.g., red triangle), and all we know is that objects A and B have the same color. Based on such weak information, we can still construct an equivalence class containing objects with the same color as object A. Then, we can regularize an encoder $f : X \to Z$ by minimizing the variance of the color representations over this equivalence class (Eq. (19)). In this way, the modularity of the encoder can be improved. The challenge arises when there is noise or only partial combinations, which is an interesting future work direction. In such cases, we may need to use semi-supervised clustering to group the data [Wagstaff et al., 2001, Basu et al., 2002, Bilenko et al., 2004].

To validate this idea, we conducted experiments where we only used a random sample of pairs and their similarities. Table 3 is an excerpt of Table 2, showing only the proposed modularity metrics, and Table 4 shows these metrics calculated using only similarity supervision. We reported the mean values of 10 random samples of pairs, and the variances are negligible. Comparing Tables 3 and 4, we can observe that weakly supervised metrics may overestimate imperfect representations, but they can still maintain the ranks. This observation suggests the potential utility of employing weak supervision for both learning and evaluating disentangled representations using the proposed metrics.

Table 5: Supervised disentanglement metrics on image datasets

| | Modularity | | | | | Informativeness | | | | | Existing metrics | | | | | |
| --- | --- | --- | --- | --- | --- | --- | --- | --- | --- | --- | --- | --- | --- | --- | --- | --- |
| | Product approx. | | | Constancy | | Retraction approx. | | | Contraction | | Pair | | Info. | Regressor | | |
| | Rad. | MAD | Var. | Diam. | MPD | ME | MAE | MSE | Max | Mean | Beta[a] | Factor[b] | MIG[c] | Dis.[d] | Com.[d] | Info.[d] |
| **3D Cars** [Reed et al., 2015] | | | | | | | | | | | | | | | | |
| VAE | 0.27 | 0.76 | 0.95 | 0.07 | 0.83 | 0.44 | 0.82 | 0.94 | 0.21 | 0.75 | 0.90 | 0.22 | 0.02 | 0.07 | 0.05 | 0.54 |
| $\beta$-VAE | 0.26 | 0.76 | 0.95 | 0.07 | 0.82 | 0.42 | 0.82 | 0.94 | 0.20 | 0.74 | 0.90 | 0.21 | 0.01 | 0.13 | 0.10 | 0.54 |
| FactorVAE | 0.24 | 0.75 | 0.95 | 0.06 | 0.82 | 0.35 | 0.82 | 0.94 | 0.21 | 0.74 | 0.89 | 0.20 | 0.03 | 0.11 | 0.08 | 0.54 |
| $\beta$-TCVAE | 0.28 | 0.77 | 0.95 | 0.08 | 0.83 | 0.43 | 0.82 | 0.94 | 0.21 | 0.74 | 0.90 | 0.21 | 0.02 | 0.14 | 0.11 | 0.59 |
| **dSprites** [Matthey et al., 2017] | | | | | | | | | | | | | | | | |
| VAE | 0.24 | 0.64 | 0.94 | 0.06 | 0.74 | 0.37 | 0.82 | 0.94 | 0.18 | 0.68 | 0.54 | 0.26 | 0.09 | 0.16 | 0.15 | 0.40 |
| $\beta$-VAE | 0.14 | 0.64 | 0.93 | 0.02 | 0.73 | 0.41 | 0.83 | 0.94 | 0.20 | 0.70 | 0.58 | 0.29 | 0.13 | 0.20 | 0.24 | 0.44 |
| FactorVAE | 0.18 | 0.63 | 0.93 | 0.03 | 0.73 | 0.38 | 0.82 | 0.94 | 0.17 | 0.67 | 0.48 | 0.26 | 0.13 | 0.20 | 0.23 | 0.36 |
| $\beta$-TCVAE | 0.22 | 0.64 | 0.93 | 0.05 | 0.73 | 0.42 | 0.83 | 0.94 | 0.21 | 0.70 | 0.56 | 0.27 | 0.18 | 0.29 | 0.31 | 0.59 |
| **3D Shapes** [Burgess and Kim, 2018] | | | | | | | | | | | | | | | | |
| VAE | 0.25 | 0.76 | 0.96 | 0.06 | 0.82 | 0.39 | 0.88 | 0.96 | 0.20 | 0.78 | 0.99 | 0.94 | 0.19 | 0.35 | 0.29 | 0.76 |
| $\beta$-VAE | 0.20 | 0.72 | 0.95 | 0.04 | 0.79 | 0.39 | 0.84 | 0.94 | 0.19 | 0.69 | 0.86 | 0.77 | 0.26 | 0.69 | 0.62 | 0.99 |
| FactorVAE | 0.24 | 0.69 | 0.96 | 0.06 | 0.78 | 0.34 | 0.82 | 0.93 | 0.16 | 0.64 | 0.83 | 0.57 | 0.15 | 0.40 | 0.40 | 0.84 |
| $\beta$-TCVAE | 0.25 | 0.69 | 0.94 | 0.06 | 0.77 | 0.32 | 0.82 | 0.93 | 0.18 | 0.63 | 0.76 | 0.50 | 0.11 | 0.68 | 0.57 | 0.98 |
| **MPI3D** [Gondal et al., 2019] | | | | | | | | | | | | | | | | |
| VAE | 0.04 | 0.56 | 0.91 | 0.00 | 0.66 | 0.30 | 0.76 | 0.89 | 0.12 | 0.46 | 0.48 | 0.11 | 0.12 | 0.29 | 0.33 | 0.64 |
| $\beta$-VAE | 0.02 | 0.84 | 0.97 | 0.00 | 0.87 | 0.28 | 0.75 | 0.89 | 0.10 | 0.41 | 0.39 | 0.11 | 0.07 | 0.15 | 0.18 | 0.47 |
| FactorVAE | 0.09 | 0.60 | 0.93 | 0.01 | 0.70 | 0.27 | 0.75 | 0.89 | 0.11 | 0.43 | 0.47 | 0.09 | 0.12 | 0.26 | 0.30 | 0.60 |
| $\beta$-TCVAE | 0.07 | 0.65 | 0.95 | 0.01 | 0.74 | 0.28 | 0.76 | 0.89 | 0.12 | 0.46 | 0.45 | 0.07 | 0.12 | 0.24 | 0.31 | 0.57 |

[a] [Higgins et al., 2017] [b] [Kim and Mnih, 2018] [c] [Chen et al., 2018] [d] [Eastwood and Williams, 2018]

## E.3 Evaluation of existing models on image datasets

We also report the results of several widely used unsupervised disentangled representation learning methods (VAE [Kingma and Welling, 2014], $\beta$-VAE [Higgins et al., 2017], FactorVAE [Kim and Mnih, 2018], and $\beta$-TCVAE [Chen et al., 2018]) evaluated on four image datasets (**3D Cars** [Reed et al., 2015], **dSprites** [Matthey et al., 2017], **3D Shapes** [Burgess and Kim, 2018], and **MPI3D** [Gondal et al., 2019]) in Table 5.

We used a public PyTorch implementation [Paszke et al., 2019] of these methods and used the same encoder/decoder architecture with the default hyperparameters described in Locatello et al. [2019b] for all methods for a fair comparison. We used linear projection to find the most informative representations for each factor. The experiments were conducted on a NVIDIA Tesla V100 GPU.

Before analyzing these results, it is important to note that the evaluation of these learning models is *not* meant to be a proof of the correctness of the proposed metrics, since we cannot tell whether a bad result is due to the insufficiency of a learning method, to the quality of the datasets, or to the problem of the evaluation, if we have no theoretical guarantee for the metrics. We can trust the results of the proposed metrics because the properties of their minimizers are guaranteed by Theorem 1.

From Table 5 we can observe that the considered learning methods do not exhibit significant difference in terms of modularity and informativeness. This result supports the theoretical finding of Locatello et al. [2019b] that unsupervised learning of disentangled representations by matching the distributions of observations is fundamentally impossible (see also Fig. 14) as well as their empirical finding that there is no evidence that learning disentangled representations in an unsupervised manner is reliable.

## E.4 Kendall tau distance between metrics

To analyze the relationship between these metrics, we report the Kendall tau distance [Kendall, 1938, Virtanen et al., 2020] averaged over experimental settings in Table 6. The Kendall tau distance is a correlation measure for ordinal data valued in $[-1, 1]$ which counts the number of pairwise disagreements between two ranking lists. Values close to 1 indicate strong agreement, and values close to $-1$ indicate strong disagreement.

From Table 6 we can observe that even though different metrics derived from the same logical definition may rank imperfect representations differently (see also Fig. 12), they still have positive correlations with each other, indicating that they measure the same property. The metrics proposed by Higgins et al. [2017] and Kim and Mnih [2018] have the highest correlations with each other (except for themselves), and we hypothesize that this is because they are both based on the pairwise distance

Table 6: Average Kendall tau rank distances bewteen disentanglement metrics

| | Modularity | | | | | Informativeness | | | | | Existing metrics | | | | | |
| | Product approx. | | | Constancy | | Retraction approx. | | | Contraction | | Pair | | Info. | Regressor | | |
| | Rad. | MAD | Var. | Diam. | MPD | ME | MAE | MSE | Max | Mean | Beta[a] | Factor[b] | MIG[c] | Dis.[d] | Com.[d] | Info.[d] |
|---|---|---|---|---|---|---|---|---|---|---|---|---|---|---|---|---|
| Rad. | 1.00 | 0.08 | 0.25 | 1.00 | 0.33 | −0.08 | 0.08 | 0.17 | 0.08 | 0.17 | −0.25 | 0.17 | 0.00 | 0.00 | 0.17 | −0.08 |
| MAD | 0.08 | 1.00 | 0.50 | 0.08 | 0.75 | 0.33 | 0.67 | 0.58 | 0.00 | 0.42 | −0.50 | −0.58 | −0.25 | 0.25 | 0.08 | −0.00 |
| Var. | 0.25 | 0.50 | 1.00 | 0.25 | 0.75 | 0.33 | 0.33 | 0.42 | −0.17 | 0.25 | −0.00 | −0.25 | −0.08 | 0.58 | 0.42 | 0.33 |
| Diam. | 1.00 | 0.08 | 0.25 | 1.00 | 0.33 | −0.08 | 0.08 | 0.17 | 0.08 | 0.17 | −0.25 | 0.17 | 0.00 | 0.00 | 0.17 | −0.08 |
| MPD | 0.33 | 0.75 | 0.75 | 0.33 | 1.00 | 0.25 | 0.42 | 0.50 | −0.08 | 0.33 | −0.25 | −0.33 | −0.17 | 0.33 | 0.17 | 0.08 |
| ME | −0.08 | 0.33 | 0.33 | −0.08 | 0.25 | 1.00 | 0.50 | 0.58 | 0.33 | 0.42 | −0.17 | −0.25 | −0.42 | 0.08 | −0.08 | 0.00 |
| MAE | 0.08 | 0.67 | 0.33 | 0.08 | 0.42 | 0.50 | 1.00 | 0.92 | 0.17 | 0.75 | −0.67 | −0.58 | −0.25 | 0.08 | −0.08 | −0.17 |
| MSE | 0.17 | 0.58 | 0.42 | 0.17 | 0.50 | 0.58 | 0.92 | 1.00 | 0.25 | 0.83 | −0.58 | −0.50 | −0.33 | 0.00 | −0.17 | −0.25 |
| Max | 0.08 | 0.00 | −0.17 | 0.08 | −0.08 | 0.33 | 0.17 | 0.25 | 1.00 | 0.42 | −0.33 | 0.08 | −0.42 | −0.25 | −0.42 | −0.33 |
| Mean | 0.17 | 0.42 | 0.25 | 0.17 | 0.33 | 0.42 | 0.75 | 0.83 | 0.42 | 1.00 | −0.75 | −0.50 | −0.33 | −0.17 | −0.33 | −0.42 |
| Beta | −0.25 | −0.50 | −0.00 | −0.25 | −0.25 | −0.17 | −0.67 | −0.58 | −0.33 | −0.75 | 1.00 | 0.58 | 0.25 | 0.25 | 0.25 | 0.50 |
| Factor | 0.17 | −0.58 | −0.25 | 0.17 | −0.33 | −0.25 | −0.58 | −0.50 | 0.08 | −0.50 | 0.58 | 1.00 | −0.17 | −0.17 | −0.17 | 0.08 |
| MIG | 0.00 | −0.25 | −0.08 | 0.00 | −0.17 | −0.42 | −0.25 | −0.33 | −0.42 | −0.33 | 0.25 | −0.17 | 1.00 | 0.17 | 0.33 | 0.08 |
| Dis. | 0.00 | 0.25 | 0.58 | 0.00 | 0.33 | 0.08 | 0.08 | 0.00 | −0.25 | −0.17 | 0.25 | −0.17 | 0.17 | 1.00 | 0.83 | 0.75 |
| Com. | 0.17 | 0.08 | 0.42 | 0.17 | 0.17 | −0.08 | −0.08 | −0.17 | −0.42 | −0.33 | 0.25 | −0.17 | 0.33 | 0.83 | 1.00 | 0.75 |
| Info. | −0.08 | −0.00 | 0.33 | −0.08 | 0.08 | 0.00 | −0.17 | −0.25 | −0.33 | −0.42 | 0.50 | 0.08 | 0.08 | 0.75 | 0.75 | 1.00 |

[a] [Higgins et al., 2017] [b] [Kim and Mnih, 2018] [c] [Chen et al., 2018] [d] [Eastwood and Williams, 2018]

Table 7: Computation time (seconds) of supervised disentanglement metrics on image datasets

| | Modularity | | | | | Informativeness | | | | | Existing metrics | | | |
| | Product approx. | | | Constancy | | Retraction approx. | | | Contraction | | Pair | | Info. | Regressor |
| | Rad. | MAD | Var. | Diam. | MPD | ME | MAE | MSE | Max | Mean | Beta[a] | Factor[b] | MIG[c] | DCI[d] |
|---|---|---|---|---|---|---|---|---|---|---|---|---|---|---|
| **3D Cars** [Reed et al., 2015] | 0.35 | 2.22 | 0.01 | 0.03 | 0.03 | 0.14 | 2.11 | 0.00 | 1.09 | 1.12 | 4.12 | 3.77 | 2.46 | 896.99 |
| **dSprites** [Matthey et al., 2017] | 0.47 | 4.04 | 0.00 | 0.03 | 0.03 | 0.51 | 3.84 | 0.00 | 1.36 | 1.37 | 7.00 | 6.37 | 11.51 | 353.04 |
| **3D Shapes** [Burgess and Kim, 2018] | 0.60 | 6.02 | 0.00 | 0.03 | 0.03 | 0.66 | 4.77 | 0.00 | 1.67 | 1.52 | 4.85 | 4.65 | 10.16 | 169.11 |
| **MPI3D** [Gondal et al., 2019] | 0.86 | 9.30 | 0.00 | 0.09 | 0.09 | 1.61 | 5.98 | 0.00 | 1.89 | 1.94 | 9.54 | 8.60 | 21.70 | 310.38 |

[a] [Higgins et al., 2017] [b] [Kim and Mnih, 2018] [c] [Chen et al., 2018] [d] [Eastwood and Williams, 2018]

approach. The DCI disentanglement metric [Eastwood and Williams, 2018] weakly agrees with the modularity metrics. However, the DCI informativeness metric [Eastwood and Williams, 2018] weakly disagrees with the informativeness metrics. It is possible that this is because of the different regressors (`sklearn.ensemble.GradientBoostingClassifier`, `sklearn.linear_model.LinearRegression`, and `sklearn.linear_model.QuantileRegressor` [Pedregosa et al., 2011]) used in predicting factors from codes, showing that random seeds and hyperparameters of the metrics may matter more than the models when additional predictors need to be trained to evaluate the learning methods. This result indicates the advantage of $q_{\text{injective}}$ over $q_{\text{retractable}}$.

However, it is important to note the limitations of these experimental results. Since the representations were learned from data and not fully controlled, it is possible that such results are due to the choices of datasets, learning algorithms, hyperparameters, and optimization errors. A high rank correlation coefficient between two metrics in this specific setting cannot guarantee that these metrics always measure the same property, or that they rank imperfect representations similarly in other settings. To gain a deeper understanding of these metrics, it is preferable to analyze their minimizers theoretically (Theorem 1) or test them in a fully controlled environment (Section 5).

### E.5 Computation time of metrics

Finally, we report the computation time of the considered metrics in Table 7 to support our claim that the proposed metrics are much faster than those that require training additional predictors and hyperparameter tuning. We can see that, in an extreme case, the calculation of the DCI metrics [Eastwood and Williams, 2018] using `GradientBoostingClassifier` [Pedregosa et al., 2011] takes around 15 minutes, while other metrics can be calculated within seconds. This computation time may be acceptable if the metrics are only used in the evaluation phase, but it is not feasible to use them as learning objectives even in derivative-free optimization.

Table 8: Factor-wise modularity metrics on **3D Cars** [Reed et al., 2015]

| | Product approx. | | | Constancy | |
|---|---|---|---|---|---|
| | Rad. | MAD | Var. | Diam. | MPD |
| **Elevation (4)** | | | | | |
| VAE | 0.49 | 0.87 | 0.97 | 0.24 | 0.90 |
| $\beta$-VAE | 0.52 | 0.86 | 0.97 | 0.27 | 0.90 |
| FactorVAE | 0.50 | 0.86 | 0.97 | 0.25 | 0.90 |
| $\beta$-TCVAE | 0.50 | 0.87 | 0.97 | 0.25 | 0.90 |
| **Azimuth (24)** | | | | | |
| VAE | 0.62 | 0.91 | 0.99 | 0.39 | 0.94 |
| $\beta$-VAE | 0.60 | 0.91 | 0.98 | 0.37 | 0.93 |
| FactorVAE | 0.57 | 0.90 | 0.98 | 0.32 | 0.93 |
| $\beta$-TCVAE | 0.65 | 0.91 | 0.99 | 0.42 | 0.94 |
| **Object (183)** | | | | | |
| VAE | 0.87 | 0.97 | 1.00 | 0.75 | 0.98 |
| $\beta$-VAE | 0.84 | 0.97 | 1.00 | 0.71 | 0.98 |
| FactorVAE | 0.85 | 0.97 | 1.00 | 0.72 | 0.98 |
| $\beta$-TCVAE | 0.86 | 0.97 | 1.00 | 0.75 | 0.98 |

## E.6 Factor-wise modularity metrics

An advantage of the proposed modularity metrics is that we can even evaluate each factor separately, which is impossible for those metrics that entangle modularity and informativeness. The results were reported in Tables 8 to 11. We found that some learning methods may outperform others on one factor but underperform on others, and different modularity metrics may rank learning methods differently (see also Appendix D.5).

For example, on **MPI3D** [Gondal et al., 2019] (Table 11), $\beta$-VAE [Higgins et al., 2017] has the highest scores (radius, MAD, variance, diameter, and MPD) on the horizontal and vertical axis factors, but the lowest scores (radius and diameter) on the object size, camera height, and background color factors. However, as measured by the MAD and MPD, it still has the highest scores on these factors. This means that $\beta$-VAE may generally encode the object size, camera height, and background color well compared to other considered methods, but has a small number of outliers. We believe that such fine-grained evaluation can guide the design of learning objectives, data collection, and further refinement of trained representation learning models.

Table 9: Factor-wise modularity metrics on **dSprites** [Matthey et al., 2017]

| | Product approx. | | | Constancy | |
|---|---|---|---|---|---|
| | Rad. | MAD | Var. | Diam. | MPD |
| Shape (3) | | | | | |
| VAE | 0.74 | 0.89 | 0.98 | 0.55 | 0.92 |
| $\beta$-VAE | 0.70 | 0.88 | 0.98 | 0.50 | 0.92 |
| FactorVAE | 0.73 | 0.89 | 0.98 | 0.53 | 0.92 |
| $\beta$-TCVAE | 0.72 | 0.88 | 0.98 | 0.53 | 0.92 |
| Scale (6) | | | | | |
| VAE | 0.71 | 0.90 | 0.98 | 0.51 | 0.93 |
| $\beta$-VAE | 0.55 | 0.88 | 0.98 | 0.30 | 0.92 |
| FactorVAE | 0.67 | 0.90 | 0.98 | 0.45 | 0.93 |
| $\beta$-TCVAE | 0.77 | 0.90 | 0.98 | 0.59 | 0.93 |
| Orientation (40) | | | | | |
| VAE | 0.92 | 0.97 | 1.00 | 0.84 | 0.98 |
| $\beta$-VAE | 0.86 | 0.98 | 1.00 | 0.74 | 0.98 |
| FactorVAE | 0.95 | 0.98 | 1.00 | 0.90 | 0.99 |
| $\beta$-TCVAE | 0.88 | 0.97 | 1.00 | 0.77 | 0.98 |
| Position X (32) | | | | | |
| VAE | 0.71 | 0.90 | 0.98 | 0.51 | 0.93 |
| $\beta$-VAE | 0.66 | 0.92 | 0.99 | 0.43 | 0.95 |
| FactorVAE | 0.62 | 0.89 | 0.98 | 0.39 | 0.92 |
| $\beta$-TCVAE | 0.67 | 0.91 | 0.99 | 0.44 | 0.94 |
| Position Y (32) | | | | | |
| VAE | 0.68 | 0.91 | 0.99 | 0.47 | 0.94 |
| $\beta$-VAE | 0.66 | 0.92 | 0.99 | 0.44 | 0.94 |
| FactorVAE | 0.64 | 0.90 | 0.98 | 0.41 | 0.93 |
| $\beta$-TCVAE | 0.67 | 0.90 | 0.98 | 0.45 | 0.93 |

Table 10: Factor-wise modularity metrics on **3D Shapes** [Burgess and Kim, 2018]

| | Product approx. | | | Constancy | |
|---|---|---|---|---|---|
| | Rad. | MAD | Var. | Diam. | MPD |
| Floor hue (10) | | | | | |
| VAE | 0.85 | 0.97 | 1.00 | 0.72 | 0.98 |
| $\beta$-VAE | 0.87 | 0.97 | 1.00 | 0.75 | 0.98 |
| FactorVAE | 0.75 | 0.93 | 0.99 | 0.57 | 0.95 |
| $\beta$-TCVAE | 0.99 | 1.00 | 1.00 | 0.97 | 1.00 |
| Wall hue (10) | | | | | |
| VAE | 0.85 | 0.97 | 1.00 | 0.73 | 0.98 |
| $\beta$-VAE | 0.94 | 0.99 | 1.00 | 0.87 | 0.99 |
| FactorVAE | 0.78 | 0.94 | 0.99 | 0.60 | 0.96 |
| $\beta$-TCVAE | 0.82 | 0.94 | 0.99 | 0.68 | 0.96 |
| Object hue (10) | | | | | |
| VAE | 0.77 | 0.95 | 0.99 | 0.59 | 0.96 |
| $\beta$-VAE | 0.75 | 0.92 | 0.99 | 0.56 | 0.95 |
| FactorVAE | 0.75 | 0.93 | 0.99 | 0.56 | 0.95 |
| $\beta$-TCVAE | 0.77 | 0.92 | 0.99 | 0.59 | 0.95 |
| Scale (8) | | | | | |
| VAE | 0.80 | 0.95 | 0.99 | 0.65 | 0.96 |
| $\beta$-VAE | 0.56 | 0.91 | 0.98 | 0.32 | 0.93 |
| FactorVAE | 0.79 | 0.93 | 0.99 | 0.63 | 0.95 |
| $\beta$-TCVAE | 0.80 | 0.95 | 1.00 | 0.63 | 0.97 |
| Shape (4) | | | | | |
| VAE | 0.59 | 0.91 | 0.98 | 0.35 | 0.93 |
| $\beta$-VAE | 0.62 | 0.91 | 0.98 | 0.38 | 0.93 |
| FactorVAE | 0.77 | 0.93 | 0.99 | 0.60 | 0.95 |
| $\beta$-TCVAE | 0.69 | 0.92 | 0.98 | 0.47 | 0.94 |
| Orientation (15) | | | | | |
| VAE | 0.95 | 0.99 | 1.00 | 0.91 | 0.99 |
| $\beta$-VAE | 0.94 | 0.99 | 1.00 | 0.89 | 0.99 |
| FactorVAE | 0.89 | 0.98 | 1.00 | 0.80 | 0.99 |
| $\beta$-TCVAE | 0.72 | 0.91 | 0.98 | 0.52 | 0.94 |

Table 11: Factor-wise modularity metrics on **MPI3D** [Gondal et al., 2019]

| | Product approx. | | | Constancy | |
|---|---|---|---|---|---|
| | Rad. | MAD | Var. | Diam. | MPD |
| Object color (6) | | | | | |
| VAE | 0.52 | 0.92 | 0.99 | 0.27 | 0.94 |
| $\beta$-VAE | 0.51 | 0.97 | 0.99 | 0.26 | 0.97 |
| FactorVAE | 0.66 | 0.94 | 0.99 | 0.43 | 0.96 |
| $\beta$-TCVAE | 0.57 | 0.92 | 0.99 | 0.33 | 0.94 |
| Object shape (6) | | | | | |
| VAE | 0.88 | 0.97 | 1.00 | 0.77 | 0.98 |
| $\beta$-VAE | 0.94 | 1.00 | 1.00 | 0.89 | 1.00 |
| FactorVAE | 0.93 | 0.99 | 1.00 | 0.87 | 0.99 |
| $\beta$-TCVAE | 0.95 | 1.00 | 1.00 | 0.91 | 1.00 |
| Object size (2) | | | | | |
| VAE | 0.70 | 0.93 | 0.99 | 0.49 | 0.95 |
| $\beta$-VAE | 0.63 | 0.98 | 1.00 | 0.40 | 0.98 |
| FactorVAE | 0.68 | 0.94 | 0.99 | 0.46 | 0.95 |
| $\beta$-TCVAE | 0.75 | 0.96 | 1.00 | 0.56 | 0.97 |
| Camera height (3) | | | | | |
| VAE | 0.48 | 0.87 | 0.97 | 0.23 | 0.90 |
| $\beta$-VAE | 0.34 | 0.95 | 0.99 | 0.11 | 0.96 |
| FactorVAE | 0.58 | 0.85 | 0.96 | 0.34 | 0.90 |
| $\beta$-TCVAE | 0.69 | 0.91 | 0.99 | 0.47 | 0.94 |
| Background color (3) | | | | | |
| VAE | 0.60 | 0.92 | 0.99 | 0.36 | 0.94 |
| $\beta$-VAE | 0.34 | 0.96 | 0.99 | 0.11 | 0.97 |
| FactorVAE | 0.76 | 0.94 | 0.99 | 0.58 | 0.96 |
| $\beta$-TCVAE | 0.59 | 0.93 | 0.99 | 0.34 | 0.95 |
| Horizontal axis (40) | | | | | |
| VAE | 0.72 | 0.93 | 0.99 | 0.52 | 0.95 |
| $\beta$-VAE | 0.74 | 0.99 | 1.00 | 0.55 | 0.99 |
| FactorVAE | 0.69 | 0.92 | 0.99 | 0.47 | 0.94 |
| $\beta$-TCVAE | 0.73 | 0.94 | 0.99 | 0.54 | 0.96 |
| Vertical axis (40) | | | | | |
| VAE | 0.67 | 0.91 | 0.99 | 0.45 | 0.94 |
| $\beta$-VAE | 0.75 | 0.99 | 1.00 | 0.56 | 0.99 |
| FactorVAE | 0.73 | 0.93 | 0.99 | 0.53 | 0.95 |
| $\beta$-TCVAE | 0.60 | 0.93 | 0.99 | 0.36 | 0.95 |

