# OpenReview forum: "Enriching Disentanglement: From Logical Definitions to Quantitative Metrics"
_NeurIPS.cc/2024/Conference — NeurIPS 2024 poster_

### Official Review · Reviewer_sy6j · 2024-06-30

**Soundness:** 2
**Presentation:** 1
**Contribution:** 3
**Rating:** 5
**Confidence:** 3

**Summary:**

This paper investigates relating logical definitions of disentanglement and existing quantitative metrics via formal derivations of novel metrics.

**Strengths:**

The topic is very interesting and it seems the authors were very rigorous in their investigations especially concerning the amount of all of the background material and derivations.

**Weaknesses:**

However, the paper is very poorly motivated, structured and written. It is very difficult to follow the authors along what they are trying to achieve (the motivation), what they are doing (their ideas) and the details on how this is connected with other research (context via related work) and how this should be used for future ML research/ what the significance is of their work for the future. E.g. the Introduction dives right into formal background notations without any motivation/overview of what the authors wish to achieve/investigate in the work.

Maybe I have misunderstood some things, but what exactly are the results of the section 3? I.e which of the many metrics are the relevant ones and what do they mean intuitively? The authors provide many  examples of specific “implementations” (for lack of a better word here) and then conclude by statements such as “Upon analyzing the metrics above, it becomes evident that what we need is not the best approximation  itself (e.g., the mean) but rather the approximation error “ on page 7. So it seems the previous derivations are irrelevant for the overall goal (what this is is further unclear). So I wonder whether we really require this material then and can remove it, e.g., for the sake of more details on experiments (see below).

Overall, I noticed the supplementary materials has much more relevant information that I believe needs to be in the main text. E.g. related works as these are necessary to understand the context of the work.

Further, it was also very difficult for me to understand the experiments of the main paper. The experimental setup is not described in the main paper (there is not even a caption to Table 2). This needs to be greatly improved. There seem to be a lot of information in the supplementary concerning experiments and I would suggest moving this into the main paper. Overall, it would help me if the authors could summarize again what they were investigating with these experiments.

The claims made in the experiments of the main paper are not backed up by the experiment shown in Table 2, e.g. "The metrics derived from equivalent definitions may differ in terms of computation cost and differentiability.". Where are the results for this? If these can only be found in the supplementary currently, I think they should also be moved to the main text.

I am also puzzled by all of the experiments in the supplementary material (on interesting datasets), but that are not mentioned in the main text (unless I have missed this). Was there a reason for this?

Lastly, I am missing a thorough discussion of the overall findings and particularly potential limitations of the findings/proposed metrics.

Overall, it is too difficult to assess the quality of the work and proposed ideas with the current structure. I would recommend putting some time into rewriting and restructuring to make this more clear to the reader.

**Questions:**

see above

**Limitations:**

see above

---

> ### Author Rebuttal · Authors · 2024-08-06
>
> Thank you very much for your detailed review of our work!
> We'd like to address your concerns as follows.
>
> ---
>     However, the paper is very poorly motivated, structured and written. It is very difficult to follow the authors along what they are trying to achieve (the motivation), what they are doing (their ideas) and the details on how this is connected with other research (context via related work) and how this should be used for future ML research/ what the significance is of their work for the future. E.g. the Introduction dives right into formal background notations without any motivation/overview of what the authors wish to achieve/investigate in the work.
> ---
>
> Please allow us to clarify the motivation and why we chose the current presentation.
>
> One of the major problems in the current research of disentangled representation learning is that we lack a clear, logical definition of properties such as modularity and informativeness, and **we don't know if a metric truly quantifies a property or not** (abstract `l.4`).
> This is the research problem we want to address in this paper.
>
> We started the introduction section with a discussion on the parallel between loss function and function equality, which all machine learning researchers are familiar with, to show our goal: "**measuring and optimizing other properties like this**" (`l.29`) *by connecting logical definitions and quantitative metrics and extending this parallel to other properties*.
> This is the main idea of this paper.
>
> We used informativeness as an example in Section 1.1, which is an important property in (disentangled) representation learning.
> We used this example to show that there could be multiple ways (e.g., injectivity and left-invertibility) to formally define a property (e.g., informativeness), and we need to deal with **logical operations** (e.g., implication), **quantifiers** (e.g., universal quantifier and existential quantifier), composition, and identity, which are the **basic building blocks** that will be used in the following sections.
>
> We then contextualized our work in Section 1.2 (and further explained related work in detail in Appendix D), gave a logical definition of an essential property of disentanglement (modularity), provided a concrete example to help the reader understand this concept, and introduced the research questions we want to answer: **how to find a (preferably differentiable) metric for a logically defined property**.
> Due to the page limit, we had to move the detailed discussions on related work to Appendix D.
> If we have a chance, we will move some of them back to the introduction section.
>
> In Section 2, we immediately answered the questions we asked, and we summarized the proposed technique in **Table 1**.
> The "meta-theorem" was stated in **Theorem 1**.
> The rest of this paper is just explanations and concrete instantiations of this technique.
> We tried our best to present this work in a way that a machine learning researcher may get the gist of this work by reading the first three pages, and a practitioner can implement the derived metrics using the technique summarized in Table 1.
>
> Therefore, the introduction section is not just the notation; We chose the examples very carefully to show the **goal, motivation, basic concepts, research questions, and our main idea**.
> We are aware that this may not be a conventional way to present a work in the machine learning community, but we believe that the current "*let the math speak for itself*" style is more suitable for this work and straight to the point.
> We are glad that `Reviewer GqiT` stated that `The paper is well-written. The math objects are introduced with good intuitions`, but we also know that this writing style is not for every reader.
> We will try our best to further improve the readability of the introduction section for the machine learning audience.
>
> ---
>     Maybe I have misunderstood some things, but what exactly are the results of the section 3?
> ---
>
> Section 3 demonstrated the use of the technique proposed in Section 2 and presented the modularity metrics (**Propositions 2 and 3**), their concrete instantiations (**Eqs. (17) and (25)**), informative metrics (**Definitions 8 and 9**), and an example of Theorem 1 (**Eq. (28)**).
>
> "We need the approximation error but not the approximation itself" can be interpreted as "we can calculate the variance without calculating the mean".
> **Metrics derived from logically equivalent definitions and using different aggregators may have different computational costs, sensitivity to outliers, and learning dynamics.**
> They are all relevant and useful in different scenarios (e.g., robustness evaluation, gradient-based optimization), and further research is needed to understand their characteristics.
> Please see Appendix E.3 for further explanation.
>
> ---
>     Further, it was also very difficult for me to understand the experiments of the main paper.
> ---
>
> The experiments in the main text were meant to confirm Theorem 1: **minimizers of the derived metrics must satisfy the properties the metrics quantify**.
> We focused on the theoretical exposition of our work, and the experiments were indeed supplementary to demonstrate the benefits of the derived metrics.
> We believe that the proposed logic-metric theory is our main contribution, which was supported by the formal proofs.
> Nevertheless, we will follow your suggestions and clarify the meaning of experiments by moving some materials from the supplementary material to the main text in the revised version.
> Thank you!

---

> > ### Comment · Reviewer_sy6j · 2024-08-10
> >
> > Thanks for your detailed response and your patience to explain these things. I have now understood better what the work is about and that I had misunderstood a few things after the first read. However, I stand by the point that it takes very long (within the text) until the reader understands what the motivation is of the work. I.e. the goal is to investigate “we don't know if a disentanglement metric truly quantifies a property or not”. I do not agree with the authors that one requires the sections until section 1.2 to understand this. I would rather put in related work into the main paper and start the introduction from an updated version of section 1.2, i.e. rather than start the motivation from very basic definitions whereby the reader does not know where these are leading to.
> >
> > Overall, I understand the work now and find the contributions sufficient to justify to raise my score. However, I really recommend the authors restructure/rewrite the motivation, etc, to make it easier for any reader to understand what the goal is upfront rather than after several formalizations which are difficult to follow without knowing the goal.

---

> > > ### Author Response · Authors · 2024-08-11
> > >
> > > Thank you very much for your reply and for raising your score! We are glad that our explanation was helpful.
> > >
> > > We agree with you that the current introduction might be confusing on a first read for some readers. If granted an additional page, we will follow your suggestions by **clearly stating our goal and motivation in plain words** before explaining them in technical and mathematical terms. This approach should make the content more accessible to a broader audience.
> > >
> > > Your suggestions from a reader's perspective have greatly helped us improve the readability of our paper. We sincerely appreciate your valuable feedback!

---

### Official Review · Reviewer_GEk9 · 2024-07-03

**Soundness:** 3
**Presentation:** 3
**Contribution:** 3
**Rating:** 6
**Confidence:** 2

**Summary:**

The paper consider the connection between logical definitions and quantitative metrics, proposing a systematic approach to design metrics from logical definitions. Particularly, the paper is focused on the measure of disentanglement. The paper theoretically justifies the correspondence between logical definitions of disentanglement and quantitative metrics via topos theory and enriched category theory and empirically demonstrates the effectiveness in isolating various aspects of disentangled representations compared to existing metrics.

**Strengths:**

The idea presented in the paper, especially the systematic method of converting logical predicates into quantitative metrics, is novel and very interesting. The authors also theoretically support this idea with topos theory and enriched category theory, which are sophisticated and advanced mathematical frameworks not commonly employed in machine learning literature.

**Weaknesses:**

The superiority of the proposed metrics over the the existing ones does not seem fully validated, either empircally or theoretically. The evidence provided does not show the proposed metrics dominate the existing ones universally.

**Questions:**

Could you provide more evidence, either empirical or theoretical, to show the proposed metrics dominate the existing ones universally? Or if the proposed metrics do not outperform the existing ones universally, when are they better?

---

> ### Author Rebuttal · Authors · 2024-08-06
>
> Thank you for your appreciation of the novelty and soundness of this work!
> We'd like to address your concerns as follows.
>
> ---
>     The superiority of the proposed metrics over the the existing ones does not seem fully validated, either empircally or theoretically. The evidence provided does not show the proposed metrics dominate the existing ones universally.
> ---
>
> Please let us summarize the benefits of the derived metrics over the existing ones:
>
> ### **No failure modes**
>
> First, the biggest advantage is that the derived metrics are rooted in the logical definitions of the property we want to quantify and are governed by **Theorem 1**. Therefore, we know that minimizing these metrics won't lead to wrong answers. In contrast, several existing metrics have "failure modes", which means that these metrics can wrongly score representations (see `Carbonneau et al., [2022]`, `Mahon et al., [2023]`). This was theoretically supported by **Theorem 1** and empirically validated in **Table 2**.
>
> ### **Differentiability**
>
> In the literature, a new evaluation metric is often introduced along with a new representation learning method `[Carbonneau et al., 2022]`, but it is usually unproven that the method can optimize the new metric. One reason is that many existing metrics are not differentiable (because, e.g., they need to train a predictor `[Higgins et al., 2017, Kim and Mnih, 2018, Eastwood and Williams, 2018]`) so we cannot directly optimize them. By choosing specific aggregators, we can obtain differentiable metrics for a property, which allows us to directly optimize the property using gradient-based optimization. This was supported by the existence of differentiable metrics in **Eqs. (17), (25), and (27)**.
>
> ### **Weakly supervised evaluation**
>
> We revealed that it is possible to evaluate some modularity metrics using only similarity information because the modularity is invariant to bijections. This is impossible for some existing metrics. We demonstrated weakly supervised modularity metrics in **Appendix F.2**.
>
> ### **Efficiency**
>
> Some existing metrics such as DCI need to train an additional predictor, which requires hyperparameter tuning and is usually computationally expensive. High computation cost may be acceptable if the metrics are only used in the evaluation phase, but it is not feasible to use them as learning objectives even in derivative-free optimization. In contrast, the derived metrics are more efficient. This was empirically supported by **Table 7** in **Appendix F.5**.
>
> ### **Fine-grained evaluation**
>
> We proposed to evaluate modularity and informativeness separately. **Table 2** shows that existing single-score metrics cannot distinguish modularity and informativeness, which provides less information about the representations. We further demonstrated how to use the proposed metrics to diagnose issues of representation learning methods in a more fine-grained manner in **Appendix F.6**.
>
> ---
>
> We will further clarify the benefits of the proposed metrics in the revised version.
>
> Please let us know if you have any other questions or suggestions.
> Thank you!

---

> ### Comment · Reviewer_GEk9 · 2024-08-11
>
> Thank you for your detailed response, which addresses some of my concerns. I will maintain my current score.

---

> > ### Author Response · Authors · 2024-08-11
> >
> > Thank you for your reply! We are glad that our explanation addressed some of your concerns.
> >
> > Due to the page limit, some of the advantages of the derived metrics were only fully explained in the appendix. We will **summarize the benefits of the proposed method** and emphasize them in the revised version. Thank you for your feedback!

---

### Official Review · Reviewer_4yVc · 2024-07-11

**Soundness:** 2
**Presentation:** 2
**Contribution:** 2
**Rating:** 5
**Confidence:** 2

**Summary:**

The paper proposed to establish a connection between logical definitions of disentanglement and quantitative metrics from the perspective of typos theory and category theory. It then propose a metrics for disentanglement with stronger theoretical guarantees and compared it with some state of the art metrics.

**Strengths:**

The paper provided theoretical justification for establishing metrics from typos theory and category theory in disentanglement of representation learning. It then proposed to convert first-order predicate into real-value quantity, which is innovative. The proposed metrics also have practical implications as the author mentioned differentiability and its effectiveness through some experimental results.

**Weaknesses:**

1, Despite that the authors included as much background information in the appendix, the advanced mathematical concepts makes the paper very hard to follow and limit its accessibility. Moreover, the theoretical connections may not be readily feasible to practical and intuitive interpretations. Hence its use scenarios should be stated.
2, Even though the paper provides empirical results, its scope and effectiveness may not be fully demonstrated through the limited range of scenarios considered. Therefore its generalizability it not clear.

**Questions:**

1, why were the specific modularity metrics used for evaluation? How do they contribute to assessing disentanglement in representation learning?
2, Line 308 mentioned that the results are transformed isomorphically using $e^{-x}$. What is the purpose of this procedure?

**Limitations:**

The authors adequately addressed the limitations

---

> ### Author Rebuttal · Authors · 2024-08-06
>
> Thank you for acknowledging the innovation of our work!
> We will answer your questions as follows.
>
> ---
>     ... the advanced mathematical concepts makes the paper very hard to follow and limit its accessibility.
> ---
>
> Thank you for pointing this out.
> We developed the theory with the help of these mathematical concepts, and we are aware that they are not as well known as other tools such as linear algebra and statistics for machine learning researchers. Therefore, in the main body of this paper, we tried to present the proposed technique (**Table 1**) and theorem (**Theorem 1**) without these terms. We simplified the complex structures into easy-to-follow rules, such as "*replacing the conjunction (logical AND) with the addition*", and used many examples to demonstrate the application of this technique. We believe that the reader does not need any category theory background to understand the proposed metrics.
>
> To make the theory more accessible, we added a **non-categorical restatement** of the proposed theory in the revised version, which is less general but easier to follow. However, we have to introduce some necessary algebraic concepts such as *homomorphism*. We hope this part can help the reader understand not only the proposed technique but also the theory behind it.
>
> ---
>     1, why were the specific modularity metrics used for evaluation? How do they contribute to assessing disentanglement in representation learning?
> ---
>
> We evaluated not only modularity but also informativeness (Section 3.3, the middle 5 columns in Table 2). Modularity is arguably the essential property of disentanglement (it is simply called *disentanglement* in `Eastwood & Williams [2018]`), informativeness is a basic property of representation learning, and they are considered to be more important than other properties such as completeness/compactness in practice `[Ridgeway and Mozer 2018, Duan et al. 2020, and Carbonneau et al. 2022]`.
>
> ---
>     2, Line 308 mentioned that the results are transformed isomorphically using e^{-x}. What is the purpose of this procedure?
> ---
>
> Originally, we used ($[0, \infty]$-valued) strict premetrics as the quantitative versions of equality. In this case, $0$ means a property holds, and non-zero numbers mean that a property does not hold (*the lower the better*). However, many existing metrics used $[0, 1]$-valued metrics, where $1$ means a property holds perfectly (*the higher the better*). A reader may feel a subtle cognitive dissonance if we compare them directly. Therefore, we transformed $[0, \infty]$-valued metrics into a $[0, 1]$-valued metrics using $e^{-x}: [0, \infty] \to [0, 1]$.
> We say it's an isomorphism in the sense that it preserves the order ($a \leq b$ implies $e^{-a} \geq e^{-b}$), quantitative operations such as the addition ($e^{-(a + b)} = e^{-a} \cdot e^{-b}$), and so on.
> If we don't need to compare the results with existing $[0, 1]$-valued metrics, we can use the original values (e.g., for optimization).
>
> ---
>
> We hope these answers address your concerns.
>
> Please let us know if you have any other questions.
> Thank you!

---

> > ### Comment · Reviewer_4yVc · 2024-08-11
> >
> > Thanks for the detailed response and I think all my concerns are addressed. I have raised my score now.

---

> > > ### Author Response · Authors · 2024-08-11
> > >
> > > Thank you for your reply and for raising your score!
> > >
> > > We will address your concerns in the revised version, and we hope the simplified theory can make this paper easier to follow. Thank you for sharing your insights and questions!

---

### Official Review · Reviewer_GqiT · 2024-07-12

**Soundness:** 3
**Presentation:** 3
**Contribution:** 2
**Rating:** 5
**Confidence:** 2

**Summary:**

This study introduces a systematic approach to quantify properties of representation learning models. By translating logical definitions into quantitative metrics, the paper evaluates two key properties: modularity and informativeness. Two sets of metrics are derived for each property, one based on approximation and the other on distance computation and aggregation. Theoretical analysis of these metrics is conducted, including examination of their minimizers.

(Full disclosure: I have reviewed this paper before)

**Strengths:**

1. I think this paper is interesting and innovative and mathematically sound.
2. The paper is well-written. The math objects are introduced with good intuitions.

**Weaknesses:**

1. It might be difficult for people with less prior knowledges to read.
2. It’s unclear if the metrics is practical be to a loss to optimize in practice because the experiments do not train with proposed metrics.

**Questions:**

1. Heyting algebra is mentioned a couple of times in the paper without a proper definition, which would be good for general ML audience.

---

> ### Author Rebuttal · Authors · 2024-08-06
>
> Thanks again for your kind evaluation of our work!
> Regarding your concerns, our answers are as follows.
>
> ---
>     It might be difficult for people with less prior knowledges to read.
> ---
>
> Thank you for pointing this out.
> In the main body of this paper, we tried our best to avoid using abstract categorical concepts, which were necessary for developing the theory. The proposed technique was explained as simple rules such as "*replacing the conjunction (logical AND) with the addition*". We believe that the reader does not need any category theory background to understand the proposed metrics.
>
> To make the theory more accessible, we added a **non-categorical restatement** of the proposed theory in the revised version, which is less general but easier to follow. However, we have to introduce some necessary algebraic concepts such as homomorphism. We hope this part can help the reader understand not only the proposed technique but also the theory behind it.
>
> ---
>     It’s unclear if the metrics is practical be to a loss to optimize in practice because the experiments do not train with proposed metrics.
> ---
>
> Learning disentangled representations with the proposed metrics is indeed the immediate step after this work. We would like to point out that some existing metrics are much less practical as learning objectives for two reasons:
>
> 1. Some metrics such as DCI `[Eastwood and Williams, 2018]` need to train an additional predictor, which requires hyperparameter tuning and more computation. Evaluation using these metrics is time-consuming. In some cases, the calculation of the DCI metrics using `GradientBoostingClassifier` takes around 15 minutes (See **Appendix F.5**). It is not feasible to use them as learning objectives, even in derivative-free optimization.
> 2. For the same reason, many existing metrics are not differentiable, so it is impossible to use them in gradient-based optimization.
>
> To use the proposed metrics to learn representations, we need to consider what supervision can be used, which is a different topic and beyond the scope of this work.
>
>
> ---
>     Heyting algebra is mentioned a couple of times in the paper without a proper definition, which would be good for general ML audience.
> ---
>
> We mentioned Heyting algebra and quantale so that a reader with an algebra background can immediately understand our main idea. It is like saying *group* instead of *a set equipped with an associative, unital, and invertible binary operation*. However, we believe that a machine learning researcher who is not familiar with algebra can still understand the proposed technique because we spelled out all the components of a Heyting algebra (true, false, conjunction, implication, etc.). Roughly speaking, a Heyting algebra may be considered as a **Boolean algebra** without excluded middle. For interested readers, the formal definition of Heyting algebra is given in **Appendix B**.
>
> ---
>
> We hope we addressed your concerns.
>
> Please let us know if you have any other questions.
> Thank you!

---

> > ### Comment · Reviewer_GqiT · 2024-08-12
> >
> > Thanks for you reply. I am still keeping my score because I have low confidence in the topic but I am leaning towards acceptance.

---

> > > ### Author Response · Authors · 2024-08-12
> > >
> > > Thank you for your reply and for confirming your positive opinion about this paper!
> > >
> > > We understand that the technical details may be inaccessible to some readers.
> > > We conjecture that this may be due to three reasons:
> > >
> > > - While logic is fundamental in math and machine learning, the *algebraic approach* to logic --- where a predicate $p: A \to \\{\top, \bot\\}$ is considered as a function from a set $A$ to the set $\\{\top, \bot\\}$ of truth values, and logical operations like conjunction $\land: \\{\top, \bot\\} \times \\{\top, \bot\\} \to \\{\top, \bot\\}$ are viewed as binary operations over this set --- is less familiar to many.
> > > - It is even less well known that the universal quantifier $\forall: \\{\top, \bot\\}^A \to \\{\top, \bot\\}$ can be viewed as a function from a set $\\{\top, \bot\\}^A$ of predicates (i.e., functions) to the set of truth values, and this function is also a kind of [algebraic structure](https://en.wikipedia.org/wiki/F-algebra). However, this algebraic perspective is valuable because it allows us to formulate the relationship between quantifiers and aggregators (e.g., sup or sum $[0, \infty]^A \to [0, \infty]$) as a homomorphism (e.g., *all values are zero if and only if the sup/sum value is zero*).
> > > - Our logic-metric theory is general and applicable to various problems, but its compositional nature may render it too abstract for some readers.
> > >
> > > Despite these technical challenges, we still think this logical and algebraic perspective is worth sharing with the machine learning community. To make it more accessible, we will **simplify the presentation of the theory and minimize the prerequisites** in the revised version. Thanks again for reviewing this work!

---

### Decision · Program_Chairs · 2024-09-25

**Decision:**

Accept (poster)

**Comment:**

This paper is a borderline case.

The positive end is that the authors are taking a very different approach to studying the notion of disentanglement. In general, I think (and the reviewers agreed) that such creative, "large step" research is valuable.

A particular aspect of such research is that one paper often cannot fully explore the direction. That means that, as some reviewers noted, the empirical significance is questionable, and indeed it's somewhat unclear what the methodological implications may be. Generally then, the paper relies on its mathematical development for a priori motivation of the approach (rather than testing predictions as a posteriori motivation). I think this is fine for a first paper on a novel direction.

The trouble is that no reviewer feels qualified to assess the mathematical content. Indeed, I also do not feel qualified.

I am recommending acceptance on the basis that novel and ambitious approaches to hard problems should be encouraged. However, I emphasize to the authors that for this line of work to have impact, significant follow up work will be required to "bring it down from the mountain" and convincingly demonstrate its practical importance.